# Potentiating cancer immunotherapies with modular albumin-hitchhiking nanobody–STING agonist conjugates

Blaise R. Kimmel [1,2], Karan Arora [1], Neil C. Chada [3], Vijaya Bharti[1], Alexander J. Kwiatkowski [1], Jonah E. Finkelstein[1], Ann Hanna[4], Emily N. Arner [4], Taylor L. Sheehy [3], Lucinda E. Pastora[1], Jinming Yang[5], Hayden M. Pagendarm [3], Payton T. Stone [1], Ebony Hargrove-Wiley [5], Brandie C. Taylor [4], Lauren A. Hubert[1], Barbara M. Fingleton[5], Katherine N. Gibson-Corley [6], Jody C. May [7,8,9,10], John A. McLean [7,8,9,10], Jeffrey C. Rathmell [9,11,12,13], Ann Richmond [5,9,10], W. Kimryn Rathmell [4,9,12,13], Justin M. Balko [4,9,11,12,13] & John T. Wilson [1,3,9,11,12,13,14] ✉

The enhancement of antitumour immunity via agonists of the stimulator of interferon genes (STING) pathway is limited by pharmacological barriers. Here we show that the covalent conjugation of a STING agonist to anti-albumin nanobodies via site-selective bioconjugation chemistries prolongs the circulation of the agonist in the blood and increases its accumulation in tumour tissue, stimulating innate immune programmes that increased the infiltration of activated natural killer cells and T cells, which potently inhibited the growth of mouse tumours. The technology is modular, as demonstrated by the recombinant integration of a second nanobody domain targeting programmed death-ligand 1 (PD-L1), which further increased the accumulation of the agonist in tumours while blocking immunosuppressive PD-1/PD-L1 interactions. The bivalent nanobody–STING agonist conjugate stimulated robust antigen-specific T-cell responses and long-lasting immunological memory and conferred enhanced therapeutic efficacy. It was also effective as a neoadjuvant treatment to adoptive T-cell therapy. As a modular approach, hitchhiking STING agonists on serum albumin may serve as a broadly applicable strategy for augmenting the potency of systemically administered cancer immunotherapies.

Immune checkpoint inhibitors (ICIs) targeting cytotoxic T-lymphocyte-associated protein 4 (CTLA-4) and programmed cell death protein 1 (PD-1) or programmed death-ligand 1 (PD-L1) receptors have revolutionized the treatment of an increasing number of cancers but are still only effective for a relatively small fraction of patients (~15%)[1]. For many cancers, this can be partially attributed to poor tumour immunogenicity and an immunosuppressive (that is, 'cold') tumour microenvironment (TME) that restricts the infiltration and/or function of antitumour T cells[2,3]. The innate immune system plays a critical role in cancer immune surveillance[4], with clinical evidence linking activation of certain pattern recognition receptor (PRR) signalling pathways to increased T-cell infiltration and responses to ICIs in cancer patients[5,6]. Accordingly, the relationship between innate and adaptive antitumour immunity has motivated the clinical exploration and continued development of agonists targeting PRRs, including toll-like receptors (TLRs), retinoic acid-inducible gene-I (RIG-I)-like

**Fig. 1 | Design, synthesis and in vitro characterization of an anti-albumin nanobody for site-selective conjugation of STING agonists. a**, Scheme depicting the concept of an albumin-hitchhiking nanobody–STING agonist conjugate for cancer immunotherapy. Anti-albumin nanobodies conjugated to STING agonists bind to circulating albumin in situ, resulting in improved pharmacokinetics and increased biodistribution to tumour sites that stimulates antitumour innate and adaptive immune responses. **b**, Computational model of the anti-albumin nanobody (nAlb) binding at domain IIB of HSA. **c**, ITC traces (top) and binding isotherms (bottom) of nAlb binding to human and mouse serum albumin at pH 7.5 with calculated dissociation constant ($K_d$). **d**, Reaction scheme for generating molecularly homogeneous nAlb conjugates through site-selective enzymatic ligation of an amine-PEG$_3$-azide followed by conjugation of agonist or dye cargo through strain-promoted azide-alkyne cycloaddition (SPAAC). **e**, Structure of diABZI STING agonist conjugated to a DBCO-PEG$_{11}$ handle for ligation to azide-functionalized nanobodies via SPAAC. **f,g**, ESI–MS (**f**) and SDS–PAGE (**g**) showing nanobody conjugate purity and molecular weight (see Source Data for uncropped gel in ref. 90). **h,i**, Dose–response curves in A549-Dual ($n = 3$) (**h**) and THP1-Dual type I interferon reporter cell lines ($n = 3$) (**i**) with estimated EC$_{50}$ values indicated in the legends; RLU, relative light unit. **j**, qPCR analysis of gene expression in mouse BMDMs treated in vitro with 0.25 μM of free diABZI or nAlb–diABZI conjugate ($n = 3$). $P$ values determined by one-way ANOVA with Dunnett's multiple comparison test with groups compared to PBS. Replicates are biological, and data are shown as mean ± s.e.m. Panel **a** created with BioRender.com.

receptors (RLRs) and stimulator of interferon genes (STING). Activation of these pathways can induce a coordinated antitumour immune response by triggering the production of type I interferons, proinflammatory cytokines, chemokines, costimulatory molecules and other mediators that potentiate T-cell responses and enhance the efficacy of ICIs[4,7,8]. PRR agonists have typically been administered intratumourally as an 'in situ vaccine' with the intent of stimulating a systemic adaptive immune response that mediates distal tumour regression and/or immune memory to protect against disease recurrence[2,9]. While promising, intralesional therapy may not be feasible or practical for patients with metastatic, poorly accessible tumours, particularly for repeated dosing[10]. It is worth noting that intratumoural administration has thus far yielded underwhelming outcomes in the clinic[11], motivating a need for systemically administered therapies targeting PRR agonists.

Among the PRRs, STING has emerged as one of the most promising targets for stimulating antitumour innate immunity[12–14], with remarkable efficacy in preclinical models leading to clinical trials of a growing arsenal of STING-activating therapeutics[15,16]. However, clinical exploration of STING agonists has been primarily restricted to intratumoural administration of cyclic dinucleotides (CDN) and, unfortunately, has yielded disappointing results[17]. This can be partially attributed to both the aforementioned limitations of intratumoural administration and the poor drug-like properties of CDNs—anionic small molecules—that limit their activity and efficacy for systemic administration[15]. This challenge has prompted the development of several promising nanoparticle-based drug carriers for CDNs[15,18–22] as well as small molecule STING agonists with improved chemical properties for systemic administration[15,23,24]. However, therapeutic targeting of STING remains a considerable challenge owing to multiple intertwined pharmacological barriers, including suboptimal pharmacokinetics and poor tumour accumulation, that limit efficacy and increase the risk of inflammatory toxicities[15,25]. Hence, there is a need for drug delivery technologies that afford increased spatiotemporal control over the delivery of systemically administered STING agonists for the treatment of advanced and metastatic disease.

Here we present the development of a modular drug delivery technology for safe and effective systemic administration of STING agonists based on the concept of 'albumin-hitchhiking'[26]. Albumin

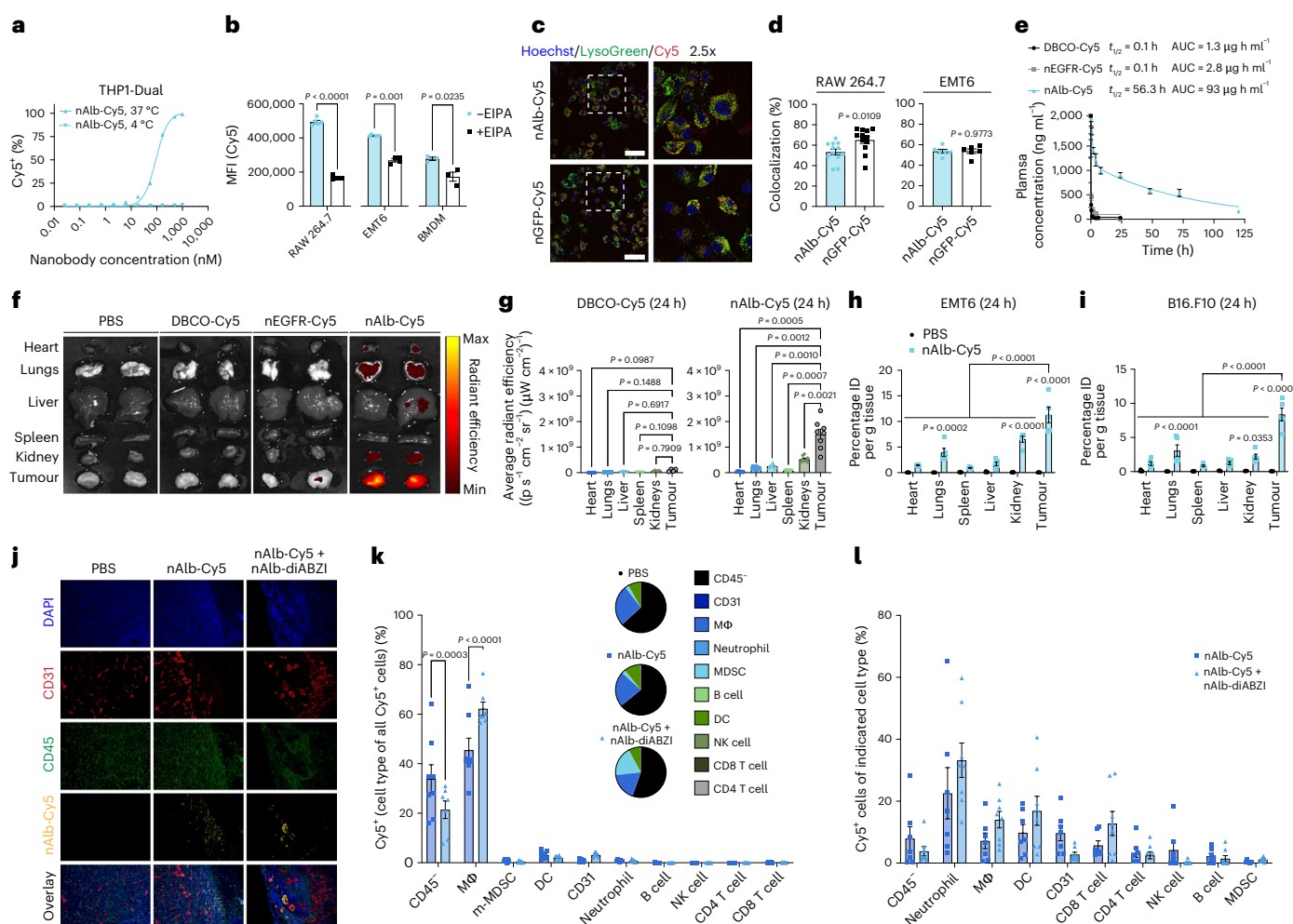

**Fig. 2 | Anti-albumin nanobodies increase cargo delivery to tumour sites to promote uptake by cancer cells and tumour-associated myeloid cells.** **a**, Representative dose–response curves for nanobody–Cy5 conjugate surface binding and intracellular uptake at 37 °C and 4 °C measured by flow cytometry in THP-1 cells in vitro. **b**, MFI (Cy5) of RAW 264.7 ($n = 5$), EMT6 ($n = 4$) and BMDM ($n = 3$) cells treated with nAlb–Cy5 (2 μM) with (+EIPA) or without (−EIPA) the macropinocytosis inhibitor EIPA. $P$ values determined by two-sided Student's $t$-test. **c**,**d**, Representative confocal micrographs showing colocalization of Cy5 (red) with lysotracker green (green) in RAW 264.7 cells; Hoechst nuclear stain (blue) (scale bars, 100 μm) (**c**) with percentage colocalization determination for nAlb–Cy5 and nGFP–Cy5 in RAW 264.7 ($n = 9$) and EMT6 ($n = 6$) cells (**d**). $P$ values determined by two-sided Student's $t$-test. **e**, Pharmacokinetics of free DBCO–Cy5 dye and indicated nanobody–Cy5 conjugates injected intravenously at 2 mg kg$^{-1}$ in healthy female C57BL/6 mice ($n = 5$). Elimination phase half-life and AUC are indicated in the legend. **f**,**g**, Representative IVIS fluorescence images of excised tumours and major organs (**f**) and quantification of average radiant efficiencies 24 h following intravenous administration of DBCO–Cy5 ($n = 5$) and nAlb–Cy5 ($n = 8$) at 2 mg kg$^{-1}$ to female Balb/c mice with orthotopic EMT6 breast tumours (**g**). $P$ values determined by one-way ANOVA with Dunnett's multiple comparison test with each organ compared to tumour. **h**,**i**, Quantification of percentage of injected dose per gram of tissue (%ID per g) 24 h following intravenous administration of nAlb–Cy5 at 2 mg kg$^{-1}$ or PBS (vehicle) to female Balb/c mice with orthotopic EMT6 breast tumours ($n = 5$) (**h**) and female C57BL/6 mice with subcutaneous B16.F10 tumours ($n = 5$) (**i**). $P$ values determined by two-way ANOVA with post hoc Tukey's correction for multiple comparisons. **j**, Representative fluorescence microscopy images of EMT6 tumour sections stained for DAPI (blue), CD45 (green) and CD31 (red) 24 h following administration of nAlb–Cy5 (yellow) alone or in combination with nAlb–diABZI. Scale bars, 200 μm. **k**,**l**, Flow cytometric analysis of nAlb–Cy5 cellular uptake in EMT6 tumours evaluated as the percentage of indicated cell type comprising all Cy5$^{+}$ live cells (**k**) or as the percentage of Cy5$^{+}$ cells (cell type of all Cy5$^{+}$ cells) within an indicated live cell population (**l**) 24 h following administration of nAlb–Cy5 alone ($n = 7$) or nAlb–Cy5 co-administered with nAlb–diABZI ($n = 8$); MFI for each cell population is shown in Supplementary Fig. 13. Inset of **k**: percentage of indicated cell population in the tumour as measured by flow cytometry. DC, dendritic cell; MΦ, macrophage; NK cell, natural killer cell. $P$ values determined by two-way ANOVA with Šídák's test for multiple comparisons. Replicates are biological, and data are shown as mean ± s.e.m.

is a promising drug carrier based on its long circulation half-life and proclivity to accumulate at tumour sites via both passive and active transport mechanisms[27–29]. Albumin and albumin-binding chaperones have been widely used to improve the delivery of chemotherapeutics, exemplified by albumin-bound paclitaxel (Abraxane)[30], as well as protein[31], peptide[32] and nucleic acid therapeutics[30]. Inspired by this previous work that motivates the unexplored potential of albumin as a carrier for STING agonists, we engineered a high-affinity anti-albumin nanobody (that is, single-domain antibody) for site-selective enzymatic bioconjugation of STING agonists via biorthogonal chemistry. Using a conjugatable diamidobenzimidazole (diABZI) STING agonist as a clinically relevant example, we demonstrate that nanobody hitchhiking of STING agonists on serum albumin dramatically improves their pharmacological properties and increases tumour accumulation, leading to a reduction in tumour burden and improved therapeutic outcomes in multiple mouse tumour models. We further demonstrate the programmability of the technology for integrating tumour targeting and additional immunoregulatory functions through the development of a bispecific nanobody–diABZI conjugate that binds to both albumin and the immune checkpoint ligand PD-L1. We demonstrate that this bivalent

nanobody carrier for STING agonist delivery further increases tumour accumulation while also inhibiting immunosuppressive PD-1/PD-L1 interactions, resulting in a reprogramming of the TME to a more immunogenic 'hot' milieu and a priming of antitumour T cells that further potentiate responses to multiple immunotherapeutic modalities. Collectively, our study positions albumin-hitchhiking nanobody–STING agonist conjugates as an enabling, multimodal and programmable strategy for cancer immunotherapy with high translational potential.

## Results

### Synthesis of albumin-hitchhiking nanobody–STING agonist conjugates

We hypothesized that conjugation of a STING agonist to an albumin-binding chaperone would extend blood circulation half-life and increase accumulation in cancerous tissue, enriching the production of cytokines and chemokines that facilitate the recruitment, proliferation and activation of leukocytes to the TME, which promotes cancer cell death (Fig. 1a). While several promising albumin-binding molecules have been described, including small molecules, fatty acids, peptides and *Streptococcus* protein G-derived domains[27,29,33], we elected to base our design on a nanobody with high affinity for albumin because nanobodies are modular and programmable via genetic engineering, are molecularly well defined, are amenable to scalable industrial manufacturing and are components of approved and clinically advanced therapeutics, including ozoralizumab, which contains an anti-albumin nanobody domain[31]. In addition, we sought to avoid the potential risk of accelerated albumin clearance that can occur due to direct covalent drug conjugation strategies[27,29] and to minimize the liver accumulation associated with the use of lipid-based albumin binders[30], a challenge also faced by many promising nanoparticle-based STING agonists[19,21,34,35]. We therefore recombinantly expressed a previously described nanobody domain—termed nAlb—that binds with nanomolar affinity to serum albumin (Fig. 1b)[36]. We modelled the binding of the nanobody domain to human serum albumin (HSA) using RoseTTAFold to generate the nAlb nanobody and RosettaDock to predict the binding site of the nanobody to the serum protein albumin. We found that the nAlb nanobody reached an optimal energy conformation through binding at domain IIB of HSA, indicating that nAlb does not compete with albumin binding to neonatal fragment crystallizable receptor, which facilities its long serum half-life (PDB, 4N0F). The binding affinity of nAlb was verified using isothermal calorimetry (ITC) both at physiological pH (7.5) and at endosomal pH (5.5), where nAlb maintained nanomolar affinity to both HSA and recombinant mouse serum albumin (Fig. 1c and Supplementary Fig. 1).

To enable site-selective ligation of STING agonists, we cloned the C-terminal of the nAlb nanobody to present a selective ligation tag (LPETGGHHHHHHEPEA) that acts as a substrate for an engineered pentamutant of sortase A designed to selectively ligate any primary amine-containing small molecule to the C-terminus of a protein[37], offering high programmability in the design. Using this approach, we ligated an amino-poly(ethylene glycol)$_3$-azide ($NH_2$-PEG$_3$-$N_3$) linker, which conferred a single azide functional handle on the nAlb nanobody and can be used to ligate cargo via strain-promoted azide-alkyne cycloaddition (Fig. 1d,f). While this strategy is amenable to ligation of diverse classes of STING agonists, we selected a diABZI compound as ongoing clinical trials are exploring similar agents as a systemically administered immunotherapy (for example, NCT03843359). To enable covalent conjugation to the nanobody, we synthesized a diABZI variant that was functionalized with an azide-reactive dibenzocyclooctyne (DBCO) group and a PEG$_{11}$ spacer (DBCO-PEG$_{11}$–diABZI) at the 7-position of one of the benzimidazole groups (Fig. 1e and Supplementary Figs. 2–4), a modification that is not predicted to interfere with diABZI binding to STING. We then used strain-promoted azide-alkyne cycloaddition to install a single DBCO-PEG$_{11}$–diABZI STING agonist or a DBCO-functionalized sulfo-Cy5 (referred to herein as Cy5) dye onto the

nanobody and verified 1:1 conjugation by electrospray ionization–mass spectrometry (ESI–MS) (Fig. 1f) and sodium dodecyl sulfate–polyacrylamide electrophoresis (SDS–PAGE) (Fig. 1g).

We evaluated the activity of the nAlb conjugated STING agonist (nAlb–diABZI) as well as the parent DBCO-PEG$_{11}$–diABZI compound and a previously optimized diABZI[23] (compound 3; referred to henceforth as diABZI) in two human reporter cell lines for type I interferon production: monocytes (THP1-Dual) and lung carcinoma cells (A549-Dual) (Fig. 1h,i). We found that the DBCO-PEG$_{11}$–diABZI variant retained a near-identical EC$_{50}$ value to the original diABZI agonist from literature, while, as expected, the in vitro activity of the nAlb–diABZI conjugate was reduced but nonetheless maintained high sub-100 nM activity for type I interferon production. Furthermore, we tested the activity of the nAlb–diABZI conjugate in mouse bone-marrow-derived macrophages (BMDMs), demonstrating that nAlb–diABZI stimulated the expression of the STING-driven cytokines *Ifnb1*, *Tnf* and *Cxcl10* after 4 h (Fig. 1j).

### Albumin-hitchhiking nanobodies show tumour tropism and enrich cargo delivery

Albumin has been reported to enter cancer cells and tumour-associated myeloid cells (for example, macrophages) through both albumin-dependent, receptor-mediated pathways and by micropinocytosis[27,29]. Although mechanisms of cellular albumin internalization may vary between tumour and cell types, we sought to gain insight into how nAlb–diABZI enters cells and activates STING. We first validated that intracellular uptake of nAlb–Cy5 was abrogated at 4 °C indicating an active endocytotic mechanism (Fig. 2a and Extended Data Fig. 1); by contrast, diABZI can enter cells by passive transport across the plasma membrane[23]. We next assessed whether albumin binding enhanced nAlb internalization by EMT6 breast cancer and myeloid cells. To test this, we first used flow cytometry to compare the cellular uptake of nAlb–Cy5 to a negative control nanobody targeting green fluorescent protein (GFP), nGFP–Cy5 (Supplementary Fig. 5), in serum-containing media, finding minor differences in cellular uptake between nAlb–Cy5 and nGFP–Cy5 (Extended Data Fig. 1). While eliminating serum from culture media decreased nAlb–Cy5 uptake, this occurred to the same extent for nGFP–Cy5, again indicating that cellular uptake occurs predominantly in an albumin receptor-independent manner in these cell types. Albumin can also be internalized by cancer and tumour-associated immune cell populations through non-receptor-mediated micropinocytosis. To evaluate this, we inhibited micropinocytosis in EMT6 cells, RAW264.7 macrophages and BMDMs using 5-(N-ethyl-N-isopropyl) amiloride (EIPA), which significantly reduced nAlb–Cy5 uptake (Fig. 2b). Given that macropinosomes often traffic to lysosomes, we next assessed colocalization of nAlb–Cy5 with lysotracker and found that a substantial and similar fraction (>50%) of nAlb–Cy5 and nGFP–Cy5 was colocalized with lysosomes or late endosomes in EMT6 and RAW264.7 cells (Fig. 2c,d). As expected, nAlb–diABZI did not mediate endosomal disruption as assessed using a previously described galectin 9 (Gal9) endosomal recruitment assay (Extended Data Fig. 1)[38].

To gain insight into how amide-linked diABZI is released from the nanobody upon cellular internalization, we incubated nAlb–diABZI with lysosomes isolated from rat liver (tritosomes), which are used to investigate stability and catabolism of molecules trafficked to an endosome–lysosome pathway, and used matrix-assisted laser desorption/ionization (MALDI) mass spectroscopy to assess the emergence of a PEGylated diABZI adduct that would be predicted due to amide bond cleavage by lysosomal proteases (Supplementary Fig. 6). We observed the presence of this peak as early as 1 h following incubation with tritosomes, suggesting that a fraction of nAlb–diABZI is lysosomally degraded to release a PEGylated diABZI variant. We synthesized this compound (Supplementary Figs. 7–9) and evaluated in vitro activity in THP1-Dual type I interferon reporter cells, finding that it had a similar

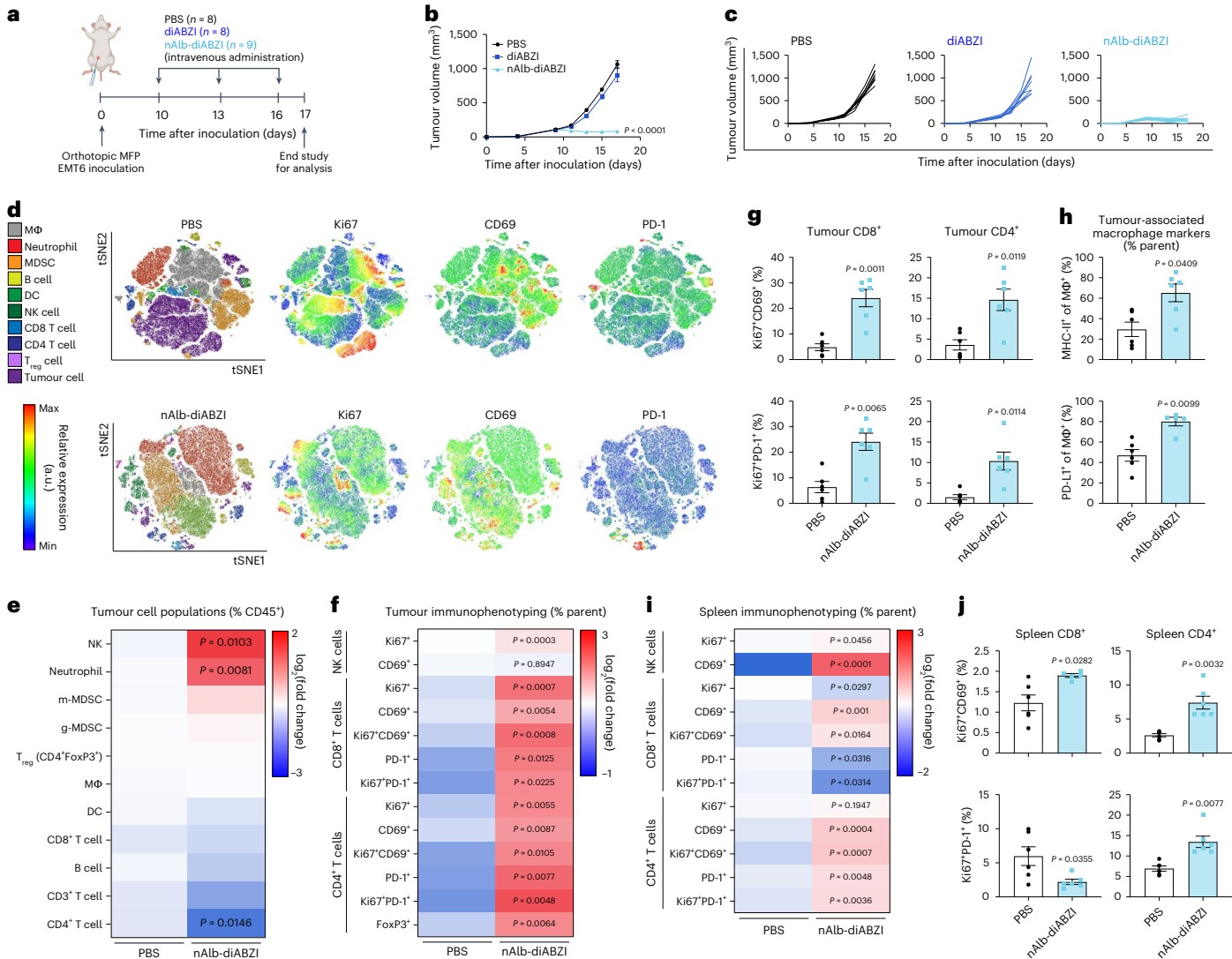

**Fig. 3 | Albumin-hitchhiking STING agonist inhibits breast tumour growth by shifting the immune cell profile of the TME. a**, Schematic of EMT6 tumour inoculation, treatment schedule and study end point for gene expression and flow cytometry analysis. **b,c**, Tumour growth curves (**b**) and spider plots of individual tumour growth curves (**c**) for each mouse with EMT6 tumours treated with nAlb–diBZI (n = 9), diABZI (n = 8) or PBS (n = 8). P value determined by two-way ANOVA with post hoc Tukey's correction for multiple comparisons with comparison to PBS on day 17 shown. **d**–**j**, Flow cytometric analysis of breast tumours and spleen 24 h following final dose of nAlb–diABZI or PBS (n = 6). **d**, t-Distributed stochastic neighbour embedding (tSNE) plots of live cells in EMT6 tumours coloured by cell population with relative expression level of Ki67, CD69 and PD-1 as indicated on heat map. **e,f**, Heat maps summarizing the fold

change in the percentage of indicated cell population (**e**) and fold change in the frequency of NK cells, CD8+ T cells and CD4+ T cells expressing the indicated marker or marker combination in EMT6 breast tumours (**f**). **g**, Quantification of Ki67+CD69+ and Ki67+PD1+ CD8+ and CD4+ T cells in EMT6 tumours following treatment with nAlb–diABZI or PBS. **h**, Quantification of frequency of major histocompatibility complex-II (MHC-II)+ and PD-L1+ macrophages in EMT-6 tumours following treatment with nAlb–diABZI or PBS. **i**, Heat map summarizing fold change in the frequency of NK cells, CD8+ T cells and CD4+ T cells expressing activation markers within splenic populations. **j**, Quantification of Ki67+CD69+ and Ki67+PD1+ CD8+ and CD4+ T cells in spleens. P values determined by two-tailed Student's t-test. Replicates are biological, and data are shown as mean ± s.e.m. Panel **a** created with BioRender.com.

EC$_{50}$ value to the previously described diABZI molecule, which can enter cells through passive transport[23] (Supplementary Fig. 10).

We next evaluated the pharmacokinetics and biodistribution of nAlb site-selectively conjugated to Cy5 as described for diABZI above (nAlb–Cy5) compared to an analogous control anti-epidermal growth factor receptor (EGFR) nanobody (nEGFR) that we cloned and Cy5 labelled using the same strategy (Supplementary Fig. 5). To assess the pharmacokinetic profile achieved by using anti-albumin nanobody hitchhiking, we intravenously administered free DBCO–Cy5, nEGFR–Cy5 and nAlb–Cy5 in healthy female C57BL/6 mice and collected blood at discrete time points over several days (Fig. 2e). By measuring the concentration of Cy5 in the serum using fluorescence spectroscopy, we determined the elimination half-life of both the free

dye and the nEGFR–Cy5 conjugate to be approximately 5 min, matching the expected half-life of a typical nanobody that is rapidly cleared via renal excretion due to its small size (~15 kDa)[39]. However, the nAlb–Cy5 conjugate had an elimination half-life of approximately 55 h, consistent with in situ binding to and hitchhiking on serum albumin, which has a half-life of ~35–40 h in mice[40]. By comparison, the reported half-life of diABZI is ~90 min[23], while that of CDNs is typically <5 min[34]. We next tracked the biodistribution of DBCO–Cy5, nEGFR–Cy5 and nAlb–Cy5 in female Balb/c mice with orthotopic EMT6 (EGFR+) breast tumours inoculated in the mammary fat pad (MFP). At 24 h after administration, mice were euthanized, and major organs and tumours were imaged with an in vivo imaging system (IVIS) instrument to evaluate Cy5 biodistribution (Fig. 2f,g), and tissue was homogenized for quantification of Cy5

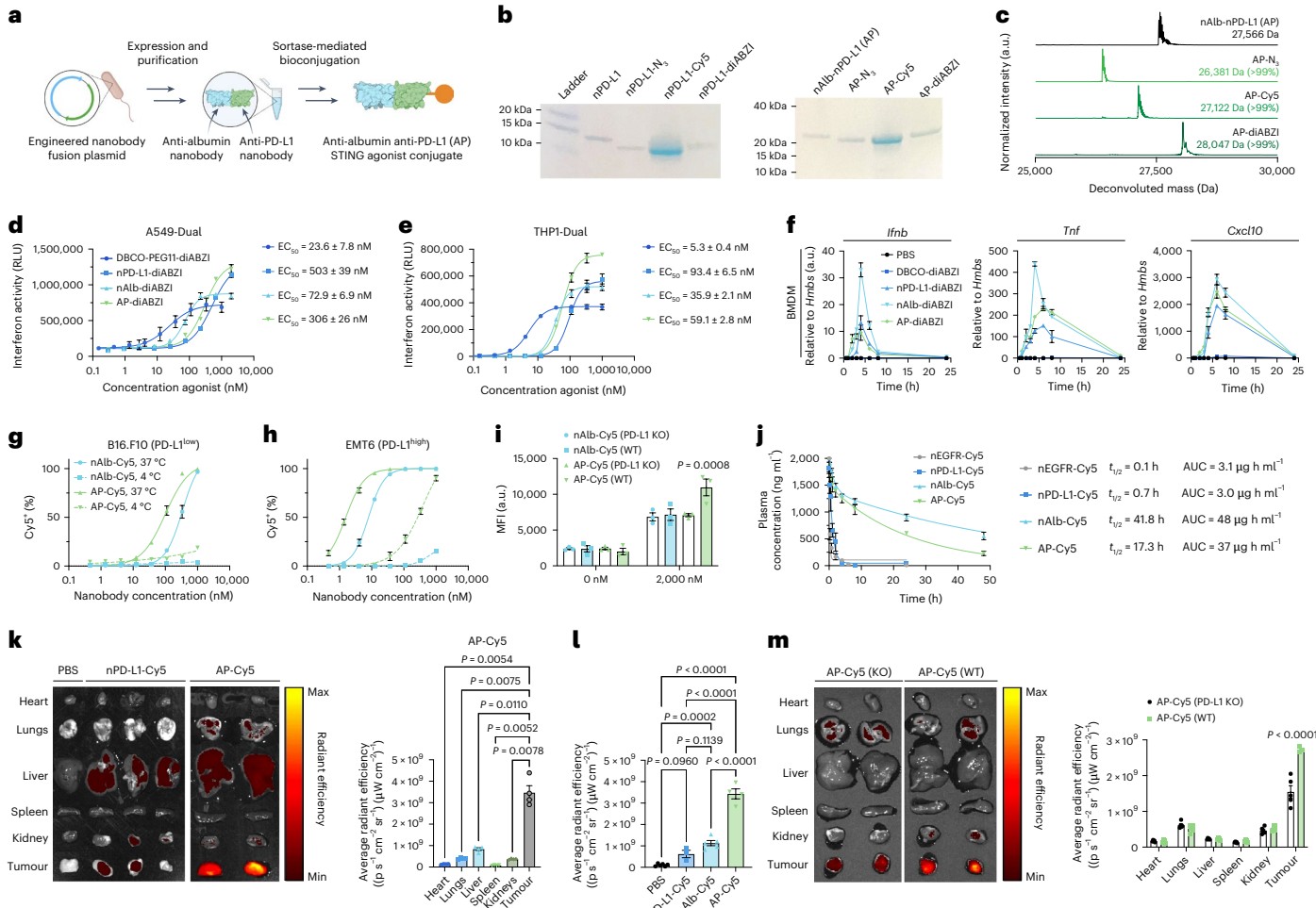

**Fig. 4 | Design, synthesis and testing of bivalent nanobody–STING agonist conjugate for albumin hitchhiking and targeting of PD-L1. a**, Scheme for the cloning, expression and bioconjugation of small molecule cargo to generate the AP–diABZI conjugate. **b,c**, SDS–PAGE (**b**) and ESI–MS (**c**) confirming the purity and molecular weight of AP conjugates (see Source Data for uncropped gels in ref. 90). **d,e**, Dose–response curves for indicated nanobody–diABZI conjugate in A549-Dual ($n = 3$) (**d**) and THP1-Dual type I interferon reporter cell lines ($n = 3$) (**e**) with estimated $EC_{50}$ values indicated in the legends. **f**, qPCR analysis of genes associated with STING activation in BMDMs in response to treatment at discrete time points with indicated agonist at 0.25 μM ($n = 3$). **g,h**, Dose–response curve for nAlb–Cy5 and AP–Cy5 conjugate intracellular uptake and surface binding at 37 °C and 4 °C as measured by flow cytometry in B16.F10 cells ($n = 2$ at 4 °C and $n = 3$ at 37 °C) (**g**) and EMT6 cells ($n = 3$) (**h**). **i**, MFI for nAlb–Cy5 and AP–Cy5 conjugate surface binding at 2 μM compared to PBS (0 μM) for EMT6 WT and EMT6 PD-L1 KO cell lines at 37 °C ($n = 3$). KO, knock-out; WT, wild type. **j**, Pharmacokinetics of indicated nanobody–Cy5 conjugate in healthy Balb/c female mice ($n = 4$ for nPD-L1–Cy5; $n = 5$ for all other groups). Elimination phase half-life and AUC are indicated in the legend. **k**, Representative IVIS fluorescence

images of excised tumours and major organs (left) and quantification of average radiant efficiencies (right) of tumours and major organs 48 h after administration of nPD-L1–Cy5 and AP–Cy5 in mice with EMT6 breast tumours ($n = 4$). $P$ values determined by repeated measures ANOVA with Dunnett's multiple comparison test for tumour compared to indicated tissue. **l**, Comparison of Cy5 radiant efficiencies in tumour tissue 48 h following administration of indicated nanobody–Cy5 conjugate ($n = 6$ for PBS and nAlb–Cy5; $n = 4$ for AP–Cy5; $n = 3$ for nPD-L1–Cy5). $P$ values determined by one-way ANOVA with post hoc Tukey's correction for multiple comparisons with comparisons between all groups and PBS and between nAlb–Cy5 and AP–Cy5 as indicated. **m**, Representative IVIS fluorescence images of excised tumours and major organs (left) and quantification of average radiant efficiencies (right) of tumours and major organs 48 h after administration of AP–Cy5 in mice with wild-type EMT6 (WT) and PD-L1 knock-out EMT6 (PD-L1 KO) breast tumours ($n = 5$). $P$ values determined by repeated measures ANOVA with Dunnett's multiple comparison test for WT versus PD-L1 KO groups. Replicates are biological, and data are shown as mean ± s.e.m. Panel **a** created with BioRender.com.

using fluorescence spectroscopy (Fig. 2h). We observed minimal Cy5 fluorescence in major organs for both nEGFR–Cy5 and nAlb–Cy5 conjugates but substantial tumour accumulation of only the nAlb–Cy5 conjugate, corresponding to ~11% injected dose per gram tissue (Fig. 2h), significantly higher than other organs; similar findings were observed in a B16.F10 tumour model (Fig. 2i). Immunofluorescence staining of excised and cryosectioned tumours (Fig. 2j and Supplementary Fig. 11) further confirmed nAlb–Cy5 accumulation at the tumour site, with the highest Cy5 fluorescence observed proximal to CD31+ tumour vasculature and with Cy5 signal also observed within the tumour stroma (for example, colocalizing with CD45+ immune cells). Albumin binding

to secreted protein acidic and rich in cysteine (SPARC) expressed in tumour tissue has also been implicated in increased accumulation of albumin-binding therapeutics[27], and we found that SPARC is expressed in both EMT6 and B16.F10 tumours (Supplementary Fig. 12) and may therefore contribute to nAlb accumulation.

Based on the preferential tumour accumulation of nAlb–Cy5, we next used flow cytometry to determine which tumour-associated cell populations internalized the conjugate (Fig. 2k,l and Supplementary Fig. 13). At 24 h after intravenous injection of nAlb–Cy5, we found that ~8% of all live cells in the tumour were Cy5+ (Supplementary Fig. 14), and, among Cy5+ cells, the majority were CD11b+F4/80+ tumour-associated

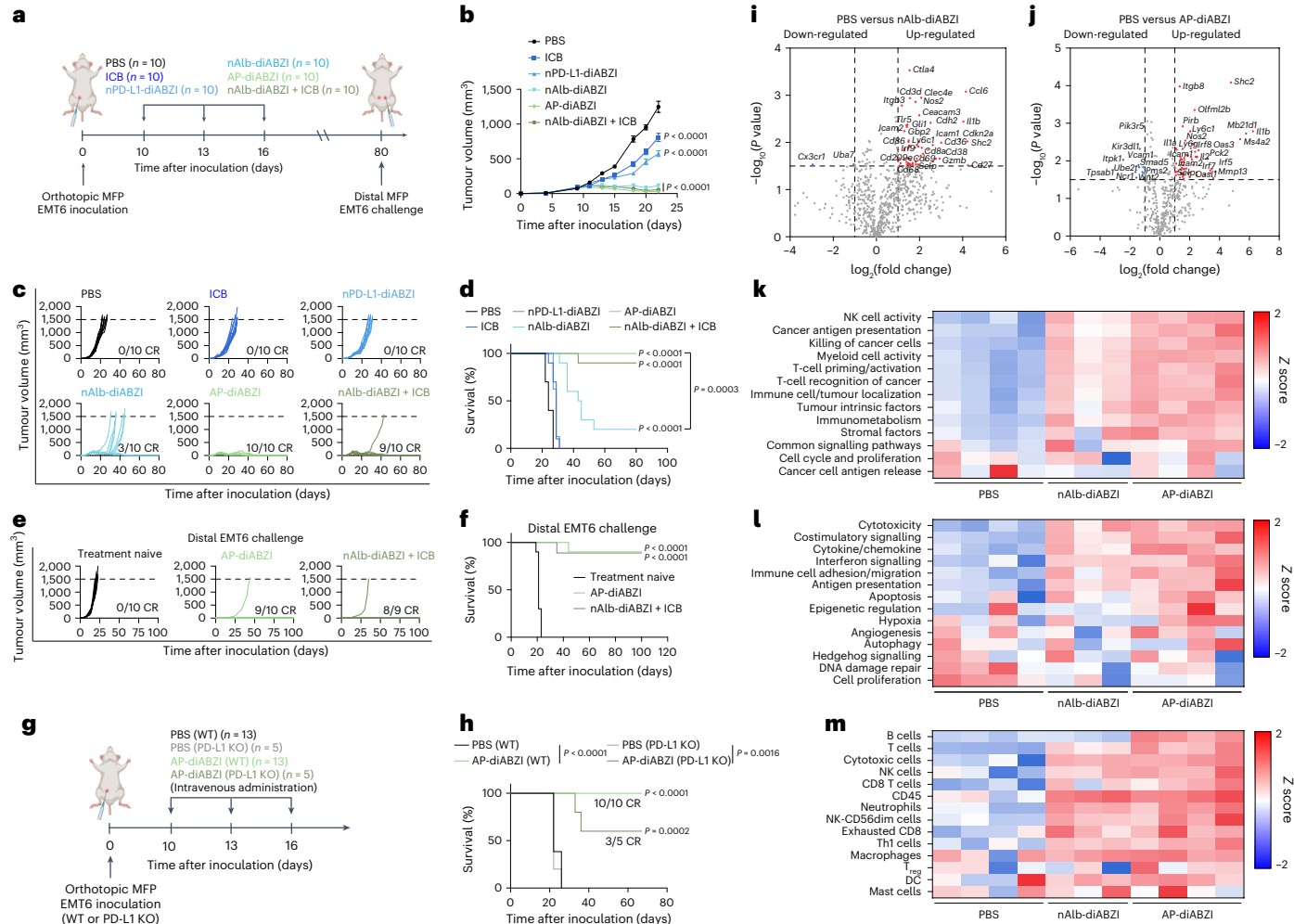

**Fig. 5 | Systemic administration of AP–diABZI conjugates enhance antitumour immune and therapeutic responses in EMT6 breast cancer model.**
**a**, Schematic of EMT6 tumour inoculation and treatment schedule; nanobody–diABZI conjugates and PBS (vehicle) were administered intravenously, and ICB (anti-PD-L1 IgG) was injected intraperitoneally. **b–d**, Tumour growth curves (**b**), spider plots of individual tumour growth curves (**c**) and Kaplan–Meier survival plots (**d**) for mice with EMT6 tumours treated as indicated ($n = 10$). CR, complete responder. $P$ values in **b** determined by one-way ANOVA with Dunnett's multiple comparison test for each group compared to PBS on day 22. In **d**, end-point criteria of 1,500 mm³ tumour volume with $P$ value determined by log-rank test compared to PBS group or between nAlb–diABZI and AP–diABZI as indicated. **e,f**, Spider plots of individual tumour growth curves (**e**) and Kaplan–Meier survival curves (**f**) of mice challenged or re-challenged (for complete responders to the treatment regimen) with EMT6 cells ($n = 10$ for treatment-naive and re-challenge of mice treated with AP–diABZI; $n = 9$ for re-challenge of mice treated with

nAlb–diABZI + ICB); end-point criteria of 1,500 mm³ tumour volume with $P$ value determined by log-rank test compared to treatment-naive group. **g**, Scheme of EMT6 WT and EMT6 PD-L1 KO tumour inoculation and treatment schedule. **h**, Kaplan–Meier survival plots for mice with EMT6 WT ($n = 13$) or PD-L1 KO ($n = 5$) tumours treated with AP–diABZI or PBS; end-point criteria of 1,500 mm³ tumour volume with $P$ value determined by log-rank test compared to PBS (WT) group or between WT and PD-L1 KO groups as indicated in the legend. **i,j**, Volcano plots representing $-\log_{10}$(significance) and $\log_2$(fold change) for gene expression analysis in nAlb–diABZI versus PBS ($n = 4$) (**i**) and AP–diABZI versus PBS ($n = 4$) (**j**). **k–m**, Heat maps of NanoString gene cluster matrices showing $Z$ score fold changes for functional gene annotations (**k**), biological signatures (**l**) and cell types ($n = 4$ for PBS and AP–diABZI; $n = 3$ for nAlb–diABZI) (**m**). Replicates are biological, and data are shown as mean ± s.e.m. Panels **a** and **g** created with BioRender.com.

---

macrophages or CD45⁻CD31⁻ cells, which are primarily cancer cells (Fig. 2k). Cancer cells (CD45⁻CD31⁻) and macrophages are the most prevalent cell populations in the EMT6 tumour model and have been reported to endocytose albumin in tumours[41,42]. Evaluating nAlb–Cy5 uptake within specific cell populations, we found that ~5–10% of cancer cells (CD31⁻CD45⁻), macrophages (CD11b⁺F4/80⁺) and dendritic cells (CD11c⁺) were Cy5⁺ with a higher (~15–20%) frequency of Cy5⁺ CD45⁻CD31⁺ endothelial cells and neutrophils (Fig. 2l). As assessed by Cy5 median fluorescence intensity (MFI), the cell populations with the highest degree of nAlb–Cy5 uptake were CD45⁻CD31⁺ endothelial cells, neutrophils, dendritic cells, macrophages and cancer (CD45⁻CD31⁻) cells (Supplementary Fig. 13). To determine whether this cellular uptake profile was influenced by STING activation, we concurrently

administered nAlb–Cy5 with nAlb–diABZI and found that the addition of nAlb–diABZI primarily impacted the myeloid cell composition of the tumour at 24 h, resulting in an increased frequency of neutrophils and myeloid-derived suppressor cells (MDSCs) and a reduction in macrophages (Fig. 2k, inset) while slightly biasing nAlb–Cy5 uptake towards macrophages, dendritic cells and neutrophils. We also evaluated cellular uptake of nAlb–Cy5 in the spleen (Supplementary Fig. 13), which, while not a major organ of distribution for nAlb–Cy5, is a potentially important secondary lymphoid organ for generating systemic antitumour immunity, finding that ~5–10% of macrophages and dendritic cells were Cy5⁺. Taken together, these data show that nanobody albumin hitchhiking can increase tumour accumulation to allow for endocytosis of cargo by multiple tumour-associated cell

types. While in vivo mechanisms of albumin transport and cellular uptake are complex and still not fully understood, taken together our data suggest that nAlb preferentially accumulates at tumour sites and is macropinocytosed, primarily by cancer cells and tumour-associated myeloid cells, resulting in lysosomal degradation and release of a diABZI variant that activates STING.

## nAlb–diABZI potently stimulates STING activation in the TME to inhibit tumour growth

Based on the ability of nAlb to enhance cargo distribution to tumour sites, we next performed a dose–response study to evaluate the therapeutic efficacy of nAlb–diABZI conjugates in an established poorly immunogenic (that is, immunologically 'cold') B16.F10 tumour model that is resistant to immune checkpoint blockade (ICB) (Supplementary Fig. 15)[43]. Using a treatment regimen that we and others have used for evaluation of STING agonists[34,44], we intravenously administered nAlb–diABZI to mice bearing ~75 mm³ B16.F10 tumours at doses ranging from 5 to 0.05 μg diABZI content, finding that all doses significantly inhibited tumour growth and extended survival time. It is worth noting that the 5 μg dose significantly enhanced efficacy relative to a 3× higher dose of diABZI, showing the enhancement in potency enabled through albumin hitchhiking. While the 5 μg dose resulted in ~10–12% weight loss, this was transient and occurred only after the first injection (Supplementary Fig. 15a). Nonetheless, towards maximizing the safety profile of the treatment, we selected a dose of 1.25 μg, confirmed antitumour efficacy of both a single and three-dose regimen in the B16.F10 model (Supplementary Fig. 15d–g and Supplementary Fig. 16) and performed a preclinical analysis of nAlb–diABZI toxicity (Extended Data Fig. 2). Healthy mice were administered vehicle (PBS) or nAlb–diABZI (1.25 μg diABZI) intravenously three times spaced 3 days apart; weight loss was monitored daily, and blood was collected 4 and 24 h after the first injection for analysis of serum cytokines. In response to nAlb–diABZI, mice experienced only a mild (~5%) and transient weight loss similar to that described for nanoparticle-based delivery of STING agonists[18,19,21] with elevated plasma levels of STING-driven cytokines with antitumour functions (for example, type I interferon, IL-12) 4 h following injection, which returned to near baseline by 24 h (Extended Data Fig. 2c). Mice were euthanized a week following the last injection, blood was collected for biochemistry analysis (Extended Data Fig. 2d), and major organs were isolated for histological evaluation (Extended Data Fig. 2e) by a board-certified veterinary pathologist, who observed no clinically notable changes between the untreated control mice and nAlb–diABZI treated mice, consistent with minor changes in blood biochemistry and cellular composition. Based on this favourable safety profile at a therapeutically effective dose in a challenging B16.F10 tumour model, we selected a dose of 1.25 μg for all subsequent studies.

Given the substantial tumour accumulation of nAlb observed in orthotopic EMT6 breast tumours—and considering that only approximately 20% of breast cancer patients benefit from PD-1/PD-L1 ICB[45]—we next evaluated the capacity of nAlb–diABZI to create a TME that inhibited tumour growth. Female Balb/c mice were inoculated with EMT6 cells in a MFP and treated with nAlb–diABZI, free diABZI or vehicle (PBS) at a tumour volume of ~75 mm³ (Fig. 3a). Treatment with nAlb–diABZI strongly suppressed tumour growth, whereas the free diABZI STING agonist did not confer a therapeutic benefit (Fig. 3b,c). Consistent with accumulation of nAlb at tumour sites, we found a notable increase in the expression of genes associated with STING pathway activation, including *Ifnb1*, *Cxcl10*, *Cxcl9* and *Tnf* (Supplementary Fig. 17).

To gain insight into the immunological mechanisms by which nAlb–diABZI inhibited tumour growth, we used multispectral flow cytometric immunophenotyping to quantify changes in key myeloid and lymphocyte populations and their phenotypes (Fig. 3d–j and Extended Data Fig. 3) in EMT6 tumours and in the spleen 24 h following the third nAlb–diABZI administration. We found that administration

of nAlb–diABZI increased the infiltration of CD8⁺ T cells with considerably elevated markers of activation (CD69) and proliferation (Ki67), as well as the frequency of Ki67⁺PD-1⁺ CD8⁺ T cells, which have been correlated with favourable responses to immunotherapy in patients[46]. While there was a reduction in the overall frequency of CD4⁺ T cells, this was associated with an increased frequency of CD69⁺Ki67⁺ and Ki67⁺PD-1⁺ CD4⁺ T cells. There was also a significant increase in the frequency of natural killer (NK) cells and Ki67⁺ NK cells in the TME; it is worth noting that the levels of splenic CD69⁺ and Ki67⁺ NK cells were also elevated, potentially suggesting mobilization of NK cells from the spleen to the tumour (Supplementary Fig. 18)[47]. Trends towards increased frequency of MDSCs (Fig. 3d,e), a significant increase in the frequency of FoxP3⁺CD4⁺ regulatory T cells (Fig. 3e,f) and elevated MHC-II and PD-L1 on macrophages (Fig. 3g,h) were also observed. Similar effects have been described for other STING agonists, which may act as counterregulatory mechanisms that contribute to resistance to nAlb–diABZI as a monotherapy. In particular, MDSCs have been reported to reduce the efficacy of some STING-activating therapies[48–50], and we therefore evaluated nAlb–diABZI in combination with orally administered SX-682, which inhibits CXCR1/2 chemokine receptors involved in MDSC recruitment[51] but, surprisingly, found that SX-682 tended to reduce nAlb–diABZI efficacy (Supplementary Fig. 19). We also used anti-Gr1 antibodies to deplete MDSCs (primarily gMDSCs)[52] and again found a modest reduction in nAlb–diABZI efficacy (Supplementary Fig. 20). Similar findings have been reported by others[53], reflecting the potentially complex roles of MDSCs in response to immunotherapy given their capacity to differentiate into mature antitumour effectors. Nonetheless, our data suggest that MDSCs do not strongly restrict the efficacy of nAlb–diABZI, at least in the EMT6 breast tumour model.

In addition to immunological resistance mechanisms, the efficacy of nAlb–diABZI may also be inhibited through generation of anti-nAlb or anti-diABZI antibodies that may lead to accelerated blood clearance[54]. Although albumin has been described to generate immune tolerance to antigenic cargo[55] and nanobodies typically have low immunogenicity[56], we nonetheless addressed this possibility by intravenously administering healthy wild-type C57BL/6 mice with nAlb–diABZI on days 0, 3 and 6 and evaluated anti-variable heavy domain of heavy chain (VHH) antibody titre in serum on day 14 and also compared the plasma half-life of nAlb–Cy5 to untreated mice. We did not detect an anti-VHH antibody response in serum (Supplementary Fig. 21) and observed a similar nAlb–Cy5 half-life between untreated mice (~59 h) and nAlb–diABZI-treated mice (~64 h) (Supplementary Fig. 22), suggesting that antibody-mediated nanobody clearance was unlikely to reduce nAlb–diABZI efficacy in our studies; however, this possibility cannot be discounted in humans where dose and treatment regimen, among other variables, will be different and therefore will need to be further investigated.

## Engineering an albumin-binding, bivalent nanobody fusion for combined STING agonist delivery and immune checkpoint inhibition

Having demonstrated the potent antitumour effects of our albumin-hitchhiking STING agonist, we next sought to leverage the modularity of nanobody engineering to confer additional immunotherapeutic functionality and demonstrate the programmability of the system. As a translationally relevant example, we introduced a second previously described nanobody domain that binds to PD-L1[57]. Our rationale for selecting PD-L1 was twofold. First, we, and others, have demonstrated synergy between STING agonists and PD1/PD-L1 ICB in suppressing tumour growth, including evidence that STING activation can directly upregulate PD-L1 expression[44,58]. Second, PD-L1 can be expressed by both cancer cells and immunosuppressive myeloid cells in solid tumours[59], providing a molecular target for increasing tumour accumulation; indeed, anti-PD-L1 nanobodies have been used previously in imaging applications with high selectivity for tumour

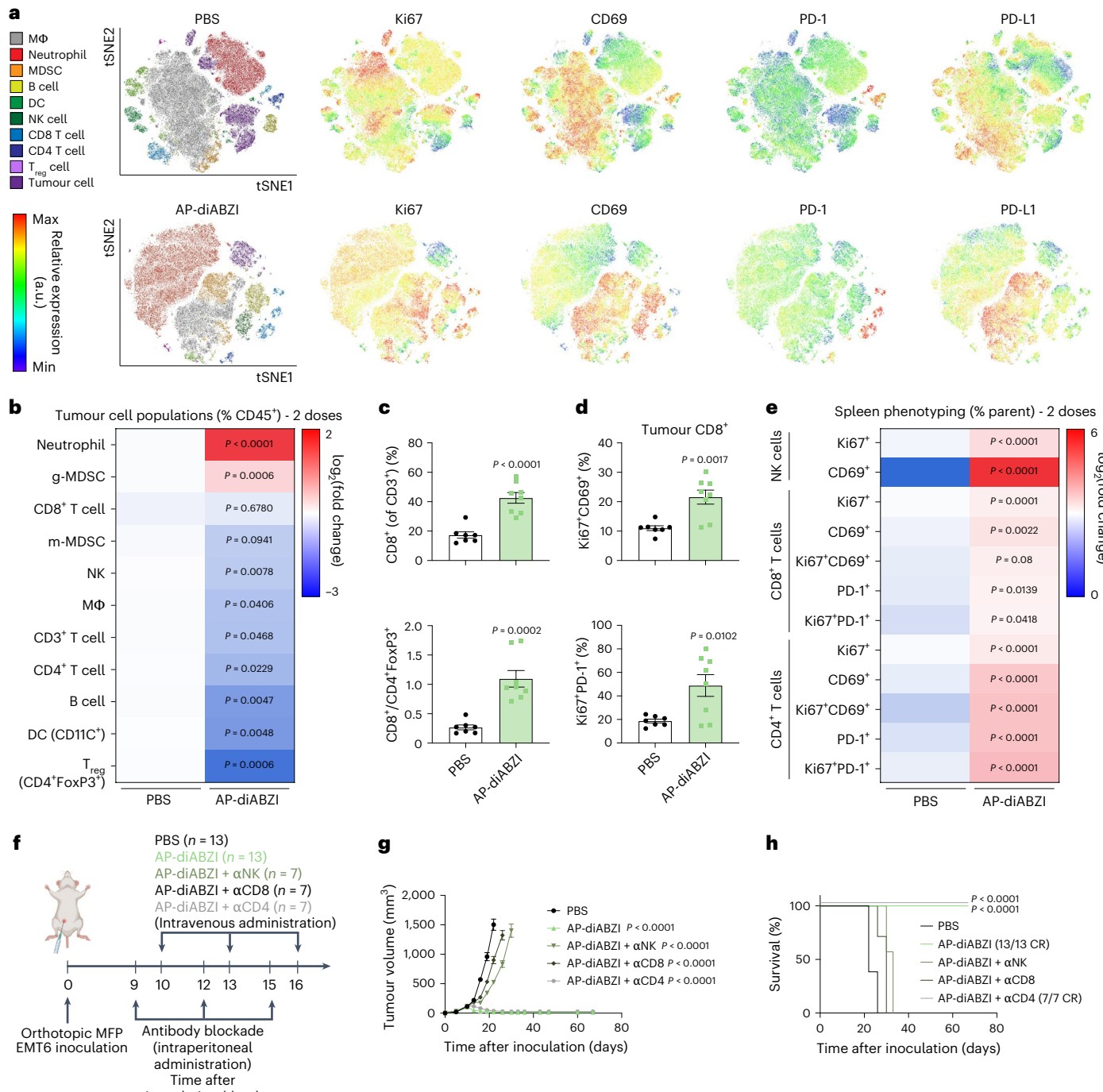

**Fig. 6 | AP−diABZI activates a tumoricidal NK and T-cell response.** Flow cytometric analysis of orthotopic EMT6 breast tumours 24 h following two intravenous doses of AP−diABZI (n = 8) or PBS (n = 7). **a**, tSNE plots of live cells in EMT6 tumours coloured by cell population with relative expression level of Ki67, CD69, PD-1 and PD-L1 as indicated on heat map. **b**, Heat map summarizing the fold change in the percentage of indicated cell populations in EMT6 tumours. **c**, Bar plots showing an increase in CD8+ cells and the ratio of CD8+ to CD4+FoxP3+ cells (% of CD3+ tumour cells). **d**, Quantification of Ki67+CD69+ and Ki67+PD1+ CD8+ T cells in EMT6 tumours. **e**, Spleen phenotyping heat map of frequency of NK cells, CD8+ T cells and CD4+ T cells (n = 7). In **b**−**e**, P values determined by two-tailed Student's t-test. **f**, Schematic of EMT6 tumour inoculation and treatment schedule with depletion antibodies anti-Asialo GM1 (αNK) IgG, anti-CD8 IgG and anti-CD4 IgG (n = 13 for PBS and AP−diABZI and n = 7 for AP−diABZI combined with anti-Asialo GM1, anti-CD8 or anti-CD4 IgG). **g**,**h**, Tumour growth curves (**g**) and Kaplan–Meier survival plots (**h**) for mice with EMT6 tumours treated as indicated. In **g**, P values determined by two-way ANOVA with post hoc Tukey's correction for multiple comparisons for all groups compared to PBS on day 22. In **h**, end-point criteria of 1,500 mm³ tumour volume with P values determined by log-rank test compared to PBS. Replicates are biological, and data are shown as mean ± s.e.m. Panel **f** created with BioRender.com.

tissue[57]. We therefore hypothesized that an anti-albumin/anti-PD-L1 nanobody fusion would increase tumour targeting, while inhibiting immunoregulatory PD1/PD-L1 interactions that restrain responses to STING agonists. Thus, we generated a fusion protein that uses a genetic linker to connect both nanobody domains and maintained the C-terminal sortase ligation tag to generate an anti-albumin/anti-PD-L1 (AP)−STING agonist conjugate, termed AP−diABZI (Fig. 4a). We characterized the synthesis and generation of both anti-PD-L1 nanobody

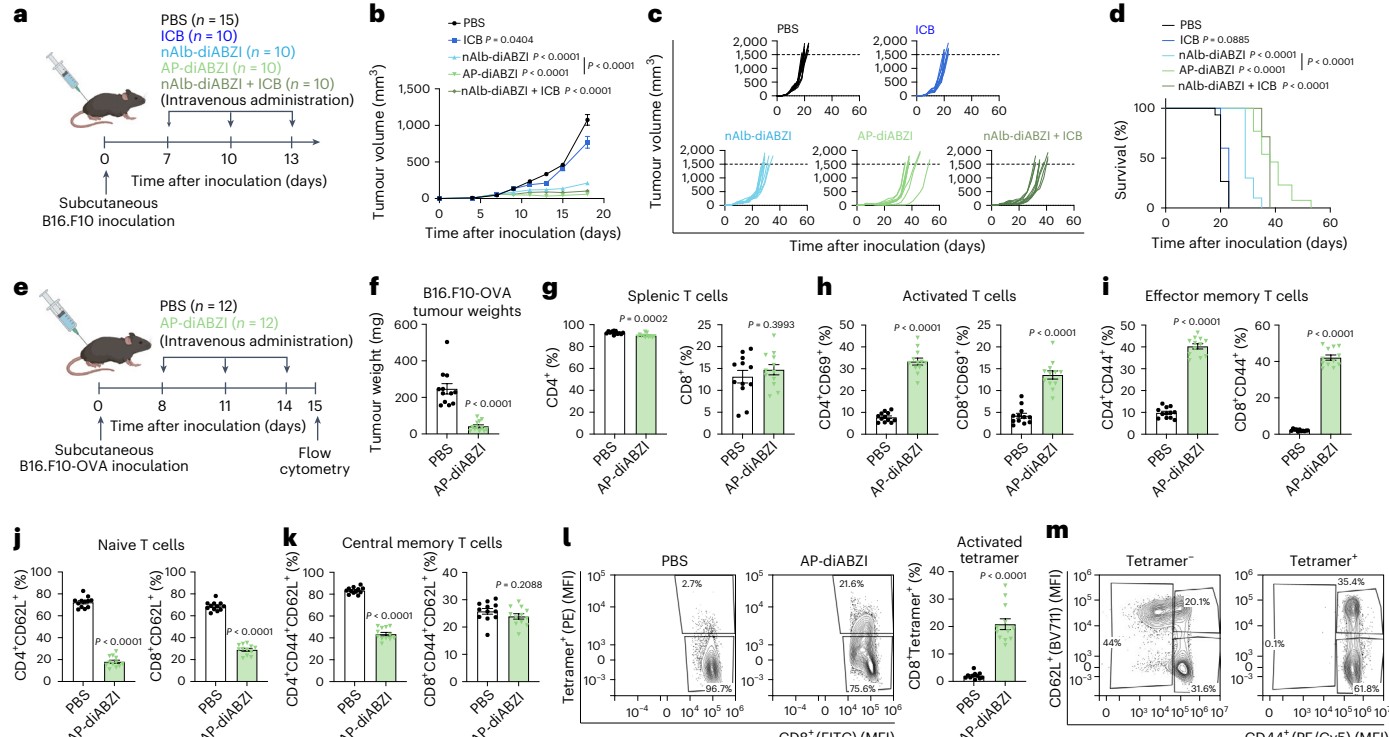

**Fig. 7 | Nanobody–STING agonist conjugates stimulate antitumour immunity in B16.F10 melanoma tumour model. a**, Schematic of B16.F10 tumour inoculation and treatment schedule; nanobody–diABZI conjugates and PBS (vehicle) were administered intravenously, and ICB (anti-PD-L1 IgG) was injected intraperitoneally. **b–d**, Tumour growth curves (**b**), spider plots of individual tumour growth curves (**c**) and Kaplan–Meier survival plots (**d**) (n = 15 for PBS; n = 10 for all other groups). In **b**, P values determined by two-way ANOVA with post hoc Tukey's correction for multiple comparisons for all groups compared to PBS on day 18. In **d**, end-point criteria of 1,500 mm³ tumour volume with P values determined by log-rank test compared to PBS control or between nAlb–diABZI and AP–diABZI as indicated. **e**, Schematic of B16.F10-OVA tumour inoculation, treatment schedule and study end point for flow cytometry analysis (n = 12). **f**, Tumour weight on day 15 for mice with B16.F10-OVA tumours treated with

AP–diABZI or PBS. **g**, Frequency of CD4+ and CD8+ T cells in the spleen at study end point. **h–k**, Flow cytometric analysis of the frequency of CD69+ CD8+ and CD4+ T cells (**h**), CD44+CD62L− effector memory T cells (**i**), CD44−CD62L+ naive T cells (**j**) and CD44+CD62L+ central memory T cells (**k**). **l**, Representative flow cytometry dot plots (left) and analysis of the frequency of SIINFEKL/H-2Kb tetramer+ ((PE) (MFI)) CD8+ T cells ((FITC) (MFI)) (right) in the spleen at study end point. **m**, Representative flow cytometry dot plots showing the distribution of CD8+ T_EM (CD44+CD62L−) and T_CM (CD44 + CD62L+) (CD44: (PE/Cy5) (MFI); CD62L: (BV711) (MFI)) within the OVA-specific (tetramer+) and non-OVA-specific (tetramer−) populations. P values determined by two-tailed Student's t-test. Replicates are biological, and data are shown as mean ± s.e.m. Panels **a** and **e** created with BioRender.com.

(nPD-L1) and AP conjugates to Cy5 and diABZI, showing that a single, homogeneous product that contained all three functional elements was formed (Fig. 4b,c and Supplementary Figs. 1 and 10). The in vitro activity of nPD-L1–diABZI and AP–diABZI was tested in A549-Dual and THP1-Dual type I interferon reporter cells (Fig. 4d,e) and by quantitative PCR (qPCR) for analysis of STING-associated cytokines/chemokine gene expression in primary BMDMs and bone-marrow-derived dendritic cells (BMDCs) (Fig. 4f and Supplementary Fig. 23). We found that all nanobody–diABZI conjugates were potently active in both reporter cell lines without evidence of cytotoxicity (Supplementary Fig. 24) and that nanobody–diABZI conjugates were more active than the parent DBCO–diABZI in BMDMs and triggered STING-associated gene expression with similar kinetics (Fig. 4f); both nAlb–diABZI and AP–diABZI were also active in mouse BMDCs (Supplementary Fig. 23). In addition, we showed using flow cytometry that the incorporation of the PD-L1 targeting domain enhanced binding and internalization in B16.F10 (PD-L1low) and EMT6 (PD-L1high) cells (Fig. 4g,h) relative to the albumin binding nanobody domain alone, which we further confirmed by comparing internalization by wild-type and PD-L1 knock-out EMT-6 cells (Fig. 4i).

We next tested the hypothesis that integrating a PD-L1 binding domain would increase tumour accumulation. We administered 2 mg kg⁻¹ of Cy5-conjugated nEGFR, nPD-L1, nAlb and AP nanobodies to healthy Balb/c mice intravenously and collected blood at discrete

time points to evaluate pharmacokinetics (Fig. 4j). We also administered Cy5-conjugated nanobodies to mice with orthotopic EMT6 breast tumours and euthanized mice at 48 h to quantify nanobody–Cy5 conjugate biodistribution to major organs and tumours using IVIS (Fig. 4k,l). While the AP–Cy5 conjugate had a shorter elimination half-life than nAlb–Cy5 (17 h to 55 h, respectively), likely due to binding of target PD-L1 in tissue and removal from circulation, both carriers maintained an increased elimination half-life and area under the curve (AUC) relative to either targeted nanobody (nEGFR and nPD-L1) alone, which were cleared rapidly from circulation (Fig. 4j). While AP is approximately twice the size (~28 kDa) of the anti-PD-L1 nanobody, both are below the threshold for renal clearance[39], and therefore, the increased circulation time of AP can be primarily attributed to the albumin-hitchhiking functionality. Furthermore, while the nPD-L1–Cy5 conjugate was observed at similarly low levels in major organs (liver and kidneys) and the tumour at 48 h (Fig. 4k), the AP–Cy5 conjugate showed significant tumour accumulation (corresponding to 2.19 ± 0.43% injected dose (ID) per g tumour) relative to major organs (Fig. 4k) and significant increase over nAlb alone (Fig. 4l). To further demonstrate increased tumour targeting, we compared the relative tumour accumulation of AP–Cy5 in breast tumours established using parental or PD-L1 knockout EMT-6 cells and found a significant decrease in tumour accumulation in the PD-L1 knockout model (Fig. 4m). It should be noted that PD-L1 was only knocked out

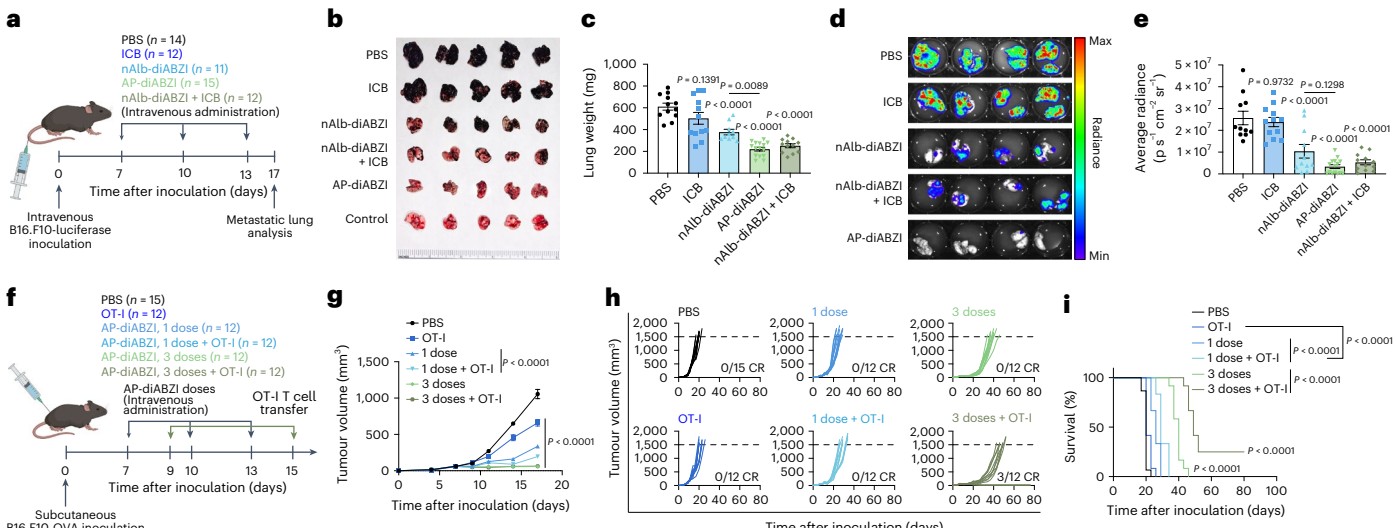

**Fig. 8 | Albumin-hitchhiking STING agonists improve immunotherapy responses in a model of lung metastatic melanoma and adoptive T-cell transfer therapy. a**, Schematic of B16.F10-Luc intravenous tumour inoculation, treatment schedule and study end point for analysis of lung tumour burden; nanobody–diABZI conjugates and PBS (vehicle) were administered intravenously, and ICB (anti-PD-L1 IgG) was injected intraperitoneally (*n* = 15 for AP–diABZI; *n* = 14 for PBS; *n* = 12 for ICB and nAlb–diABZI + ICB; *n* = 11 for nAlb–diABZI). **b,c**, Representative images of lungs (**b**) and lung weights (**c**) of mice treated as indicated. **d,e**, Representative IVIS luminescence images (**d**) and quantification of average radiance from luciferase expressing B16.F10 cells within isolated lung tissue (**e**). *P* values determined by one-way ANOVA with post hoc Tukey's correction for multiple comparisons for all groups versus PBS or nAlb–diABZI versus AP–diABZI as indicated. **f–i**, Evaluation of AP–diABZI as an adjuvant

therapy for adoptive OT-I T-cell transfer therapy in a B16.F10-OVA model. **f**, Schematic of B16.F10-OVA tumour inoculation and of treatment schedule with OT-I T cells (0.5 million cells) on either day 9 (OT-I alone or single dose AP–diABZI pre-treatment) or day 15 (three-dose AP–diABZI pre-treatment). **g–i**, Tumour growth curves (**g**), spider plots of individual tumour growth curves (**h**) and Kaplan–Meier survival curves (**i**) (*n* = 15 for PBS; *n* = 12 for all other treatments). In **g**, *P* values determined by two-way ANOVA with post hoc Tukey's correction for multiple comparisons for all groups compared to PBS on day 17. In **i**, end-point criteria of 1,500 mm³ tumour volume with *P* value determined by log-rank test for comparison to PBS group or for the comparisons indicated in the legend. Replicates are biological, and data are shown as mean ± s.e.m. Panels **a** and **f** created with BioRender.com.

of cancer cells in this model and that infiltrating myeloid cells can also express PD-L1 which may explain the modest <2-fold decrease in AP–Cy5 accumulation. Nonetheless, these studies support our hypothesis that integrating a PD-L1 binding domain further improves delivery to tumour tissue.

### AP–diABZI reprograms the TME to eliminate breast tumours and generate immunological memory that prevents recurrent disease

We next investigated the antitumour effects of systemically administered AP–diABZI fusion in the orthotopic EMT6 tumour model, comparing effects to those elicited by the constitutive components nAlb–diABZI and nPD-L1–diABZI (Fig. 5a–d). All nanobody carriers were administered intravenously at 1.25 μg of agonist. In addition, mice were treated with commercially available anti-PD-L1 ICB immunoglobulin G (IgG) antibody to model a US Food and Drug Administration-approved anti-PD-L1 ICI (for example, atezolizumab). A standard preclinical dose of 100 μg ICI was delivered intraperitoneally, which is a near-equivalent molar dose of administered nanobody based on antigen binding domains (that is, single domain for nanobody and two domains for antibody). Treatment with AP–diABZI completely eliminated observable EMT6 tumours, resulting in a 100% complete response rate (10/10 mice), whereas treatment with nAlb–diABZI, while still very effective, yielded a 30% complete response rate (3/10 mice); nPD-L1–diABZI only modestly inhibited tumour growth, although to slightly greater extent than the conventional anti-PD-L1 IgG ICI, which conferred only minimal activity in this model. It is worth noting that no additional toxicity was observed for AP–diABZI relative to nAlb–diABZI (Extended Data Fig. 2), although minor hepatic extramedullary haematopoiesis was noted. To further assess this, we compared AP–diABZI to a combination regimen of nAlb–diABZI and ICB (that is, anti-PD-L1 IgG) and observed comparably effective antitumour

responses, suggesting that the improved efficacy of AP–diABZI over nAlb–diABZI can largely be attributed to immune checkpoint inhibition. Mice treated with AP–diABZI and nAlb–diABZI + ICB that exhibited complete responses were rechallenged 80 days after the initial tumour inoculation with the injection of EMT6 cells in a distal MFP and tumour growth monitored without additional treatment. In both groups, mice were largely resistant to tumour re-challenge with only 1/9 (AP–diABZI) or 1/8 (nAlb–diABZI + ICB) mice developing a tumour with the others remaining cancer free until at least day 100, demonstrating induction of memory lymphocytes that recognize EMT6 tumour antigens (Fig. 5e,f). We next evaluated the antitumour efficacy of AP–diABZI in mice inoculated with parental or PD-L1 knockout EMT6 cells and found that it was less effective (100% versus 60% complete response rate) when PD-L1 was not expressed by breast cancer cells (Fig. 5g,h and Supplementary Fig. 25), potentially due to decreased tumour accumulation and/or reduced checkpoint inhibition. We also evaluated AP–diABZI in a mouse mammary tumour virus–polyoma middle T antigen (MMTV-PyMT) transgenic mouse model of spontaneous breast cancer, finding that systemic administration of AP–diABZI significantly reduced tumour burden without evidence of increased lung metastasis (Extended Data Fig. 4), which has been implicated as a potentially deleterious consequence of STING signalling in some preclinical models[60,61].

To gain insight into the mechanism underlying the increased efficacy of AP–diABZI, we treated mice bearing orthotopic EMT6 tumours with AP–diABZI, nAlb–diABZI or PBS, collected serum at 4 h following the first dose for analysis of serum cytokines (Supplementary Fig. 26) and euthanized mice 24 h after the third dose for gene expression analysis of tumour tissue using the NanoString PanCancer IO 360 panel (Fig. 5i–m and Extended Data Fig. 5). Administration of nAlb–diABZI and AP–diABZI increased serum levels of antitumour type I interferons (interferon-α, (IFNα), interferon-β (IFNβ)) and Th1 cytokines

(for example, IL-12, TNF), whereas nPD-L1–diABZI did not stimulate response, consistent with its low therapeutic efficacy; it is worth noting that only AP–diABZI notably increased levels of interferon-γ (IFNγ), a cytokine with an established role in antitumour immunity. Likewise, both nAlb–diABZI and AP–diABZI mediated considerable shifts in the gene expression profile, with transcript signatures associated with increased immune cell infiltrate (immune cell trafficking, CD8+ T cells, NK cells, Th1 cells), tumour immunogenicity (antigen presentation, T-cell priming, T-cell recognition, costimulation, cytokine/interferon signalling) and cancer cell death/apoptosis, with AP–diABZI tending to exert a stronger effect relative to nAlb–diABZI (Fig. 5k–m and Supplementary Fig. 27).

To further understand how AP–diABZI exerts potent antitumour effects, we performed flow cytometric immunophenotyping of EMT6 tumours 48 h following the first dose of nAlb–diABZI and AP–diABZI (Extended Data Fig. 6). We observed a decreasing frequency of live cancer cells (CD45−) within the tumour and found a significant decrease in proliferating (Ki67+) cancer cells, consistent with the potent antitumour effects induced by AP–diABZI and gene expression analysis supporting increased cancer cell death. It is worth noting that there was also an observed trend towards a decrease in PD-L1 expression within cancer cells. We found that a single dose of either nAlb–diABZI or AP–diABZI increased the infiltration of neutrophils and NK cells; more granulocytic MDSCs were also present, potentially contributing as an immunoregulatory mechanism to acute STING activation. However, as observed with nAlb–diABZI treatment, inhibition of MDSCs using SX-682 or anti-GR1 antibody depletion reduced AP–diABZI treatment efficacy (Supplementary Figs. 19, 20 and 28). While no change in the overall frequency of CD8+ T cells was observed at this early time point, tumour-infiltrating CD8+ T cells tended to display a more activated CD69+ phenotype, which was also reflected in the splenic T-cell population (Supplementary Fig. 29). Motivated by these data, we studied the tumour and spleen immune cell dynamics after treatment with one, two or three doses of AP–diABZI (Fig. 6 and Supplementary Figs. 30 and 31). We found that AP–diABZI increased the frequency of CD4+ T cells, CD8+ T cells and NK cells expressing markers of activation and proliferation, with a trend towards a stronger response after two and three doses, where a robust antitumour effect was observed (Fig. 6a–e). Consistent with observations following a single dose and the potent antitumour efficacy of AP–diABZI, the frequency of CD45−Ki67+ cancer cells was also reduced (Fig. 6a–d and Supplementary Fig. 32). This is also consistent with gene expression profiling (Fig. 5j–l) indicating increased NK and T-cell infiltration and tumouricidal activity. Within the tumour-infiltrating T-cell compartment, the percentage of CD8+ T cells increased with similar trends towards a more activated phenotype, and importantly, the ratio of CD8+ cells to FoxP3+ regulatory T cells was increased (Fig. 6b,c), indicative of a more immunogenic 'hot' immune profile within the TME. Furthermore, within CD8+ and CD4+ T-cell populations—both within the tumour and spleen—we observed a shift towards Ki67+CD69+ and Ki67+PD-1+ cells, indicating the prevalence of both proliferating and activated lymphocytes in response to AP–diABZI (Fig. 6d,e and Extended Data Fig. 7). Together, these data show that AP–diABZI increases the infiltration of CD8+ T cells and NK cells with an activated phenotype and that this effect is enhanced over the use of nAlb–diABZI alone, potentially implicating CD8+ T cells and NK cells as the primary antitumour effectors. To test this, we antibody-depleted NK cells, CD8+ T cells and CD4+ T cells and evaluated antitumour responses elicited by AP–diABZI treatment. Again, we observed a 100% complete response to AP–diABZI, but treatment efficacy was almost completely inhibited with CD8+ T cell or NK cell depletion, with CD8+ T-cell depletion having a slightly stronger effect (Fig. 6f–h); no effect of CD4+ T-cell depletion was observed. Therefore, both NK cells and CD8+ T cells are essential to the potent efficacy of AP–diABZI in an EMT-6 breast tumour model.

## AP–diABZI inhibits B16.F10 tumour growth and primes an antigen-specific memory CD8+ T-cell response in situ

We next assessed the efficacy of AP–diABZI in a more challenging and immunosuppressive B16.F10 melanoma model, initiating the three-dose treatment regimen when subcutaneous tumours reached an average size of ~75 mm³. As expected in this model, anti-PD-L1 ICB exerted no therapeutic benefit, whereas both nAlb–diABZI and AP–diABZI suppressed tumour growth and elongated median survival time, with AP–diABZI conferring the most survival benefit, consistent with findings in the EMT6 model (Fig. 7a–d). We also found that AP–diABZI was more effective than free diABZI administered at 24 times the dose (30 μg) in the B16.F10 model (Supplementary Fig. 33). We again evaluated cytokine levels in plasma 4 h following the first injection and found that anti-PD-L1 ICB increased only IL-1α levels, while nAlb–diABZI and AP–diABZI stimulated the production of cytokines and chemokines associated with antitumour immunity, including IFNα, IFNβ, IFNγ, IL12p70 and CXCL10 (Extended Data Fig. 8). To determine the primary cellular effectors to AP–diABZI in the B16.F10 model, we antibody-depleted CD4+ and CD8+ T cells and NK cells again finding that the antitumour response was mediated predominantly by CD8+ T and NK cells (Extended Data Fig. 9).

STING activation can prime the immune system to stimulate a systemic, antigen-specific, antitumour T-cell response with potential to lead to generation of T-cell memory[19,62]. Given evidence of increased antigen presentation, cancer cell killing and T-cell priming, as well as protection from tumour re-challenge in mice with EMT6 tumours treated with AP–diABZI (Fig. 5), we next assessed the capacity of AP–diABZI to stimulate a de novo tumour antigen-specific CD8+ T-cell response. To test this, we inoculated C57BL/6 mice with B16.F10 melanoma cells expressing ovalbumin (B16.F10-OVA) as a model antigen and treated mice with either PBS or AP–diABZI on a three-dose regimen once tumours reached a size of 75–100 mm³ (Fig. 7e–m). At 24 h after the final dose, mice were euthanized for flow cytometric evaluation of splenic T-cell response. Consistent with results in mice with parental B16.F10 tumours, AP–diABZI treatment significantly reduced tumour burden (Fig. 7f). Treatment with AP–diABZI resulted in a significant increase in activated CD69+ CD4+ and CD8+ T cells (Fig. 7h) and effector memory (CD44+CD62L−) CD4+ and CD8+ T cells, with a reduction in CD4+ central memory (CD44+CD62L+) T cells. Using SIINFEKL/H-2Kb tetramer staining, we also found that AP–diABZI treatment stimulated a strong peripheral ovalbumin (OVA)-specific CD8+ T-cell response (Fig. 7l), characterized by a predominantly (~60%) CD44+CD62L− effector memory phenotype (Fig. 7m and Extended Data Fig. 10). Hence, in addition to remodelling the TME, systemic administration of AP–diABZI primes antigen-specific CD8+ T-cell effector and memory responses capable of targeting tumour-associated antigens.

## Albumin-hitchhiking STING agonists inhibit lung metastatic disease

Based on the evidence that AP–diABZI can stimulate an effective antitumour immune response in the immunologically 'cold' B16.F10 model, we extended our investigations to evaluate therapeutic efficacy in an aggressive model of lung metastatic melanoma induced through intravenous inoculation of luciferase-expressing B16.F10 (B16.F10-Luc) cells (Fig. 8a). A week following inoculation, we used the three-dose combination therapy regimen described previously. On day 17 after inoculation, mice were euthanized, and lungs were collected for quantification of tumour burden via measurement of lung mass, immunohistochemistry and bioluminescence imaging (Fig. 8b–e and Supplementary Fig. 34). High metastatic tumour burden was evident in mice receiving anti-PD-L1 ICB alone but significantly reduced in mice receiving nAlb–diABZI and nearly eliminated in mice receiving AP–diABZI. It is worth noting that these data show that albumin-hitchhiking STING agonists are effective against metastases in the lung, one of the most common metastatic sites for many cancers. This also suggests a potential to treat

micrometastases, which typically lack the leaky vasculature required for tumour accumulation via the enhanced permeation and retention effect[63]; by contrast, albumin-binding molecules have been shown to accumulate in micrometastases[64].

### AP–diABZI opens a therapeutic window for adoptive T-cell transfer therapy

Finally, we sought to demonstrate the versatility of our strategy by extending the application of AP–diABZI to the setting of adoptive cellular immunotherapy[65], which includes tumour-infiltrating lymphocyte therapy, chimeric antigen receptor T cells, and T-cell-receptor-engineered T cells. Adoptively transferred T cells face major barriers to tumour infiltration and function, which continues to limit their clinical impact in the treatment of solid tumours[66,67]. Based on data showing that nAlb–diABZI and AP–diABZI enhance the infiltration of endogenous antitumour T cells, we hypothesized that the approach could be used to pre-condition the TME to generate a therapeutic window for adoptive T-cell therapy. To test this, we inoculated female C57BL/6 mice with subcutaneous B16.F10-OVA cells and allowed the tumours to reach approximately 75 mm$^3$ (Fig. 8f). We then treated mice with either one or three doses of AP–diABZI, followed by a single intravenous dose of activated OVA-specific activated CD8$^+$ T cells (OT-I T cells). Treatment with OT-I T cells only (no STING agonist) on day 9 resulted in marginal therapeutic benefit (Fig. 8g,h), consistent with the highly immunosuppressive B16.F10 TME that restricts T-cell infiltration and effector function. However, treatment with OT-I T cells 48 h after either one or three AP–diABZI doses conferred significant reduction in tumour growth and prolonged mouse survival (Fig. 8i). It is worth noting that the treatment regimen of three doses of AP–diABZI before one dose of OT-I T cells resulted in a 25% complete response rate (3/12 mice). This provides additional evidence that albumin-hitchhiking STING agonists can establish an inflammatory milieu that supports T-cell infiltration and function. While here we used a simplified model of an adoptive T-cell therapy, these studies highlight the potential to leverage nanobody–STING agonist conjugates to enhance responses to multiple T-cell-based immunotherapies, including autologous tumour-infiltrating lymphocyte therapy, chimeric antigen receptor T cells and cancer vaccines.

## Discussion

Innate immunity fuels the cancer immunity cycle, playing critical roles in antitumour T-cell priming, recruitment of cytotoxic immune cells, and recognition of tumour antigens[68–70]. However, the development of innate immune agonists targeting specific PRRs has been limited by pharmacological barriers that have largely restricted their application to intralesional administration[4], which has yet to deliver on its clinical promise[11]. This challenge has been recently exemplified by the clinical exploration of STING agonists, which have demonstrated impressive results when administered intratumourally in mouse models but have not yet proven effective in patients. To address this, we developed a drug carrier for systemic delivery of STING agonists based on an albumin-hitchhiking nanobody (nAlb) engineered for precisely defined and site-selective ligation of a DBCO-functionalized 'clickable' diABZI cargo that we synthesized. Our data show that intravenously administered nAlb conjugates bind to circulating albumin in situ, increasing nanobody half-life from minutes to days and harnessing the capacity of albumin to accumulate in tumours for delivery of cargo to cancer cells and tumour-associated myeloid cells in the TME. This triggered potent STING activation at tumour sites, initiating an inflammatory program that increased the infiltration of activated NK cells and CD8$^+$ T cells with antitumour function. Accordingly, nAlb–diABZI conjugates show improved efficacy in mouse models of breast cancer and melanoma relative to a leading free diABZI agonist.

An appealing feature of anti-albumin nanobodies and other protein-based albumin-hitchhiking agents (for example, affibodies, humabodies, albumin binding domains derived from *Streptococcus* protein G) over other albumin binders (for example, lipids, Evans blue) is the high degree of molecular programmability that can be achieved through protein engineering. Here we illustrate this modularity by recombinantly integrating a PD-L1 binding domain to create a bivalent fusion protein for covalent conjugation of diABZI. This yielded a single, well-defined, multifunctional STING agonist that increased tumour accumulation in a PD-L1-dependent manner, while also blocking an important immune checkpoint, resulting in spontaneous induction of tumour antigen-specific T cells that inhibited tumour growth and provided immunological memory that protected against tumour rechallenge. While we selected PD-L1 on translational considerations, the bivalent nanobody system is readily amenable to integration of other immunoregulatory features and/or molecular targeting ligands. To date, there are sparingly few reports describing the targeted delivery of STING agonists[71,72], with most using surface-decorated nanoparticles for CDN delivery[73–75]. Our albumin-hitchhiking nanobody approach offers several potential advantages including plug-and-play programmability, precise and site-selective ligation of STING agonists and a smaller size that has been reported to improve tumour penetration, a limitation of nanoparticles and full-length antibodies in tumours with dense stroma[76–78].

Although there are vast future opportunities for bivalent nanobody–agonist conjugates, it is also notable that nAlb–diABZI was highly effective as a single agent, which may be advantageous for cancers that lack a defined cell surface target. While still incompletely understood, albumin can accumulate in tumour tissue through several interrelated mechanisms that are largely enabled by its long circulation time, including the enhanced permeability and retention effect, active transport via endothelial cell transcytosis, binding to SPARC produced by cancer cells, and cellular uptake and catabolism by cancer and tumour-associated immune cells such as macrophages[27,29]. Indeed, our data show that albumin hitchhiking dramatically increases nanobody and drug half-life, allowing nAlb–diABZI to accumulate in tumour tissue where it is internalized by cancer and tumour-associated immune cells to activate STING. To date, most research on albumin-based drug carriers for cancer (for example, Abraxane, aldoxorubicin) has focused on delivery of chemotherapy drugs that target cancer cells. By contrast, immunostimulatory agents such as STING agonists can stimulate complex antitumour immunological programmes that may be more dependent on immunological variables (for example, neoantigen load, immune status of the TME) than on the efficiency of drug accumulation in tumour tissue or delivery to cancer cells. For example, in our analysis of nAlb–Cy5 biodistribution, we found ~11%ID per g tumour in the EMT6 model and a comparable ~8.4%ID per g tumour in B16.10 model (Fig. 2), yet a substantial difference in response to both nAlb–diABZI alone and in combination with anti-PD-L1 that may be attributed to the relatively low immunogenicity of B16.F10 tumours. It is worth noting that the efficacy of nAlb–diABZI was enhanced when delivered in combination with anti-PD-L1 ICB and therefore may hold promise when combined with other ICIs and as an adjuvant therapy for patients with acquired resistance to ICIs. In addition, nAlb–diABZI was much more effective than nPD-L1–diABZI, which was cleared rapidly with minimal tumour accumulation despite a capacity to activate STING, bind PD-L1 and inhibit immunoregulatory PD-L1/PD-1 signalling[79]. This finding contributes to an evolving understanding of how the spatiotemporal dynamics of immunomodulatory signals impact the efficacy and safety of systemically delivered innate immune agonists and other immunotherapies[55,80–82]. Indeed, anti-albumin nanobodies have been engineered with variable affinity[83], and this may afford a future opportunity for precisely modulating plasma half-life to establish immunopharmacological principles for optimizing systemic innate immune agonist delivery. However, clinical imaging has demonstrated that albumin accumulation varies among cancer types and patients[29], and the implications of this for the activity and efficacy of nAlb–diABZI must be considered and further investigated.

Also critical to the efficacy of our technology was the design and synthesis of a diABZI STING agonist functionalized with a DBCO group for biorthogonal conjugation to azide-presenting nanobodies. Despite being stably linked to the nanobody via an amide bond, the STING agonist showed high potency in vitro and in vivo, which we attribute to lysosomal degradation of endocytosed diABZI–nanobody conjugates and release of an active species (Supplementary Fig. 6 and Extended Data Fig. 1). While there may be an advantage to using such stable linkers to minimize premature drug release into the circulation[84], our strategy also opens the possibility of installing cleavable linkers (for example, enzyme cleavable, reactive oxygen species cleavable) that enable environmentally responsive, 'logic-gated' drug release with potential to further improve tumour specificity and reduce systemic exposure[85,86]. In addition, while our selection of a diABZI agonist was largely motivated by their recent advancement into clinical trials, the strategy is also amenable to conjugation to other STING agonists (for example, recently described conjugatable CDNs)[18,22] as well as agonists targeting other PRRs[87,88].

In summary, we have integrated synthetic biology tools to engineer precisely defined nanobody–STING agonist conjugates for cancer immunotherapy. We leveraged albumin-binding nanobodies as a scaffold from which diverse immunomodulatory components can be readily integrated via recombinant and chemical design. We demonstrated that albumin-hitchhiking nanobodies enhanced the potency and efficacy of a diABZI STING agonist, and we showcased the versatility of the system by introducing a PD-L1 binding nanobody that affords increased tumour targeting and immune checkpoint inhibition to further potentiate antitumour immunity and efficacy. We found nanobody–diABZI conjugates to be highly effective in an orthotopic breast cancer model and an aggressive model of lung metastatic melanoma, and we further demonstrated their utility as a neoadjuvant therapy to improve responses to adoptive T-cell transfer. It is worth noting that nanobody–diABZI conjugates were well tolerated with a favourable preclinical toxicity profile and are amenable to established scalable manufacturing workflows and translational pipelines. Collectively, our study establishes a preclinical foundation for future development of nanobody–STING agonist and other protein–STING agonist conjugates as enabling technologies for cancer immunotherapy.

## Methods

### Materials and cell lines
All chemicals involved in synthesis of target compounds were reagent grade unless stated otherwise. DNase, isopropyl thiogalactoside and dimethyl sulfoxide (DMSO) were purchased from Sigma-Aldrich. Azido-PEG$_3$-amine and DBCO-PEG$_{12}$-N-hydroxysuccinimide (NHS) ester were purchased from BroadPharm. Magnesium sulfate, sodium hydroxide, sodium azide, sodium acetate, sodium chloride, sodium bicarbonate, sodium hydroxide, 2xYT media, kanamycin, protease inhibitor cocktail tablets (EDTA free), nickel-nitrilotriacetic acid (NTA) resin and all other organic solvents were purchased from Thermo Fisher Scientific. All DNA block segments involved in cloning protein inserts were purchased from Integrated DNA Technologies with standard desalting as the means of purification. For protein expression, pET28-b(+) expression vector, Q5 Hot Start Master Mix 2x, T4 DNA ligase, Golden Gate Master Mix (BsaI-HF v2), DH5α *Escherichia coli*, and T7 SHuffle Express *E. coli* chemically competent cells were purchased from New England Biolabs. QIAprep Spin Miniprep kits were purchased from Qiagen. THP1-Dual and A549-Dual cell lines were purchased from InvivoGen. A549-Dual cells were cultured in Dulbecco's modified Eagle medium (DMEM; Gibco) supplemented with 2 mM L-glutamine, 4.5 g l$^{-1}$ glucose, 10% heat-inactivated fetal bovine serum (HI-FBS; Gibco), 100 U ml$^{-1}$ penicillin/100 µg ml$^{-1}$ streptomycin (Gibco) and 100 µg ml$^{-1}$ Normocin. THP1-Dual cells were cultured in Roswell Park Memorial Institute (RPMI) 1640 Medium (Gibco) and was supplemented with 2 mM L-glutamine, 4.5 g l$^{-1}$ glucose, 10% HI-FBS (Gibco),

100 U ml$^{-1}$ penicillin, 100 µg ml$^{-1}$ streptomycin (Gibco) and 100 µg ml$^{-1}$ Normocin. Every other passage, both blasticidin (InvivoGen) and Zeocin (InvivoGen) were added at a concentration of 200 µg ml$^{-1}$ to the cell culture flask. The mouse breast cancer cell line EMT6 and melanoma cell lines B16.F10 and B16.F10-Luc2 were purchased from American Type Culture Collection, where EMT6 cells were grown in RPMI supplemented with 2 mM L-glutamine, 4.5 g l$^{-1}$ glucose, 10% HI-FBS, 100 U ml$^{-1}$ penicillin and 100 µg ml$^{-1}$ streptomycin. B16.F10 and B16.F10-Luc2 cells were cultured in DMEM supplemented with 2 mM L-glutamine, 4.5 g l$^{-1}$ glucose, 10% HI-FBS, 100 U ml$^{-1}$ penicillin and 100 µg ml$^{-1}$ streptomycin. B16.F10-OVA cells were a gift from A. Lund and were cultured in DMEM supplemented with 2 mM L-glutamine, 4.5 g l$^{-1}$ glucose, 10% HI-FBS, 100 U ml$^{-1}$ penicillin and 100 µg ml$^{-1}$ streptomycin with continuous selection using geneticin (G418; Gibco) after every cell passage at a concentration of 500 µg ml$^{-1}$. NCI-H358 cells stably expressing Gal9–mCherry were a gift from M. J. Munson and cultured and used to evaluate endosomal disruption as previously described[38]. All cell types used in the study were grown in a humidified atmosphere at 37 °C in 5% CO$_2$.

### Cloning of proteins
Gene cassette was purchased from Integrated DNA Technologies in the form of a gene block, with cloning restriction sites placed on both flanking regions (BsaI–GGTCTC). In the case of a fusion protein, a genetic sequence was placed between the two domains (XTEN–SGSET-PGTSESA). For sortase-mediated bioconjugation of nanobodies, a C-terminal sequence was incorporated (LPETGGHHHHHHEPEA). The gene fragment was digested with BsaI-HF v2 in a Golden Gate master mix (New England Biolabs) and ligated into a pET28-b(+) plasmid. The construct was transformed into chemically competent DH5α *E. coli* (New England Biolabs) and plated on Luria–Bertani agar with kanamycin. The sequence-verified pET28b plasmid was transformed into T7 Shuffle Express *E. coli* (New England Biolabs) as the expression strain. Glycerol stocks of every successfully transformed bacterial strain were maintained at −80 °C. Protein sequences are summarized in Supplementary Table 1.

### Expression and purification of proteins
About 5 µl of kanamycin (stocked at 50 mg ml$^{-1}$) was added to a culture tube containing 5 mL 2xYT media and inoculated with a stab of culture stock (cloned into a New England Biolabs T7 Shuffle Express cell line). The culture was incubated at 30 °C, with shaking at 250 rpm for 16 h. Each culture was transferred to a 2 l baffled flask containing 500 ml of autoclaved 2xYT media and 500 µl of kanamycin (25 mg) and shaken at 30 °C in an Innova 42R (New Brunswick Scientific) incubator for 4.5–5 h (until the optical density at 600 nm reached ~0.8). The cultures were then cooled to ~16 °C and induced with isopropyl thiogalactoside (2.5 mM final concentration). The induced cultures were shaken overnight (20–24 h) at 16 °C. The bacteria were collected the next day by centrifugation (3,900 rpm for 10 min), and the pellet was reconstituted in 1× PBS with DNase I and a tablet of protease inhibitor cocktail (EDTA free). The cells were lysed by sonication on an ice bath in 5 s increments over 10 min. The resulting bacterial lysate was centrifuged (11,000 rpm for 20 min) to remove cellular debris. The supernatant was added to a 50 ml Kontes Flex column (Kimbal Kontes Glassware) containing 3 ml of nickel-NTA histidine binding resin that was pre-equilibrated with 1× PBS buffer. This column was placed on a rotating shaker at room temperature for 1–2 h. After this period, the supernatant was drained from the column using gravity, and the column washed with 1× PBS buffer twice. Weakly bound proteins were first washed off the resin using a low-concentration elution buffer (2 × 10 ml, 10 mM imidazole, 1× PBS pH 7.4 at 25 °C). The bound protein was then eluted from the resin using elution buffer (15 ml, 150 mM imidazole, 1× PBS pH 7.4 at 25 °C). The eluate was then concentrated to 0.5 ml in a 15 ml Microcon 10 kDa Centrifugal Filter Unit (Millipore) and subsequently purified by

size exclusion chromatography using ÄKTA pure (Cytiva) fast protein liquid chromatography on a Hi-Load 16/60 Superdex 200 column using 1× PBS at pH 7.4 as the running buffer at 4 °C. Pure fractions were determined by SDS–PAGE, pooled together with buffer exchange to 1× PBS and stocked at either −20 °C or 4 °C.

### Enzymatic bioconjugation and click chemistry reactions
Bioconjugation reactions occurred in mild conditions (20 mM HEPES at pH 7.4, 150 mM NaCl and 10 mM $CaCl_2$) between eSrtA (100 μM) and a nanobody containing a C-terminal ligation tag (75 μM) using a primary amine containing functional group (20 mM), here azido-$PEG_3$-amine (BroadPharm). Reactions occurred with mixing by a rotary shaker overnight (16 h) and were quenched by the addition of a 1:1 volume of a chelating agent EDTA containing solution (20 mM HEPES at pH 7.4, 300 mM NaCl and 10 mM EDTA) under rotation for 1 h. After the reaction was stopped, the solution was concentrated and buffer changed to 1× PBS (without NaCl or $MgCl_2$) three times by centrifugal dialysis. The protein solution was then immobilized to nickel-NTA histidine binding resin for at least 2 h, and unbound protein was collected by washing the resin with 1× PBS. For nanobodies that contain a histidine in the native sequence, proteins were eluted in mild conditions (10 mM imidazole in 1× PBS). Collected protein was concentrated and buffer changed to 1× PBS by centrifugal dialysis and verified by ESI–MS and SDS–PAGE. Click chemistry reactions were made to proceed by the addition of 5 equiv. (molar) of the complementary handle (for example, if an azide was placed on the nanobody, the click chemistry reaction would proceed with the addition of 5 equiv. of DBCO-containing moiety). For Cy5 conjugations, sulfo-Cy5–DBCO was purchased from Sigma Aldrich, and Cy5–DBCO was purchased from BroadPharm. After 48 h of reaction between the protein-azide and the DBCO-moiety under rotation at room temperature, the mixture was purified by centrifugal dialysis four times with 1× PBS and verified for purity by ultraviolet–visible spectroscopy, ESI–MS and SDS–PAGE.

### SDS–PAGE
Protein samples were diluted in 1× PBS to 10 μM before analysis. About 10 μl of the protein sample was mixed with 10 μl of reducing Laemmli buffer. Samples were boiled at 95 °C for 5 min, and 15 μl of each sample was loaded into a 15-well, 4–15% Tris-glycine precast polyacrylamide gel (Biorad) and ran at a constant 150 V with 343 mA for 30 min. The gel was then either first imaged for Cy5 fluorescence on a UV-transilluminator or directly stained using Coomassie-B-250.

### ESI–MS
Proteins were buffer exchanged into ammonium acetate (pH 5.5) and concentrated to approximately 100 μM. ESI–MS data were collected using an Agilent 6210A time-of-flight (TOF) mass spectrometer at a range of 50–20,000 $m/z$ over a period of 2 min. Data were analysed with Agilent MassHunter IM-MS Acquisition Data software (version 5.3) to reveal $m/z$ data, where files were condensed across the 2 min run. These $m/z$ data were deconvoluted using a maximum entropy deconvolution calculation using UniDec (version 8.0.1) to give the deconvoluted mass spectra using background subtraction in a range of 1,000–5,000 $m/z$ and with an export range of 5,000–50,000 Da.

### Computational modelling and analysis of nAlb nanobody
nAlb was modelled in silico using RoseTTAFold (GitHub; RosettaCommons), and binding between HSA (PDB, 1AO6) and nAlb was predicted using RosettaDock through ROSIE (Rosetta Online Server that Includes Everyone; Pittsburgh). After an initial screening for best fits of the docking between HSA (receptor) and nAlb (ligand), the best fit model was then returned for rescreening to confirm an optimal energy conformation between the structures. The final structures of nAlb and the bound nAlb–HSA complex were exported to PyMOL (version 3.1) for generating a figure of the structure.

### Isothermal titration calorimetry
All proteins used were equilibrated in buffer at indicated pH values by titration using HCl (aqueous) or NaOH (aqueous) in PBST, 150 mM NaCl, 3 mM EDTA and 0.05% Tween 20. Albumin (HSA and recombinant mouse serum albumin; Sigma) and PD-L1 titrations were run on the TA Instruments Affinity ITC instrument (TA Instruments). About 350 μl of albumin or PD-L1 (Sigma) was added to the sample cell (10–20 μM), and either the nAlb nanobody (125–250 μM) or nPD-L1 nanobody (250 μM) was loaded into the injection syringe, respectively. The reference cell contained ultrapure water which was changed after each titration experiment. All runs used the following instrument settings: cell temperature 298 K, reference power 10 μCal s$^{-1}$, initial delay 240 s, stirring speed 75 rpm, feedback mode/gain high and injection volume 2 μl for 10 s, titration spacings at 120 s intervals and a filter period of 10 s. Data were analysed using the provided NanoAnalyze software (version 2.1) for the instrument to determine thermodynamics of binding from an independent model.

### Tritosome degradation assay and MALDI-TOF MS
Tritosomes (XenoTech/BioIVT) were prepared and activated by combining 70 μl of nuclease-free water, 10× of catabolic buffer (K5200, BioIVT) and 100 μl of pure lysosomes (H0610.L, BioIVT) and incubating the mixture at 37 °C for 15 min. Samples for lysosomal degradation were added at 0.5 μM (10 μl) with the tritosome mixture and incubated at 37 °C over a period of 48 h. Aliquots were taken from the reaction mixture at distinct time points, flash frozen with liquid nitrogen and stored at −80 °C to stop the reaction. Activity was determined by observing molecular weight shifts in the substrate using MALDI–time-of-flight MS (MALDI–TOF MS). About 3 μl of matrix (15 mg ml$^{-1}$ 2,4,6-trihydroxyacetophenone in dry acetone) was combined with 1 μl of aliquoted sample and spotted on a stainless steel MALDI–MS plate (Bruker). Matrix was evaporated using compressed air, read on a Bruker AutoFlex MALDI–TOF and processed with the FlexControl software (version 3.4, Bruker Daltonics). The laser pulse rate was 500 Hz, and spectra were obtained with a mass window of 400–4,000 $m/z$ at the highest resolution for the instrument (4.00 GS/s data acquisition rate). FlexAnalysis software (version 3.4, Bruker Daltonics) was used to obtain baseline spectra for all samples. Data were exported and plotted using MATLAB (version 2023b) to generate figures showing $m/z$ spectra at distinct time points.

### Synthesis and nuclear magnetic resonance verification of DBCO-$PEG_{11}$–diABZI
Synthesis of the DBCO-conjugated STING agonist (diABZI) is described in Supplementary Fig. 2, with nuclear magnetic resonance (NMR) verification presented in Supplementary Figs. 3 and 4. We first generated a STING agonist that features a reactive amine handle, which was synthesized in four steps. Briefly, aryl amination of an aryl chloride **1** with an amine **2** gave a di-nitro analogue, compound **3**. The di-nitro compound **3** was subjected to reduction using sodium dithionite in methanol, generating a di-amine moiety **4**. Compound **4** was then treated with isothiocyanate, followed by 1-ethyl-3-(3-dimethylaminopropyl)carbodiimide coupling, to reveal a boc-protected analogue, compound **5**. Next, the boc group from compound **5** was deprotected by treating with 1:1 trifluoroacetic acid:dichloromethane. To a stirred solution of amine **6** (100 mg, 0.089 mmol, 1 equiv.) in 5 ml dimethylformamide, Hunig's base was added (77 μl, 0.44 mmol, 5 equiv.) under argon atmosphere at room temperature. After stirring for 5 min, a solution of activated NHS ester (98 mg, 0.098 mmol, 1.1 equiv.) in dimethylformamide (5 ml) was added dropwise and stirred overnight (16 h). The solvent was evaporated to get crude product **7**, which was purified by silica gel column chromatography using a mixture of methanol/dichloromethane as an eluent (5% to 25% MeOH) to get the desired product as a solid (70 mg, 0.042 mmol, yield 43%). ($R_f$ = 0.5 in 20% MeOH in dichloromethane). $^1$H NMR (400 MHz, DMSO) δ 8.01–7.93 (m, 2H), 7.88 (t, $J$ = 5.7 Hz, 1H),

7.75 (t, $J$ = 5.7 Hz, 1H), 7.67–7.60 (m, 4H), 7.49–7.42 (m, 3H), 7.38–7.27 (m, 7H), 6.49 (d, $J$ = 7.1 Hz, 2H), 5.88–5.79 (m, 2H), 5.01 (d, $J$ = 13.9 Hz, 1H), 4.91 (dd, $J$ = 29.6, 4.2 Hz, 4H), 4.53–4.49 (m, 4H), 3.98 (t, $J$ = 6.0 Hz, 2H), 3.72 (s, 3H), 3.60–3.57 (m, 2H), 3.54 (t, $J$ = 6.5 Hz, 2H), 3.47 (broad s, 46H), 3.30–3.26 (m, 2H), 3.14–3.05 (m, 4H), 2.26 (t, $J$ = 6.5 Hz, 2H), 2.09 (s, 3H), 2.08 (s, 3H), 2.01–1.96 (m, 1H), 1.78–1.72 (m, 1H), 1.68 (p, $J$ = 6.5 Hz, 2H), 1.28–1.27 (m, 6H). $^{13}$C NMR (151 MHz, DMSO) δ 171.57, 171.50, 170.52, 168.06, 167.33, 152.50, 152.45, 152.06, 148.88, 145.50, 145.28, 144.65, 140.37, 140.33, 132.87, 130.54, 130.49, 130.07, 129.37, 128.62, 128.58, 128.44, 128.25, 128.13, 127.24, 125.60, 122.99, 121.86, 120.11, 120.04, 114.67, 109.72, 108.61, 106.00, 105.83, 105.58, 70.21, 70.14, 70.10, 70.00, 69.94, 69.45, 67.26, 56.45, 55.34, 53.85, 46.05, 45.05, 42.12, 38.95, 36.57, 35.61, 30.80, 30.17, 29.13, 18.46, 17.17, 16.58, 13.57, 12.74. HRMS (ESMS) calculated for $C_{84}H_{111}N_{15}O_{21}$ [M + Na]$^+$: 1,688.7977, found 1,688.7982.

### Synthesis and NMR verification of amine-PEG$_3$-Triazole-PEG$_{11}$–diABZI

To a stirred solution of amine-PEG$_3$-azide (3.14 mg, 14.4 µmol, 3 equiv.) in a 1:1 MeCN:H$_2$O mixture (4 ml), Hunig's base (3.10 mg, 24.0 µmol, 5 equiv.) and 450 µl of a stock solution of 10.8 mM DBCO-PEG$_{11}$–diABZI in DMSO (8.0 mg, 4.8 µmol, 1 equiv.) were added and stirred overnight. Acetonitrile was removed by slowly passing an air stream through the reaction flask. When the reaction mixture reduced to half, the aqueous mixture was frozen at −80 °C for 8 h and then lyophilized. Diethyl ether was added (×3) and vigorously shaken with diethyl ether. The mixture was decanted to remove excess amine-PEG$_3$-azide. After three washes with diethyl ether, it was dried overnight in a vacuum chamber to obtain the desired compound (7 mg, 3.7 µmol, 77% yield). 1H NMR (400 MHz, DMSO) δ 8.03–7.95 (m, 2H), 7.90 (t, $J$ = 5.7 Hz, 1H), 7.85–7.74 (m, 2H), 7.67 (d, $J$ = 3.8 Hz, 1H), 7.65–7.62 (m, 2H), 7.59–7.52 (m, 2H), 7.51–7.45 (m, 1H), 7.40–7.26 (m, 7H), 6.50 (d, $J$ = 7.1 Hz, 2H), 5.88–5.78 (m, 2H), 5.01–4.86 (m, 4H), 4.63–4.45 (m, 6H), 4.10–4.02 (m, 2H), 3.99 (t, $J$ = 6.0 Hz, 2H), 3.76 (t, $J$ = 5.5 Hz, 2H), 3.73 (s, 3H), 3.47 (broad s, 60H), 2.96–2.88 (m, 5H), 2.27 (t, $J$ = 6.0 Hz, 2H), 2.20–2.13 (m, 2H), 2.10 (s, 3H), 2.09 (s, 3H), 2.03–1.91 (m, 3H), 1.79–1.66 (m, 2H), 1.57–1.44 (m, 1H), 1.37–1.31 (m, 1H), 1.28–1.22 (m, 6H). MALDI-TOF MS calculated for $C_{92}H_{129}N_{19}O_{24}$ [M + H]$^+$: 1,884.9458, found 1,884.150.

### In vitro reporter cell assays

Cell reporter assays to evaluate STING agonist activity were performed in THP1-Dual and A549-Dual cell lines, as adapted from the manufacturer's protocols. Briefly, cells were plated at a density of 50,000 cells per well in a total volume of 180 µl of supplemented media in cell-culture treated 96-well plates overnight. After 24 h, cells were dosed with 20 µl of treatment groups (for a total volume of 200 µl per well in a 10:1 dilution, with either a 1:1 or 2:1 dilution down the plate) overnight. After 24 h of treatment, cells were pelleted at 1,500 rpm for 5 min in a centrifuge, and 20 µl of supernatant was plated in a white-walled 96-well plate for analysis by QUANTI-Luc (InvivoGen) assay. After loading in a plate reader with a luminescence detector (BioTek SynergyHT), 50 µl of QUANTI-Luc reagent was added to each well, and luminescence was measured for determination of cell-based activity. To the cells remaining in the 96-well plate, 30 µl of Cell-Titer Glo reagent (Promega) was added, and the plate was incubated at 37 °C for 1 h. After incubation, the plate was loaded into a plate reader, and luminescence was measured to determine cell viability.

### In vitro BMDC/BMDM maturation and activation

Bone marrow primary cells were collected from both the femur and tibia of female C57BL/6 mice, aged between 6 and 8 weeks. After collecting bones, cells were flushed with cold 1× PBS, centrifuged at 1,500 rpm for 5 min and resuspended in complete media (RPMI 1640 supplemented with 10% HI-FBS (Gibco), 100 U ml$^{-1}$ penicillin, 100 µg ml$^{-1}$ streptomycin (Gibco), 2 mM L-glutamine, 10 mM HEPES, 1 mM sodium pyruvate, 1× non-essential amino acids and 50 µM β-mercaptoethanol.

About 20 ng ml$^{-1}$ GM-CSF was added to culture BMDCs, and 20 ng ml$^{-1}$ of M-CSF was added to culture BMDMs. A single-cell suspension was generated by passing the collected cells through a 70 µm sterile cell strainer (Fisherbrand; Thermo Fisher Scientific), and cells were then plated in non-tissue-culture-treated petri dishes (Corning) and incubated at 37 °C with 5% CO$_2$. Cells were provided with fresh culture media supplemented with GM-CSF or M-CSF on days 3, 5 and 7. On day 8, the cells were collected and confirmed for either CD11c$^+$ expression (BMDCs) or CD11b$^+$F4/80$^+$ expression (BMDM) using flow cytometry (CellStream, Cytek Biosciences) with fluorescent anti-CD11c (Clone N418; BioLegend), anti-CD11b (Clone M1/70, BioLegend) and anti-F4/80 (Clone BM8, BioLegend) antibodies. Primary cells were seeded into 12-well plates for analysis by qPCR or 96-well plates for in vitro flow cytometry.

### Quantitative reverse transcriptase PCR

RNA was extracted either from animal tissue using a TissueLyser II (Qiagen) or from in vitro cell cultures using the RNeasy Plus Mini Kit (Qiagen) according to the manufacturer's protocol. Complementary DNA was generated through a reverse transcriptase reaction using the iScript cDNA synthesis kit (Bio-Rad) by following the manufacturer's instructions. To run the qPCR, cDNA was mixed with TaqMan gene expression kits (primer and master mix) to a final volume of 20 µl and run on the Bio-Rad CFX Connect Real-time System, with a threshold cycle number determination made by the Bio-Rad CFX manager software V.3.0. Primers used included mouse *Ifnb1* (Mm00439552_s1), mouse *Tnf* (Mm00443258_m1), mouse *Cxcl10* (Mm00445235_m1), mouse *Cxcl1* (Mm04207460_m1) and mouse *Hmbs* (Mm01143545_m1). Gene expression was first normalized to the housekeeping gene, *Hmbs*, and then normalized to the PBS treatment within groups using the 2$^{-ddCt}$ analysis method.

### Colocalization analysis

EMT6 or RAW 264.7 cells were plated at a density of 10,000 cells per well in a glass-bottom 96-well plate. nAlb–Cy5 (2 µM) and nGFP–Cy5 (2 µM) were added to each well, and these were further incubated for 4 h. Cells were treated with 50 nM of Lysotracker Green (Invitrogen) and 2 µM Hoechst (Invitrogen) for 10 min after the end of the treatment period. Wells were washed three times with phenol red free medium and visualized using a confocal microscope (Zeiss LSM880). High-magnification images were obtained using ×40 objective lens. Manders' coefficient was calculated by using ImageJ software (version 1.5.1) for colocalization analysis.

### Flow cytometry for in vitro uptake studies

EMT-6 and RAW 264.7 cells were seeded in 6-well plates at 5 × 10$^5$ cells per well. After 24 h, cells were treated with and without EIPA (50 µM) for 1.5 h. nAlb–Cy5 (2 µM) and nGFP–Cy5 (2 µM) were added to each well, and these were further incubated for 4 h. To collect cells from plates, EMT6 cells were treated with 0.25% trypsin, and RAW 264.7 cells were collected by scraping. Both cells were washed three times with cold 1× PBS (1 ml) and stained with DAPI for live/dead staining. Cells were washed and suspended in a staining buffer (1× PBS, 2.5% FBS, 2.5 mM EDTA), and flow cytometry was performed to evaluate the frequency and MFI of Cy5-positive cells. BMDMs were generated from the bone marrow of 6- to 8-week-old female C57BL/6 mice as described above and confirmed to be CD11b$^+$/F4/80$^+$ by flow cytometry. Fluorescence acquisition was carried out on Cytek Aurora spectral flow cytometer and analysed on FlowJo V.10.8.1.

### Evaluation of nanobodies in tumour models

For B16.F10, B16.F10-Luc or B16.F10-OVA tumour models, female 6- to 8-week-old C57BL/6 mice (The Jackson Laboratory) were used. For EMT6 tumour models, 6- to 8-week-old old Balb/C female mice (The Jackson Laboratory) were used. Tumours were generated in B16.F10

and B16.F10-OVA models by subcutaneous injection of $5 \times 10^5$ cancer cells, suspended in 100 µl of PBS, at the right flank of the mouse. B16.F10-Luc inoculations were performed by intravenous injection of $1 \times 10^6$ cancer cells in 100 µl of PBS. EMT6 inoculations were orthotopic, and $5 \times 10^5$ cancer cells were injected into the left-side fourth MFP in 100 µl PBS. When the volume of subcutaneous B16.F10 or orthotopic EMT6 tumours reached ~75–100 mm$^3$, mice were treated by intravenous injection of nanobodies or free diABZI compound 3 (SelleckChem) using 40% PEG400 (Sigma) as an excipient, or intraperitoneal injection of commercial anti-PD-L1 IgG (Clone BE0101, Bio X Cell) (100 µl per injection). For mice inoculated with B16.F10-Luc cells intravenously, mice were treated 7 days following tumour inoculation and euthanized on day 17 for evaluation of lung metastatic burden. Mice were treated with diABZI or nanobodies intravenously, and anti-PD-L1 IgG was administered intraperitoneally. For MDSC inhibition studies with CXCL1/2 inhibitor SX-682 inhibitor, mice were fed SX-682 formulated chow (provided by A. Richmond) or control chow at day 4 after tumour inoculation and through the course of the study. For studies evaluating the effects of MDSC, NK cell, CD4$^+$ T cell and CD8$^+$ T-cell depletion, mice with either EMT6 or B16.F10 tumours were intraperitoneally administered anti-Ly6G/Gr-1 (RB6-8C5; 200 µg), anti-asialo-GM1 (Poly21460; 100 µg for EMT6), anti-NK1.1 (PK136; 200 µg for B16.F10), anti-CD4$^+$ (YTS-191; 200 µg) or anti-CD8$^+$ (2.43; 200 µg) antibodies 1 day before each treatment with nanobody conjugates. Tumour volume calculations were calculated using $V_{tumour} = L \times W^2 \times 0.5$, in which $V_{tumour}$ is tumour volume, $L$ is tumour length and $W$ is tumour width. Tumour volume, total mouse mass and mouse well-being were recorded for the duration of the study. The end point for maximum tumour volume was 1,500 mm$^3$.

### Evaluation of nanobodies in spontaneous breast cancer model
A cohort of female *FVB/N-Tg (MMTV-PyVT)*$^{634Mul}$ mice, bred inhouse, was used for these studies. Study animals were weighed and mammary glands palpated twice weekly starting at 6 weeks of age. Tumour diameters in two dimensions were obtained using callipers. Treatment for the full cohort was initiated when first palpable tumours appeared at approximately 8–10 weeks of age. The mice received the nanobody–STING agonist conjugate or vehicle control for a total of 3 treatments at 7 day intervals. The study was terminated 22 days after the first treatment. At necropsy, tumours were removed, and a wet weight of all excised tumours was obtained for each mouse. One tumour was fixed in 10% buffered formalin for histological analysis. Lung metastases were assessed histologically as described previously[89].

### Adoptive OT-I T-cell transfer in B16.F10-OVA tumour model
Six- to eight-week-old OT-I mice (C57BL/6-Tg(TcraTcrb)1100Mjb/J) were purchased from Jackson Laboratory. Mice were euthanized, spleens were collected, and CD8$^+$ T cells isolated using an EasySep Mouse CD8$^+$ T Cell Isolation Kit (STEMCELL Technologies). T cells were activated in vitro in RPMI 1640 (Gibco) media supplemented with 10% HI-FBS (Gibco), 1% penicillin/streptomycin (Gibco), 50 µM β-mercaptoethanol (MilliporeSigma), 1 mM sodium pyruvate, minimum essential medium NEAA (non-essential amino acids) (Gibco), 10 mM HEPES (Gibco), recombinant mouse interleukin-2 (10 U ml$^{-1}$; MilliporeSigma) and Dynabeads Mouse T-Activator CD3/CD28 (at a bead-to-cell ratio of 1:1; Gibco) at 37 °C in a CO$_2$ (5%) incubator. After 5 days, T cells were magnetically separated from Dynabeads and allowed to rest in culture for 24 h before use. The following day, $5 \times 10^5$ OT-I CD8$^+$ T cells were adoptively transferred by retro-orbital injection to mice with B16.F10-OVA tumours.

### Western blot analysis
Mice were euthanized, tumours (EMT6 and B16.F10) were collected and 500 µl RIPA buffer (Sigma) supplemented with protease inhibitors (Sigma) was added in approximately 10 mg of tissue. Tissue was homogenized using a bead mill tissue homogenizer (TissueLyser II; Qiagen) and kept on ice for 30 min. Protein concentration was measured using a BCA protein assay kit (Thermo Scientific). Equal amount of protein (30 µg) was subjected to SDS−PAGE and transferred onto nitrocellulose membranes using the semi-dry transfer protocol (Bio-Rad). After transfer, membranes were probed with anti-SPARC and anti-β-actin antibodies overnight at 4 °C. Following incubation, the membranes were probed with HRP-conjugated secondary antibodies. All antibodies were purchased from Cell Signaling. Protein bands were visualized using ECL western blotting substrate (Thermo Scientific). Images of immunoblots were obtained using a LI-COR Odyssey Imaging System. Antibodies used for western blots are summarized in Supplementary Table 2.

### Immunofluorescence analysis of EMT6 tumours
Paraffin-embedded tissue sections (5 µm) were prepared for immunofluorescence and stained with anti-CD31 (Cell Signaling 77699; 1:500), anti-SPARC (Cell Signaling 5420, 1:500) and anti-CD45 (Cell Signaling 70257; 1:500). Tissue slides were deparaffined in xylene and rehydrated in serial ethanol dilutions. Antigen retrieval was performed by heating slides for 17 min in Tris EDTA buffer, pH 9 in a pressure cooker at 110 °C. Slides were cooled to room temperature and then blocked with 2.5% horse serum (vector labs). After blocking, slides were incubated overnight at 4 °C with primary antibody in horse serum. Slides were then incubated in anti-rabbit HRP secondary (Vector Labs) for 1 h at room temperature the following day and subsequently incubated in 1:500 Opal 520 (green) or Opal 570 (red) (Akoya) for 10 min. For serial staining, slides were stripped using citric acid buffer, pH 6.1 in a pressure cooker at 110 °C for 2 min, and then staining was repeated using different antibody and opal fluorophore. After the last Opal staining, slides were mounted using antifade gold mount with DAPI (Invitrogen). Stained images were acquired using a Keyence digital microscope system. Images were analysed with Fiji software (version 2.9.0). Quantification of markers was done by measuring total amount of fluorescence divided by total number of cells (DAPI).

### Flow cytometric experiments and analysis
EMT6 tumour-bearing Balb/c and B16.F10-OVA-bearing C57BL/6 mice were euthanized either 24 h or 48 h after treatment as indicated. Spleens and tumours were collected, weighed and placed on ice. Tumours were digested in RPMI 1640 media using a tumour dissociation kit (Miltenyi Biotech) or 500 µg ml$^{-1}$ collagenase III (Worthington) and 125 µg ml$^{-1}$ deoxyribonuclease I. Tumours were further dissociated using an OctoMACS separator (Miltenyi Biotech) and incubated for 30 min at 37 °C for complete digestion. Then subsequently tumours and spleens were mashed and separated into single-cell suspensions using a 70 µm cell strainer (Fisherbrand; Thermo Fisher Scientific), and red blood cells were lysed twice using ACK lysis buffer (Gibco). Cells were resuspended in flow buffer (1× PBS supplemented with 2% FBS and 50 µM dasatinib), counted and stained with Fc-block (anti-CD16/32, 2.4G2, Tonbo) for 15 min at 4 °C, and then stained with the appropriate antibodies for 1 h at 4 °C; antibodies used for flow cytometry staining are summarized in Supplementary Tables 3–5. After staining, cells were then washed again with FACS buffer, fixed with 2% paraformaldehyde for 10 min, washed again with FACS buffer containing AccuCheck counting beads, and analysed on a Cytek Aurora multispectral flow cytometer. All flow cytometry data were analysed using FlowJo software (version 10; Tree Star; https://www.flowjo.com/solutions/flowjo). Representative flow cytometry plots and gating schemes are shown in Supplementary Figs. 35–42.

### Pharmacokinetics and ex vivo imaging experiments
Healthy (Balb/c or C57BL/6) and EMT6 tumour-bearing (Balb/c) mice were injected intravenously with 100 µl of Cy5 (either as free dye or as a nanobody−Cy5 conjugate) at a dose of 2 mg kg$^{-1}$ in PBS. For

pre-treatment with the diABZI conjugated nanobody, a dose was prepared at 1.25 µg diABZI in 100 µl total and injected 3 days before Cy5 dosing. Blood draws were taken using heparinized capillary tubes (DWK Life Sciences) at discrete time points up to 5 days after injection. About 1 µl of blood was mixed with 50 µl of PBS and centrifuged, and the diluted plasma was collected for analysis. The concentration of Cy5 was determined using fluorescence spectroscopy (BioTek Synergy H1) using an excitation wavelength of 645 nm and an emission wavelength of 675 nm. Pharmacokinetic analysis was performed in GraphPad Prism (V10) using either a one-phase decay or two-phase decay, in which the reported half-life is the second phase (elimination). Biodistribution studies were performed by excising and weighing hearts, lungs, livers, spleens, kidneys and tumours. Tissues were washed in 1× PBS and transferred to the stage of the IVIS Lumina III (PerkinElmer). After IVIS, tissues were mechanically homogenized (TissueLyser II; Qiagen) in a volume of 200 µl 1× PBS. Homogenized tissues were centrifuged, and the supernatant containing the Cy5 dye was analysed by fluorescence spectroscopy (BioTek Synergy H1) to determine Cy5 concentration. A standard curve was generated of free DBOC–Cy5 dye in 1× PBS, and concentrations of Cy5 in tissue were calculated by fitting the standard curve to a linear regression. Fluorescence (radiant efficiency) was measured with a maximum value of $1.56 \times 10^{10}$ and a minimum of $8.21 \times 10^{8}$, and areas were drawn manually for organs to generate average radiant efficiency values (per $cm^2$) using the Living Image software (version 4.5). For luminescence imaging of lungs excised from mice intravenously inoculated with B16.F10-Luc cells, lungs were placed in black 12-well plates (Cellvis) and incubated for 5 min in a solution of 1 mg ml$^{-1}$ Pierce D-luciferin, monopotassium salt (Thermo Fisher Scientific) in 1× PBS. Images were taken on the IVIS Lumina III, and luminescence was quantified as total radiant flux (photons per second (p s$^{-1}$)) for each set of lungs.

### Serum analysis for anti-VHH antibodies

Mice were pre-treated with PBS or nAlb–diABZI (1.25 µg dose of diABZI) three times every 3 days or treated once with nAlb–diABZI. Fourteen days after the first dose, blood was collected by cardiac puncture in microfuge tubes and allowed to clot to extract serum. Tubes were centrifuged at 2,000 × g for 15 min at 4 °C, and the serum was then collected and diluted directly in PBS (1:4 to 1:8,192) for analysis. MonoRab rabbit anti-camelid VHH antibody plates (GenScript) were used to determine anti-VHH antibodies in mouse serum. About 3 µg in 100 µl of anti-albumin nanobody was loaded into each well of the 96-well plate and allowed to incubate in the pre-coated antibody plate, sealed and incubated at 37 °C for 30 min. The plate was washed with 200 µl of PBST four times. Either the diluted mouse serum or a commercial Rabbit anti-Camelid VHH antibody (GenScript; A01860) was added in serial dilutions to the wells of the plate at a volume of 100 µl. The plate was sealed and incubated at 37 °C for 30 min, followed by washing four times with 200 µl of PBST. A commercial secondary Goat anti-Mouse IgG–FITC conjugate (Invitrogen; 31547) or secondary anti-Rabbit IgG–FITC conjugate (Sigma; F9887) was added to the mouse serum or commercial anti-VHH, respectively, at 100 µl and incubated for 30 min at 37 °C before washing with 200 µl PBST four times. The fluorescence intensity per well was determined using a fluorescent plate reader (extinction, 495 nm; emission, 515 nm) and the concentration of anti-VHH antibody in serum determined based on the standard curve.

### Ex vivo plasma analyte analysis

Blood was collected by either cheek bleed or cardiac puncture in K$_2$EDTA-coated tubes (BD Biosciences). Tubes were centrifuged at 2,000 × g for 15 min at 4 °C, and the plasma was collected for analysis. Cytokine levels were evaluated using either the LEGENDplex Mouse Anti-Virus Response Panel (BioLegend) or the LEGENDplex Mouse Cytokine Panel 2 (BioLegend), both with V-bottom plates, according to manufacturer's instructions, and data were collected using flow cytometry (CellStream, Cytek Biosciences). Cytokine concentrations were interpolated from standard curves using asymmetric sigmoidal five-parameter logistic curve fits (GraphPad Prism V10). Bar plots comparing groups and heat maps of averaged values for groups were generated to analyse results.

### NanoString nCounter analysis of EMT6 tumours

After administration of three treatments of nAlb–diABZI, AP–diABZI or PBS to EMT6 tumour-bearing female Balb/c mice, tumours were isolated and digested, and 100 ng of RNA was isolated, as described above for qPCR analysis. RNA was hybridized to the IO360 PanCancer panel, as well as through a selected gene panel, of target-specific fluorescent barcodes and analysed using NanoString nCounter MAX Analysis system. The fold change for genes within groups was calculated by comparing against the average normalized gene expression values within PBS-treated mice. All statistical significance and clustering analysis was performed in R (http://cran.r-project.org) based on the genes provided in the IO360 PanCancer panel.

### Safety statement

All research performed in this study was done so with careful consideration of any risks that are inherent to the materials, instruments and experiments performed. All research safety guidelines and considerations as provided by the safety data sheets and university guides were adhered to for the duration of this study.

### Statistics

All data were plotted, and statistical analysis was performed using Prism 10 (GraphPad) software. Unless indicated in the figure captions, all data are presented as mean ± s.e.m. For comparisons between two groups, unpaired two-tailed Student's t-tests were performed unless otherwise indicated. For multiple comparisons, analysis of variance (ANOVA) was performed with post hoc as indicated in the figure captions. For tumour volume, statistical significance was examined through a two-way ANOVA followed by Tukey's adjustment for multiple comparisons unless otherwise indicated. A log-rank (Mantel–Cox) test was used to compare Kaplan–Meier survival data. The robust regression and outlier removal method was used to identify outliers which were removed.

### Ethics statement

Studies involving the use of animals were completed under Animal Care Protocols approved by the Vanderbilt University Animal Care and Use Committee. The health assessment of animals was completed using a standard operating procedure also approved by the Vanderbilt University Animal Care and Use Committee.

### Reporting summary

Further information on research design is available in the Nature Portfolio Reporting Summary linked to this article.

## Data availability

The main data supporting the results in this study are available within the Article and its Supplementary Information. All data generated in this study, including source data for all figures, are available via figshare at https://doi.org/10.6084/m9.figshare.c.7825349.v1 (ref. 90).

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

## Acknowledgements

We thank R. D'arcy and N. Francini for intellectual discussions, C. Duvall for the use of IVIS Imaging System, I. Georgiev for the use of protein purification system, J. Brunger for the use of molecular biology research instrumentation and M. Reyzer for assistance with MALDI. We thank S. Joyce for providing the phycoerythrin-labelled SIINFEKL/H-2Kb tetramer, A. Lund for providing the OVA expressing B16 cell line and M. Munson for the NCI-H358 Gal9-mCherry cells. We thank the core facilities of the Vanderbilt University Medical Center Flow Cytometry Shared Resource, supported by the Vanderbilt Digestive Disease Research Center (DK058404) and the Vanderbilt Ingram Cancer Center (VICC) (P30 CA68485), the Vanderbilt University Small Molecule NMR Facility, the Translational Pathology Shared Resource supported by NCI/NIH Cancer Center Support Grant P30CA068485, the Vanderbilt Center for Structural Biology, the Vanderbilt Mass Spectroscopy Research Center, the Vanderbilt Center for Innovative Technology and the Vanderbilt Cell Imaging Shared Resource, supported by NIH grants CA68485, DK20593, DK58404, DK59637, EY08126 and S10 OD021630. This research was

supported by grants from the Susan G. Komen (CCR19609205 to J.T.W.), the National Institutes of Health (R01 CA245134, R01 CA266767 and R01 CA274675 to J.T.W.; R01 CA11601 to A.R.; K00 CA253718 to E.N.A.; R01 CA217987 to J.C.R; NCI SPORE 2P50CA098131–17 to J.M.B.; F32 CA288044 to A.J.K.), the National Science Foundation (CBET-1554623 to J.T.W.), a VICC Ambassador Discovery Grant (J.T.W.), VICC Support Grant P30 CA68485, a Department of Defense Era of Hope Scholar Award (BC170037 to J.M.B.), the Department of Veterans Affairs (101BX002301 and IK6B005225 to A.R.) and funds provided by the Vanderbilt University School of Engineering (J.T.W.). B.R.K. acknowledges postdoctoral funding support from the PhRMA Foundation Postdoctoral Fellowship in Drug Delivery. A.J.K. and B.C.T. were supported by the NIH Microenvironmental Influences in Cancer Training Grant (T32CA009592), H.M.P. was supported by the NIH Integrated Training in Engineering and Diabetes Training Grant (T32DK101003), P.T.S. was supported by the NIH Chemical-Biology Interface Training Grant (T32GM065086), and N.C.C. was supported by the NIH Medical Scientist Training Program (T32GM007347). T.L.S., L.E.P., H.M.P. and P.T.S. acknowledge funding support through the National Science Foundation Graduate Research Fellowship Program under grant number 193793. L.A.H. acknowledges the Vanderbilt Institute of Nanoscale Sciences and Engineering Research Experience for Undergraduates supported by the National Science Foundation (NSF-DMR 1852157). Any opinions, findings and conclusions or recommendations expressed in this material are those of the author(s) and do not necessarily reflect the views of the National Science Foundation. J.E.F. acknowledges support from the Vanderbilt University School of Engineering Summer Research Program.

## Author contributions

B.R.K. and J.T.W. conceptualized the study. B.R.K. designed experiments and conducted most of the experiments and analyses. B.R.K., K.A., N.C.C. and J.E.F. synthesized materials for the studies. E.N.A. performed immunofluorescence staining and analysis. V.B. conducted immunophenotyping analysis of lymphocytes. V.B. and A.J.K. performed flow cytometric analysis of cell populations. A.H., B.C.T. and J.M.B. conducted NanoString studies and analysis. B.R.K. and J.C.M. performed ESI–MS studies. B.R.K., N.C.C. and J.E.F. prepared proteins and synthesized conjugates. T.L.S., L.E.P., J.Y., H.M.P., P.T.S., E.H.-W. and L.A.H. performed some experiments and provided technical assistance. K.N.G.-C. sectioned, stained and analysed tissue for histology. A.R., B.M.F., J.A.M., W.K.R., J.C.R., J.M.B. and J.T.W. provided intellectual contributions to develop the project. B.R.K. and J.T.W. wrote and edited the paper, analysed data and generated figures. B.R.K. and J.T.W. supervised the project and research design and acquired funding to support the research.

## Competing interests

J.T.W., K.A. and B.R.K. are inventors on United States Patent Application PCT/US2023/079884 'NANOBODY-DRUG CONJUGATES AND METHODS OF PREPARING THEREOF' which describes nanobody conjugation and delivery technologies. J.T.W. has received research support from Incyte Corporation within the past 3 years. J.C.R. is an employee of Vanderbilt University Medical Center and appointed to the Vanderbilt University School of Medicine. He is a scientific advisory board member of Sitryx Therapeutics. J.C.M. and J.A.M. received support from Agilent Technologies in the form of a Thought Leader Award. Agilent is a commercial manufacturer of the MS instrumentation used in aspects of this work. J.A.M. is a member of the Scientific Advisory Board for MOBILion Systems, which is a manufacturer of high-resolution ion mobility–MS instrumentation. J.A.M. certifies that his contributions are scientifically objective and not influenced by his Scientific Advisory Board participation. J.M.B. receives research support from Genentech/Roche and Incyte Corporation, has received advisory board payments from AstraZeneca, Eli Lilly and Mallinckrodt and is an inventor on patents regarding immunotherapy targets and biomarkers in cancer.

## Additional information

**Extended data** is available for this paper at https://doi.org/10.1038/s41551-025-01400-0.

**Correspondence and requests for materials** should be addressed to John T. Wilson.

[1]Department of Chemical and Biomolecular Engineering, Vanderbilt University, Nashville, TN, USA. [2]Department of Chemical and Biomolecular Engineering, The Ohio State University, Columbus, OH, USA. [3]Department of Biomedical Engineering, Vanderbilt University, Nashville, TN, USA. [4]Department of Medicine, Vanderbilt University Medical Center, Nashville, TN, USA. [5]Department of Pharmacology, Vanderbilt University Medical Center, Nashville, TN, USA. [6]Division of Comparative Medicine, Department of Pathology, Microbiology, and Immunology, Vanderbilt University Medical Center, Nashville, TN, USA. [7]Department of Chemistry, Vanderbilt University, Nashville, TN, USA. [8]Vanderbilt Center for Innovative Technology, Vanderbilt University, Nashville, TN, USA. [9]Vanderbilt Ingram Cancer Center, Vanderbilt University Medical Center, Nashville, TN, USA. [10]Department of Veterans Affairs, Tennessee Valley Healthcare System, Nashville, TN, USA. [11]Department of Pathology, Microbiology, and Immunology, Vanderbilt University Medical Center, Nashville, TN, USA. [12]Vanderbilt Center for Immunobiology, Vanderbilt University Medical Center, Nashville, TN, USA. [13]Vanderbilt Institute for Infection, Immunology and Inflammation, Vanderbilt University Medical Center, Nashville, TN, USA. [14]Vanderbilt Institute of Chemical Biology, Vanderbilt University, Nashville, TN, USA. ✉e-mail: john.t.wilson@vanderbilt.edu

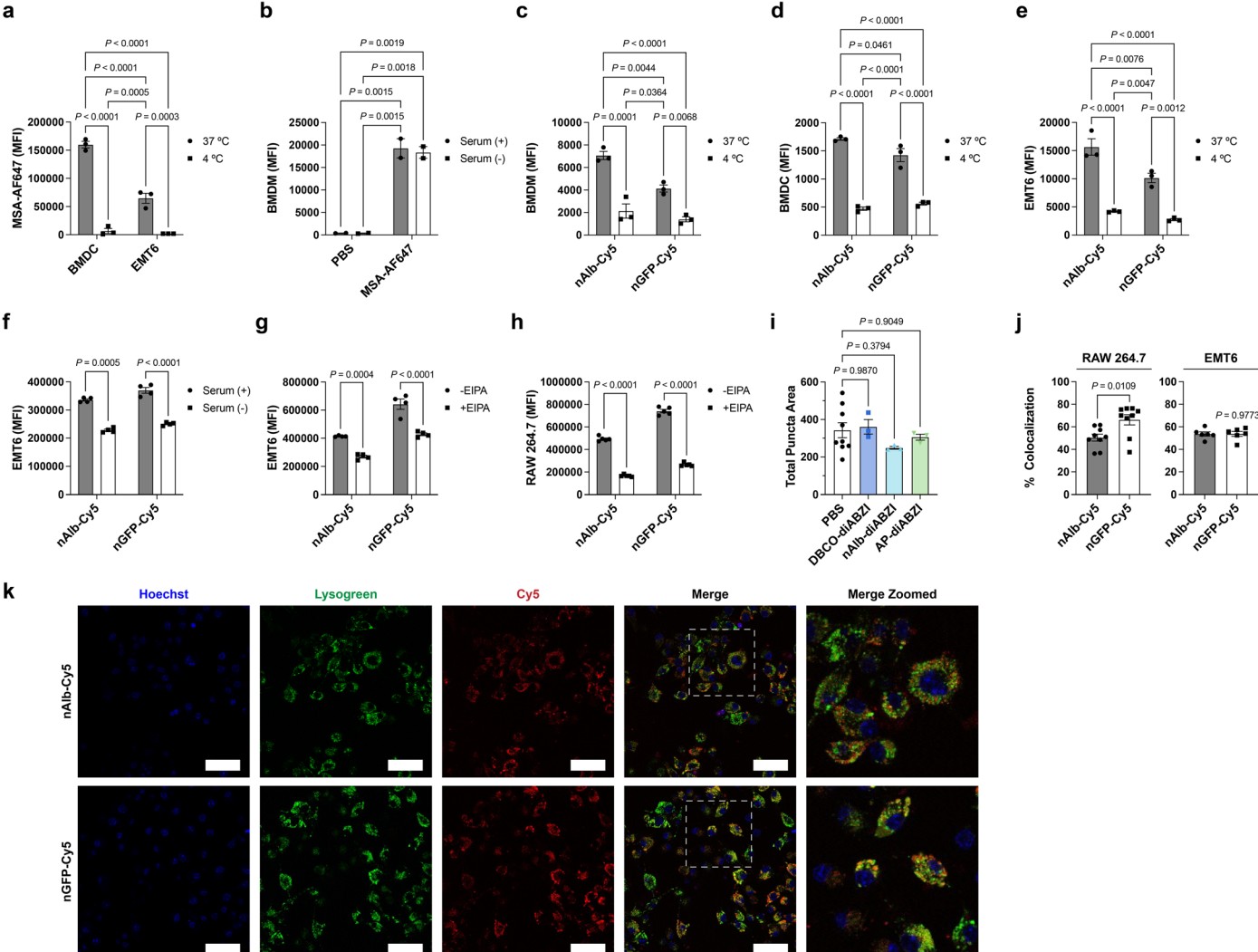

**Extended Data Fig. 1 | *In vitro* analysis of nanobody internalization.**
**(a)** Median fluorescence intensity (MFI) of BMDMs treated with AlexaFluor647-labeled mouse serum albumin (MSA-AF647) at 1 μM or PBS in serum-containing ( + serum) or serum -deficient (-serum) media as measured by flow cytometry (n = 2). *P* values determined by ANOVA with post-hoc Tukey's correction for multiple comparisons. **(b)** MFI of BMDM cells treated with 2 μM MSA-AF647 in serum containing ( + serum) or serum deficient (-serum) media as measured by flow cytometry (n = 2). *P* values determined by ANOVA with Šídák's multiple comparison test. MFI of **(c)** BMDM cells, **(d)** BMDC cells, and **(e)** EMT6 cells treated with 1 μM nAlb-Cy5 or nGFP-Cy5 at 37 °C and 4 °C as measured by flow cytometry (n = 3). *P* values determined by two-tailed Student's t-test. **(f)** MFI of EMT6 cells treated with 2 μM nAlb-Cy5 or nGFP-Cy5 in serum containing ( + serum) or serum deficient (-serum) media as measured by flow cytometry

(n = 4). P values determined by ANOVA with Šídák's multiple comparison test. **(g-h)** MFI of **(g)** EMT6, and **(h)** RAW 264.7 cells treated with nAlb-Cy5 (2 μM) with (+EIPA) or without (-EIPA) the macropinocytosis inhibitor EIPA as measured by flow cytometry (n = 3). *P* values determined by two-tailed Student's t-test. **(i)** Integrated pixel intensity of Gal9-mCherry puncta per cell for cells treated DBCO-PEG₁₁-diABZI (DBCO-diABZI), nAlb-diABZI, and AP-diABZI at 0.25 μM (n = 9 for PBS; n = 3 for all other groups). *P* values determined via ANOVA with Dunnett's multiple comparisons test for all groups vs. PBS; ns: not-significant (*P* > 0.05). **(j-k)** Colocalization analysis **(j)** of Cy5 and LysoTracker in RAW264.7 and EMT6 cells, and **(k)** fluorescent micrographs of RAW 264.7 cells treated with 2 μM nAlb-Cy5 or nGFP-Cy5 for analysis of colocalization of Cy5 (red) with lysosomes (LysoTracker Green; green); nuclei are stained with Hoechst (blue) (scale bar: 100 μm). Replicates are biological, and data are shown as mean ± SEM.

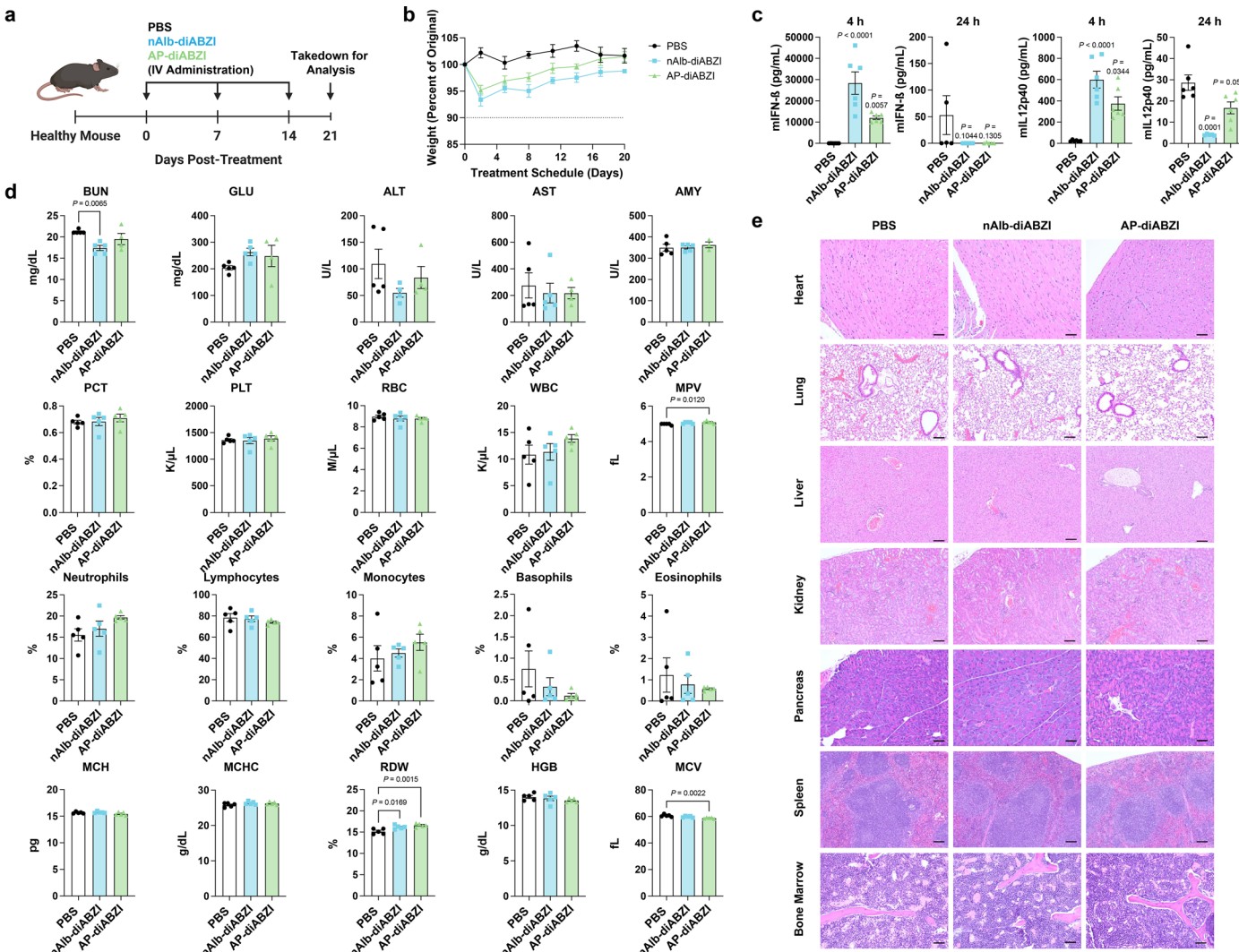

**Extended Data Fig. 2 | Evaluation of nAlb-diABZI and AP-diABZI toxicity. (a)**
Scheme for treating healthy C57BL/6 female mice with nAlb-diABZI or AP-diABZI
(1.25 μg diABZI) or PBS (vehicle). **(b)** Body weight change of mice in response to
indicated treatment (n = 3 for PBS; n = 5 for other groups). **(c)** Quantification of
serum cytokines 4 and 24 h after the first treatment (n = 6). *P* values determined
by one-way ANOVA with Dunnett's multiple comparisons test for each group
compared to PBS. **(d)** After 3 treatments, mice were euthanized and blood
samples were collected to determine changes in red blood cells (RBCs), white
blood cells (WBCs), neutrophils, platelets, and lymphocytes. Serum samples

were also used to analyze liver and kidney function by measuring changes in
alanine aminotransferase (ALT), aspartate transferase (AST), blood urea nitrogen
(BUN), and creatinine (n = 4 for AP-diABZI; n = 5 for other groups). *P* values
determined by ANOVA with Dunnett's multiple comparison test each group
compared to PBS. **(e)** Representative microscopy images of H&E stained tissue
sections from healthy C57BL/6 mice treated as indicated (scale bar: 100 μm for
heart, lung, liver, kidney, and spleen; 50 μm for pancreas and bone marrow).
Replicates are biological, and data are shown as mean ± SEM. Panel **a** created with
BioRender.com.

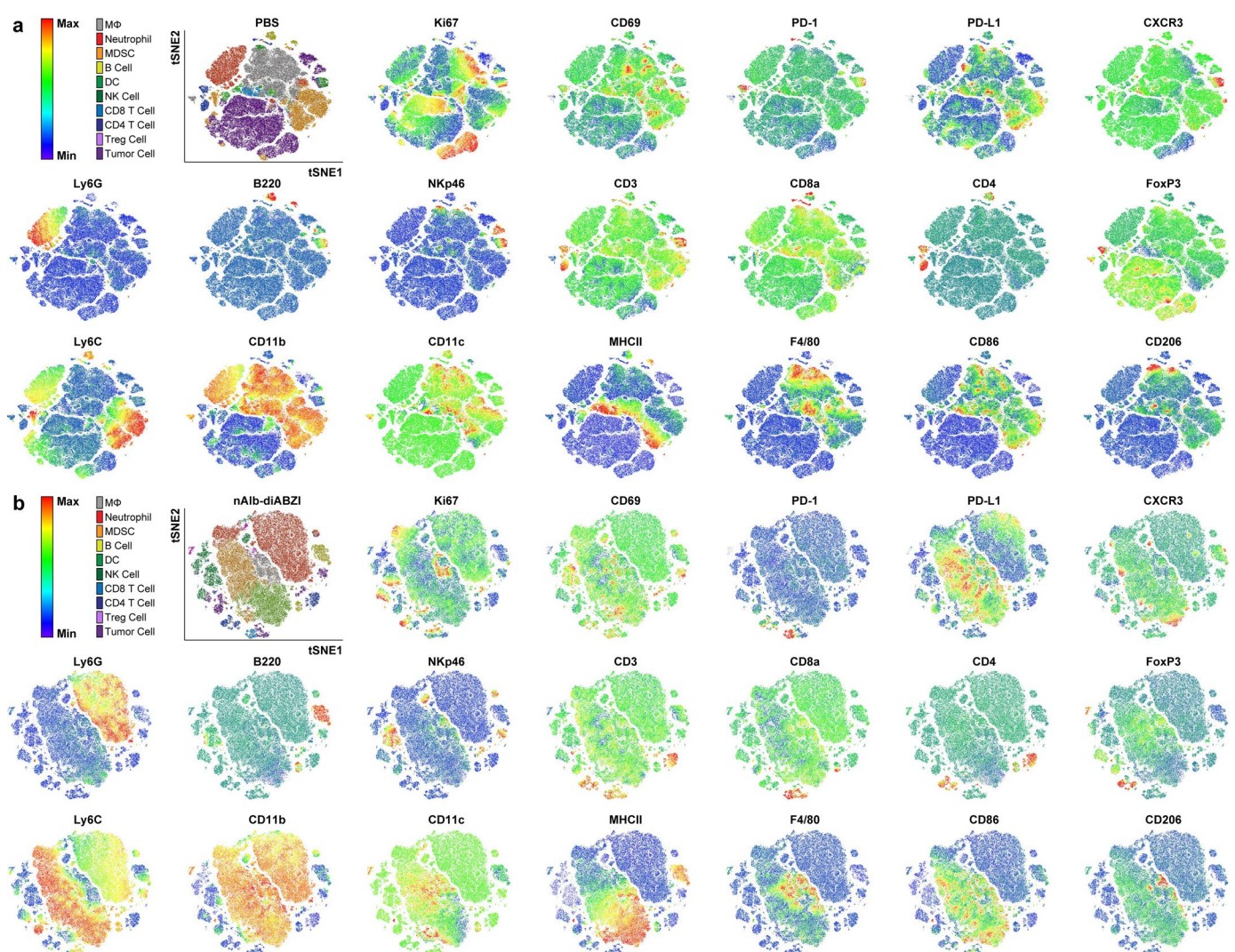

**Extended Data Fig. 3 | Flow cytometric immunophenotyping of EMT6 tumors following nAlb-diABZI treatment.** tSNE plots of live cells in EMT6 tumors after three doses of **(a)** PBS or **(b)** nAlb-diABZI, colored by cell population with relative expression levels. DC: dendritic cell; Mφ: macrophage; NK: natural killer cell; MDSC: myeloid-derived suppressor cells.

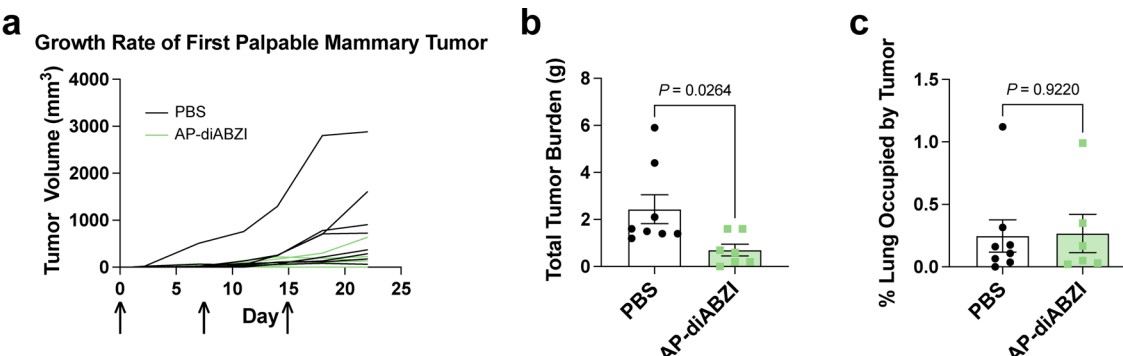

**Extended Data Fig. 4 | Evaluation of AP-diABZI in a spontaneous breast cancer model.** Female *FVB/N-Tg (MMTV-PyVT)^634Mul* mice with breast tumors were treated with AP-diABZI or PBS (vehicle) once a week for 3 weeks starting at approximately 8–10 weeks of age. **(a)** Growth rate of first palpable mammary tumor during treatment until the study was terminated on day 22 (n = 8 for PBS; n = 7 for AP-diABZI). At necropsy, all breast tumors were removed and weighed **(b)** (n = 8 for PBS; n = 7 for AP-diABZI; *P* value determined by two-tailed Mann-Whitney test) and histological analysis of lungs was performed to quantify lung metastasis **(c)** (n = 8 for PBS; n = 6 for AP-diABZI; *P* value determined by two-tailed Student's t-test). Replicates are biological, and data are shown as mean ± SEM.

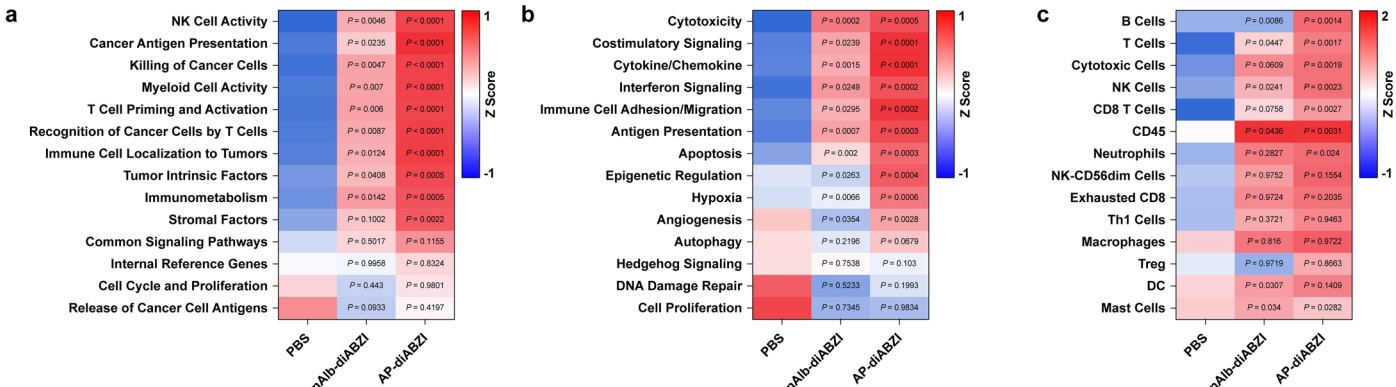

**Extended Data Fig. 5 | NanoString gene expression analysis of EMT6 tumors following treatment with nAlb-diABZI and AP-diABZI.** Annotated matrices for **(a)** functional gene annotations, **(b)** biological signatures, and **(c)** cell types from IO360 Pan Cancer NanoString gene expression panel comparing PBS, nAlb-diABZI, and AP-diABZI 24 h after three doses (n = 3 for nAlb-diABZI; n = 4 for all other groups). *P* values determined by one-way ANOVA with post-hoc Tukey's correction for multiple comparisons with comparison to PBS indicated. Replicates are biological.

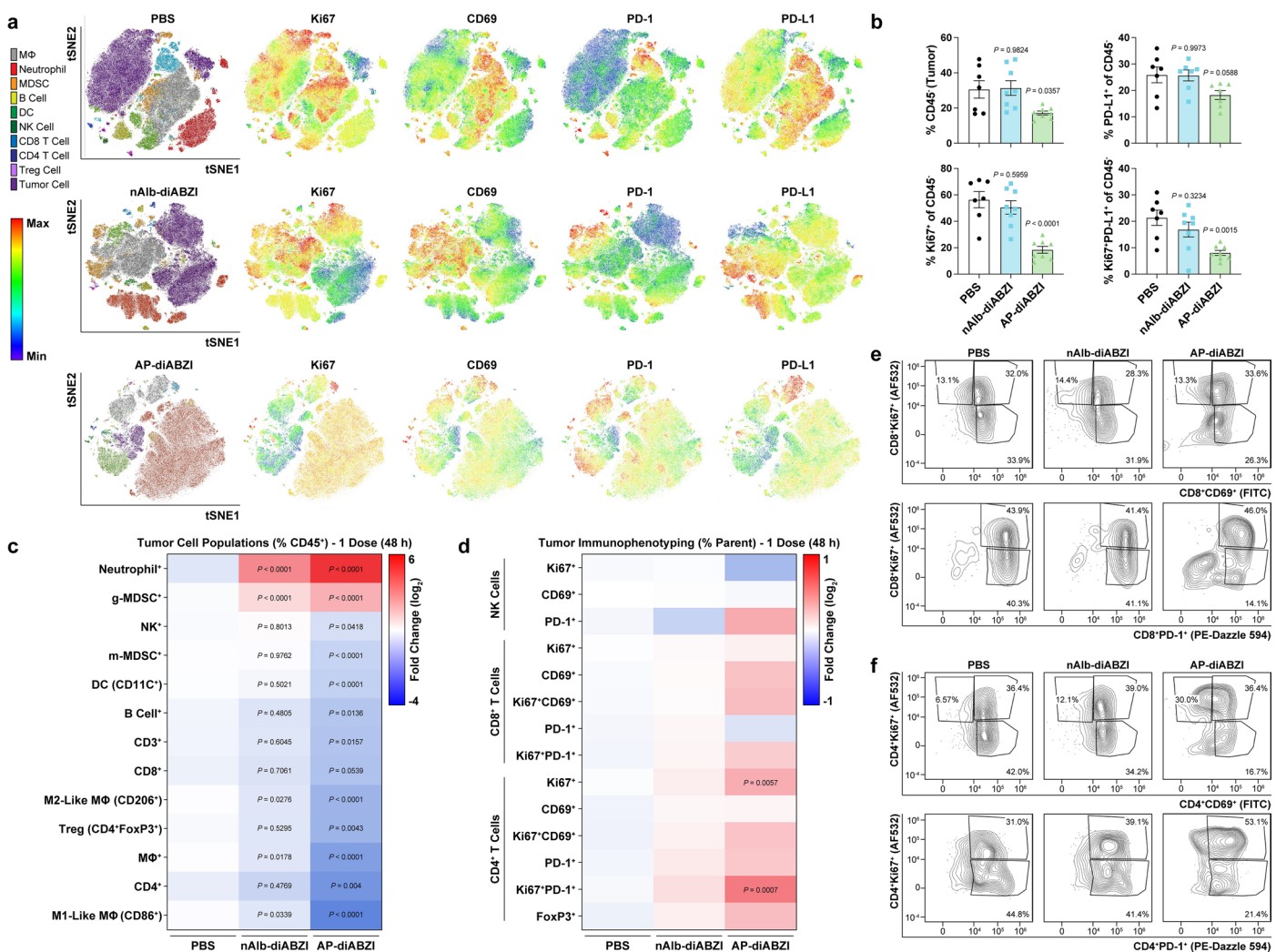

**Extended Data Fig. 6 | Flow cytometric analysis of EMT6 tumors following single dose of nAlb-diABZI or AP-diABZI.** EMT6 tumor bearing female Balb/c mice were treated with a single dose of nAlb-diABZI (n = 8), AP-diABZI (n = 8), or PBS (n = 7) and tumors isolated 48 h later for flow cytometric analysis. **(a)** tSNE plots of live cells in EMT6 tumors, colored by cell population with relative expression level of Ki67, CD69, PD-1, and PD-L1 as indicated on heat map. DC: dendritic cell; Mφ: macrophage; NK: natural killer cell; MDSC: myeloid-derived suppressor cell. **(b)** Analysis of the frequency of live CD45⁻ cells and frequency

of PD-L1⁺, Ki67⁺, and PD-L1⁺Ki67⁺ expressing CD45⁻ cells in the tumor. **(c-d)** Heat maps summarizing the fold change in the percentage of **(c)** indicated cell population and **(d)** frequency of NK cells, CD8⁺ T cells, and CD4⁺ T cells expressing the indicated marker in EMT6 tumors. **(b-d)** P values determined by one-way ANOVA with Dunnett's multiple comparisons test for all groups vs. PBS. **(e-f)** Representative flow cytometry dot plots characterizing the expression of CD69 and PD-1 on Ki67⁺CD4⁺ and Ki67⁺CD8⁺ T cells in EMT6 tumors. Replicates are biological, and data are shown as mean ± SEM.

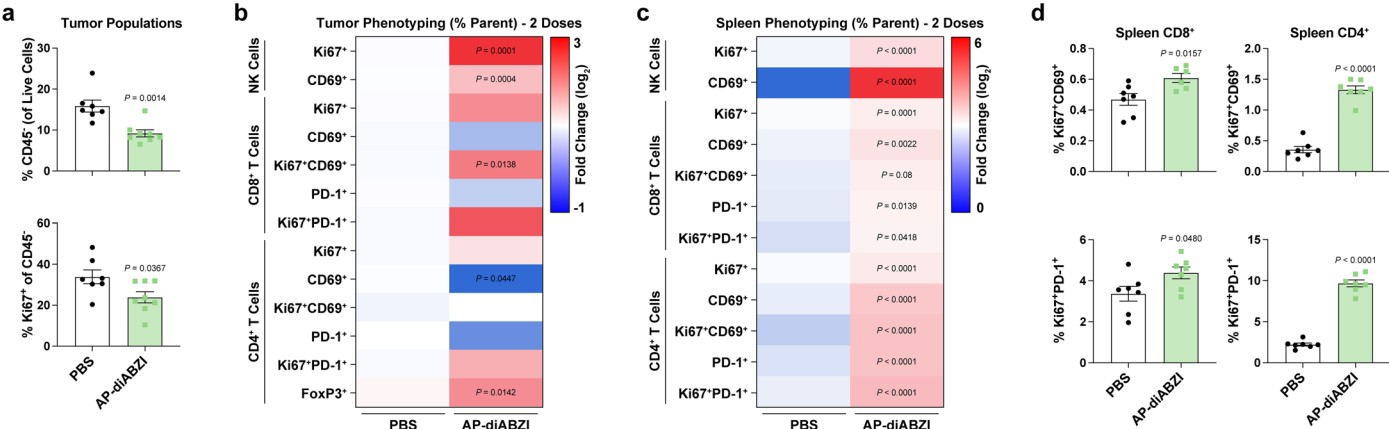

**Extended Data Fig. 7 | Flow cytometric analysis of EMT6 tumors following two doses of AP-diABZI.** EMT6 tumor bearing female Balb/c mice were treated with two intravenous doses of AP-diABZI (n = 8) or PBS (n = 7) and tumors isolated 24 h later for flow cytometric analysis of **(a)** frequency of live and Ki67+ breast cancer cells (CD45-), **(b)** frequency of NK cells, CD8+ T cells, and CD4+ T cells expressing the indicated markers in EMT6 tumors, and **(c)** frequency of NK cells, CD8+ T cells, and CD4+ T cells expressing the indicated markers in the spleen. **(d)** Frequency of Ki67+CD69+ and Ki67+PD-1+ CD8+ T cells and CD4+ T cells in the spleens of mice treated as indicated. *P* values determined by two-tailed Student's t-test. Replicates are biological, and data are shown as mean ± SEM.

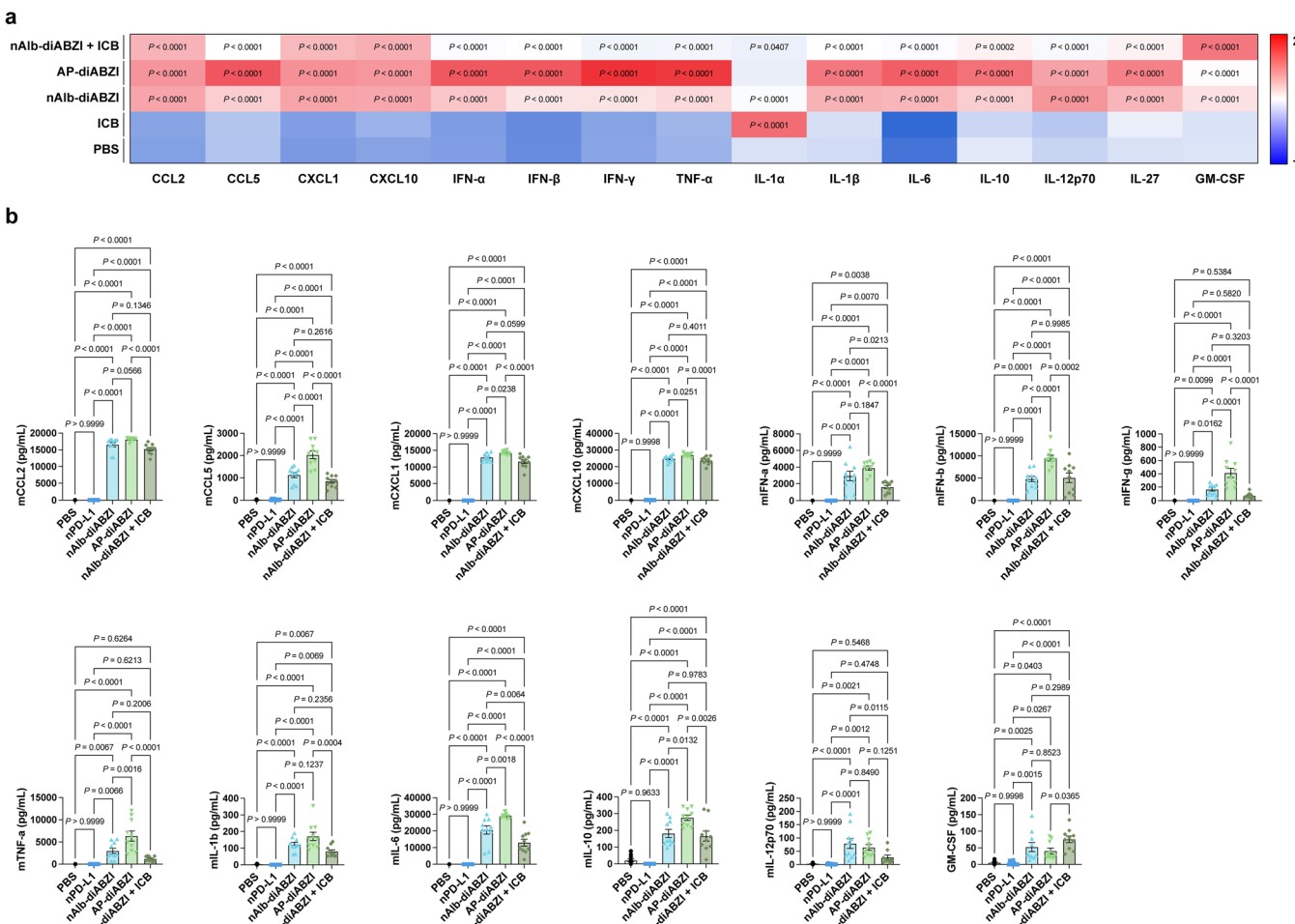

**Extended Data Fig. 8 | Evaluation of serum cytokines induced by nanobody-diABZI conjugates.** Serum cytokine concentration in B16.F10 tumor bearing C57BL/6 female mice 4 h after the first treatment represented as **(a)** heat maps and **(b)** bar plots (n = 10). ICB: Anti-PD-L1 IgG. *P* values determined by one-way ANOVA with **(a)** Dunnett's multiple comparison test for all groups vs. PBS and **(b)** post-hoc Tukey's correction for multiple comparisons. Replicates are biological, and data are shown as mean ± SEM.

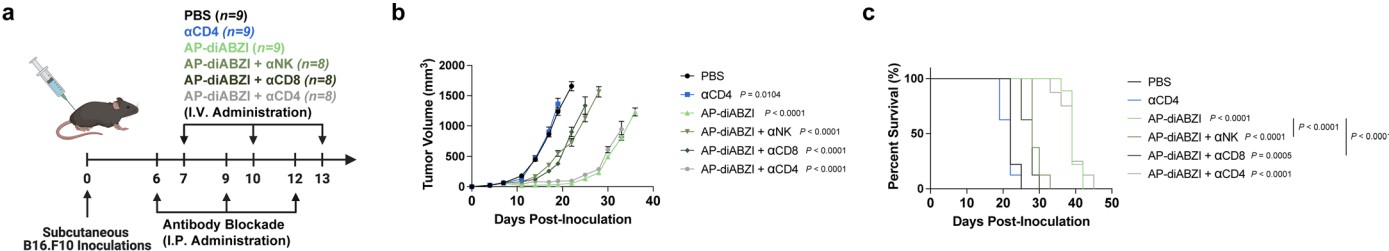

**Extended Data Fig. 9 | Effect of NK and T cell depletion on AP-diABZI efficacy in B16.F10 model. (a)** Schematic of B16.F10 tumor inoculation and treatment schedule with depletion antibodies (n = 9 for PBS, anti-CD4, and AP-diABZI; n = 8 for all other groups). Anti-NK1.1 IgG, anti-CD8 IgG, and anti-CD4 IgG were injected I.P. at 200 µg and AP-diABZI was injected I.V. at 1.25 µg of diABZI per injection. **(b)** Tumor growth curves and **(c)** Kaplan-Meier survival plots for mice with B16.F10 tumors treated as indicated. **(b)** P values determined by two-way ANOVA with post-hoc Tukey's correction for multiple comparisons; comparisons to PBS on day 19 are shown. **(c)** Endpoint criteria of 1500 mm³ tumor volume with P value determined by log-rank test for comparison to PBS group and for the comparisons indicated. Replicates are biological, and data are shown as mean ± SEM. Panel **a** created with BioRender.com.

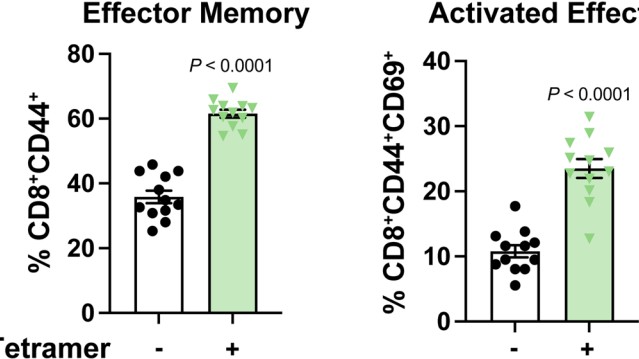

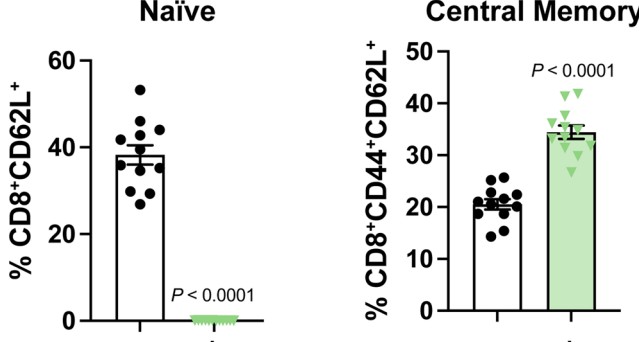

**Extended Data Fig. 10 | Characterization of T cell memory response to AP-diABZI in B16.F10-OVA model.** Quantification of flow cytometric analysis presented in Fig. 7m showing the distribution of memory of SIINFEKL-specific (SIINFEKL/H-2kB tetramer⁺) or non-SIINFEKL-specific (tetramer⁻) CD8⁺ T cells (n = 12). Represented bar plots include effector memory $T_{EM}$ (CD44⁺CD62L⁻),

activated CD69⁺ effector memory $T_{EM}$ (CD44⁺CD69⁺CD62L⁻), naïve (CD44⁻ CD62L⁺), and central memory $T_{CM}$ (CD44 + CD62L⁺). *P* values determined by by two-tailed Student's t-test. Replicates are biological, and data are shown as mean ± SEM.

# Reporting Summary

Please do not complete any field with "not applicable" or n/a.  Refer to the help text for what text to use if an item is not relevant to your study.
For final submission: please carefully check your responses for accuracy; you will not be able to make changes later.

## Statistics

For all statistical analyses, confirm that the following items are present in the figure legend, table legend, main text, or Methods section.

| n/a | Confirmed | |
|-----|-----------|---|
| ☐ | ☒ | The exact sample size (*n*) for each experimental group/condition, given as a discrete number and unit of measurement |
| ☐ | ☒ | A statement on whether measurements were taken from distinct samples or whether the same sample was measured repeatedly |
| ☐ | ☒ | The statistical test(s) used AND whether they are one- or two-sided *Only common tests should be described solely by name; describe more complex techniques in the Methods section.* |
| ☒ | ☐ | A description of all covariates tested |
| ☐ | ☒ | A description of any assumptions or corrections, such as tests of normality and adjustment for multiple comparisons |
| ☐ | ☒ | A full description of the statistical parameters including central tendency (e.g. means) or other basic estimates (e.g. regression coefficient) AND variation (e.g. standard deviation) or associated estimates of uncertainty (e.g. confidence intervals) |
| ☐ | ☒ | For null hypothesis testing, the test statistic (e.g. *F*, *t*, *r*) with confidence intervals, effect sizes, degrees of freedom and *P* value noted *Give P values as exact values whenever suitable.* |
| ☒ | ☐ | For Bayesian analysis, information on the choice of priors and Markov chain Monte Carlo settings |
| ☒ | ☐ | For hierarchical and complex designs, identification of the appropriate level for tests and full reporting of outcomes |
| ☒ | ☐ | Estimates of effect sizes (e.g. Cohen's *d*, Pearson's *r*), indicating how they were calculated |

*Our web collection on statistics for biologists contains articles on many of the points above.*

## Software and code

Policy information about availability of computer code

| Data collection | ROSIE (Version 3), Agilent MassHunter IM-MS Acquisition Data software, NanoAnalyze TA Instruments Affinity ITC instrument, FlexControl (Version 4.0, BioTek SynergyHT Version 3.4, Bio-Rad CFX manager Software Version 3.0. |
|---|---|
| Data analysis | FlowJo Version 10, GraphPad PRISM Version 10, R (Version 4.4.1), PyMOL (Version 2.5.1), FlexAnalysis (Version 4.0), Mnova 14.1.1, FIJI Version 2.9. |

For manuscripts utilizing custom algorithms or software that are central to the research but not yet described in published literature, software must be made available to editors and reviewers. We strongly encourage code deposition in a community repository (e.g. GitHub). See the Nature Portfolio guidelines for submitting code & software for further information.

## Data

Policy information about availability of data

All manuscripts must include a data availability statement. This statement should provide the following information, where applicable:
- Accession codes, unique identifiers, or web links for publicly available datasets
- A description of any restrictions on data availability
- For clinical datasets or third party data, please ensure that the statement adheres to our policy

All materials used or generated in this study are available to researchers following appropriate standard material transfer agreements. Source Data are available within FigShare at the following DOI; 10.6084/m9.figshare.27098866. Modeled PDB structures used are 4N0F and 1AO6.

## Research involving human participants, their data, or biological material

Policy information about studies with <u>human participants or human data</u>. See also policy information about <u>sex, gender (identity/presentation), and sexual orientation</u> and <u>race, ethnicity and racism</u>.

| | |
|---|---|
| Reporting on sex and gender | No human participants or biological material was used. |
| Reporting on race, ethnicity, or other socially relevant groupings | No human participants or biological material was used. |
| Population characteristics | No human participants or biological material was used. |
| Recruitment | No human participants or biological material was used. |
| Ethics oversight | No human participants or biological material was used. |

Note that full information on the approval of the study protocol must also be provided in the manuscript.

# Field-specific reporting

Please select the one below that is the best fit for your research. If you are not sure, read the appropriate sections before making your selection.

[X] Life sciences    [ ] Behavioural & social sciences    [ ] Ecological, evolutionary & environmental sciences

For a reference copy of the document with all sections, see nature.com/documents/nr-reporting-summary-flat.pdf

# Life sciences study design

All studies must disclose on these points even when the disclosure is negative.

| | |
|---|---|
| Sample size | Sample sizes were determined by pilot experiments and through previous experiments in order to obtain statistically significant representative data. |
| Data exclusions | Data were only excluded by outlier analysis using GraphPad PRISM Version 10, as deemed appropriate. Mice were not excluded from studies. |
| Replication | All studies in the paper were repeated at least two times. No reported study failed upon repeat. |
| Randomization | All samples and organisms were randomly allocated into experimental groups. |
| Blinding | Blinding was not implemented formally in the study due to constraints in resources and personnel. The investigator responsible for organizing the experimental groups and handling sample collection was aware of the group allocations; however, colleagues who assisted with data collection were blinded to these details. |

# Behavioural & social sciences study design

All studies must disclose on these points even when the disclosure is negative.

| | |
|---|---|
| Study description | |
| Research sample | |
| Sampling strategy | |
| Data collection | |
| Timing | |
| Data exclusions | |
| Non-participation | |
| Randomization | |

# Ecological, evolutionary & environmental sciences study design

All studies must disclose on these points even when the disclosure is negative.

| | |
|---|---|
| Study description | |
| Research sample | |
| Sampling strategy | |
| Data collection | |
| Timing and spatial scale | |
| Data exclusions | |
| Reproducibility | |
| Randomization | |
| Blinding | |

Did the study involve field work?   ☐ Yes   ☐ No

## Field work, collection and transport

| | |
|---|---|
| Field conditions | |
| Location | |
| Access & import/export | |
| Disturbance | |

# Reporting for specific materials, systems and methods

We require information from authors about some types of materials, experimental systems and methods used in many studies. Here, indicate whether each material, system or method listed is relevant to your study. If you are not sure if a list item applies to your research, read the appropriate section before selecting a response.

## Materials & experimental systems

| n/a | Involved in the study |
|---|---|
| ☐ | [X] Antibodies |
| ☐ | [X] Eukaryotic cell lines |
| [X] | ☐ Palaeontology and archaeology |
| ☐ | [X] Animals and other organisms |
| [X] | ☐ Clinical data |
| [X] | ☐ Dual use research of concern |
| [X] | ☐ Plants |

## Methods

| n/a | Involved in the study |
|---|---|
| [X] | ☐ ChIP-seq |
| ☐ | [X] Flow cytometry |
| [X] | ☐ MRI-based neuroimaging |

## Antibodies

| | |
|---|---|
| Antibodies used | All antibodies used include anti-CD206 (catalog 141721, Biolegend), anti-CD3 (catalog 100353, Biolegend), anti-CD4 (catalog 100480, Biolegend), anti-CD8 (catalog 100725, Biolegend), anti-B220 (catalog 103210, Biolegend), anti-NKp46 (catalog 137618, Biolegend), anti-FOXP3 (catalog , Invitrogen), anti-CD69 (catalog 104506, Biolegend), anti-PD-1 (catalog 135228, Biolegend), anti-Ki67 (catalog 58-5698-82, Invitrogen), anti-CD11b (catalog 101267, Biolegend), anti-CD11c (catalog 117320, Biolegend), anti-LY6G (catalog 127654, Biolegend), anti-LY6C (catalog 117320, Biolegend), anti-MHC-II (catalog 107643, Biolegend), anti-F4/80 (catalog 123120, Biolegend), anti-CD31 (catalog 102507, Biolegend), anti-CD45.2 (catalog 109839, Biolegend), anti-PD-L1 (catalog 748275, BD), anti-CD86 (catalog 105036, Biolegend), anti-CD183 (catalog 126506, Biolegend), anti-CD44 (catalog 103029, Biolegend), anti-CD62L (catalog 104445, Biolegend), viability due (catalog 740614, Invitrogen), anti-SPARC (catalog 8725T, cell signaling), anti-HSP90 (catalog 4877T, cell signaling), anti-IgG secondary (catalog 31460, ThermoFisher Scientific). |
| Validation | For generated nanobodies, in house validation was performed using flow cytometry or isothermal calorimetry. All antibodies utilized in this study are commercially sourced and have been previously employed for the same applications reported here. Each supplier performs quality control to guarantee the reliability and reproducibility of their products. Detailed validation information and relevant literature citations for each antibody are available on the datasheets provided by the manufacturers |

April 2023

# Eukaryotic cell lines

Policy information about cell lines and Sex and Gender in Research

| | |
|---|---|
| Cell line source(s) | A549-Duals, THP-1 Duals, and RAW 264.7 cells were purchased from InvivoGen. EMT6, B16.F10, and B16.F10-LUC2 were purchased from ATCC. B16.F10-OVA were a gift from Dr. Amanda Lund, and originally purchased from Vitro Biotech. Primary cells were harvested in house. |
| Authentication | Cell lines were authenticated by testing binding of known receptors using flow cytometry with commercial antibodies. |
| Mycoplasma contamination | Cell lines were tested routinely for mycoplasma, all cells were negative for mycoplasma. |
| Commonly misidentified lines (See ICLAC register) | The cell lines used in this study are not found on ICLAC register Version 11. |

# Palaeontology and Archaeology

| | |
|---|---|
| Specimen provenance | |
| Specimen deposition | |
| Dating methods | |

☐ Tick this box to confirm that the raw and calibrated dates are available in the paper or in Supplementary Information.

| | |
|---|---|
| Ethics oversight | |

Note that full information on the approval of the study protocol must also be provided in the manuscript.

# Animals and other research organisms

Policy information about studies involving animals; ARRIVE guidelines recommended for reporting animal research, and Sex and Gender in Research

| | |
|---|---|
| Laboratory animals | Female C57BL/6J and Balb/c, and female FVB/N-Tg (MMTV-PyVT)634Mul, mice were acquired from Jackson labs (6-8 weeks old; 6–20-week-old mice were used for challenge experiments). CD45.1+/− OT-I female mice were purchased from Jackson labs (4-8 weeks old; C57BL/6-Tg(TcraTcrb)1100Mjb/J strain). Mice were kept on a 12 light/12 dark cycle and temperatures of 65-75 °F with 40-60% humidity. |
| Wild animals | No wild animals were used in the study. |
| Reporting on sex | Studies indicated the sex of mice used. |
| Field-collected samples | No field-collected samples were used in the study. |
| Ethics oversight | All experiments were performed in accordance with protocols approved by the Institutional Animal Care and Use Committee within Vanderbilt University. IACUC protocols M1800155 and M2300004. |

Note that full information on the approval of the study protocol must also be provided in the manuscript.

# Clinical data

Policy information about clinical studies

All manuscripts should comply with the ICMJE guidelines for publication of clinical research and a completed CONSORT checklist must be included with all submissions.

| | |
|---|---|
| Clinical trial registration | |
| Study protocol | |
| Data collection | |
| Outcomes | |

# Dual use research of concern

Policy information about dual use research of concern

## Hazards

Could the accidental, deliberate or reckless misuse of agents or technologies generated in the work, or the application of information presented in the manuscript, pose a threat to:

No | Yes

☐ | ☐ Public health

☐ | ☐ National security

☐ | ☐ Crops and/or livestock

☐ | ☐ Ecosystems

☐ | ☐ Any other significant area

## Experiments of concern

Does the work involve any of these experiments of concern:

No | Yes

☐ | ☐ Demonstrate how to render a vaccine ineffective

☐ | ☐ Confer resistance to therapeutically useful antibiotics or antiviral agents

☐ | ☐ Enhance the virulence of a pathogen or render a nonpathogen virulent

☐ | ☐ Increase transmissibility of a pathogen

☐ | ☐ Alter the host range of a pathogen

☐ | ☐ Enable evasion of diagnostic/detection modalities

☐ | ☐ Enable the weaponization of a biological agent or toxin

☐ | ☐ Any other potentially harmful combination of experiments and agents

# Plants

Seed stocks

Novel plant genotypes

Authentication

# ChIP-seq

## Data deposition

☐ Confirm that both raw and final processed data have been deposited in a public database such as GEO.

☐ Confirm that you have deposited or provided access to graph files (e.g. BED files) for the called peaks.

Data access links
*May remain private before publication.*

Files in database submission

Genome browser session
(e.g. UCSC)

## Methodology

Replicates

Sequencing depth

Antibodies

Peak calling parameters

Data quality

Software

# Flow Cytometry

## Plots

Confirm that:

[x] The axis labels state the marker and fluorochrome used (e.g. CD4-FITC).

[x] The axis scales are clearly visible. Include numbers along axes only for bottom left plot of group (a 'group' is an analysis of identical markers).

[x] All plots are contour plots with outliers or pseudocolor plots.

[x] A numerical value for number of cells or percentage (with statistics) is provided.

## Methodology

| | |
|---|---|
| Sample preparation | As described in methods, at brief: EMT6 tumor bearing Balb/c and B16.F10-OVA bearing C57BL/6 mice were euthanized either 24 h or 48 h after final treatment. Spleens and tumors were harvested, weighed, and placed onto ice. Tumors were digested in RPMI 1640 media containing a tumor dissociation kit. Tumors were dissociated and incubated for 30 min at 37 ºC for complete digestion. Tumors and spleens were mashed and separated into single cell suspensions and red blood cells were used twice using ACK lysis buffer. Cells were resuspended in flow buffer (2% FBS and 50µM dasatinib), counted, and stained with Fc-block (aCD16/32, 2.4G2, Tonbo) for 15min at 4 ºC, and then stained with the appropriate antibodies for 1hr at 4 ºC. Cells were then washed again with FACS buffer, fixed with 2% paraformaldehyde for 10 min, washed again with FACS buffer containing AccuCheck counting beads. |
| Instrument | Flow cytomrtry data were analyzed on a Cytek Aurora flow cytometer and on an Amnex Luminex CellStream flow cytometer. |
| Software | All flow cytometry data were analyzed using FlowJo software (version 10; Tree Star; https://www.flowjo.com/solutions/flowjo). |
| Cell population abundance | Cell population abundance is shown in gating strategies. |
| Gating strategy | Gating strategies are shown in supplemental figures. |

[x] Tick this box to confirm that a figure exemplifying the gating strategy is provided in the Supplementary Information.

# Magnetic resonance imaging

## Experimental design

| | |
|---|---|
| Design type | |
| Design specifications | |
| Behavioral performance measures | |

| | |
|---|---|
| Imaging type(s) | |
| Field strength | |
| Sequence & imaging parameters | |
| Area of acquisition | |

Diffusion MRI      ☐ Used      ☐ Not used

## Preprocessing

| | |
|---|---|
| Preprocessing software | |
| Normalization | |
| Normalization template | |
| Noise and artifact removal | |
| Volume censoring | |

## Statistical modeling & inference

| | |
|---|---|
| Model type and settings | |
| Effect(s) tested | |

Specify type of analysis:      ☐ Whole brain      ☐ ROI-based      ☐ Both

Statistic type for inference

(See Eklund et al. 2016)

Correction

## Models & analysis

| n/a | Involved in the study |
|-----|----------------------|
| ☐ ☐ | Functional and/or effective connectivity |
| ☐ ☐ | Graph analysis |
| ☐ ☐ | Multivariate modeling or predictive analysis |

Functional and/or effective connectivity

Graph analysis

Multivariate modeling and predictive analysis

