## [Peer Review File · Nature Biomedical Engineering]

Potentiating cancer immunotherapies with modular albumin-hitchhiking nanobody-STING agonist conjugates

Corresponding Author: Prof John Wilson

Version 0:

Decision Letter:

Dear Prof Wilson,

Thank you again for submitting to *Nature Biomedical Engineering* your manuscript, "Programable Albumin-Hitchhiking Nanobodies Enhance the Delivery of STING Agonists to Potentiate Cancer Immunotherapy". The manuscript has been seen by 5 experts, whose reports you will find at the end of this message.

You will see that the reviewers appreciate the work. However, they express concerns about the degree of support for the claims, and provide useful suggestions for improvement. We hope that with significant further work you can address the criticisms and convince the reviewers of the merits of the study. In particular, we would expect that a revised version of the manuscript provides:

- * Assessment of the off-target toxicity of the STING–anti-albumin nanobody conjugates.
- * Improved characterization of the mechanisms of cellular uptake of the conjugates and of the subsequent release of the STING agonist, as suggested by Reviewers #4 and #5.
- * A more detailed analysis of the in vivo results, supplemented with any needed additional data, to clarify any apparently inconsistent molecular and cellular readouts, as per the various relevant comments of Reviewers #2, #3 and #4.
- * Investigation of the specific immune-cell populations responsible for instigating the in vivo antitumor responses.

When you are ready to resubmit your manuscript, please upload the revised files, a point-by-point rebuttal to the comments from all reviewers, the [reporting summary](https://www.nature.com/authors/policies/ReportingSummary.pdf), and a cover letter that explains the main improvements included in the revision and responds to any points highlighted in this decision.

Please follow the following recommendations:

- * Clearly highlight any amendments to the text and figures to help the reviewers and editors find and understand the changes (yet keep in mind that excessive marking can hinder readability).
- * If you and your co-authors disagree with a criticism, provide the arguments to the reviewer (optionally, indicate the relevant points in the cover letter).
- * If a criticism or suggestion is not addressed, please indicate so in the rebuttal to the reviewer comments and explain the reason(s).
- * Consider including responses to any criticisms raised by more than one reviewer at the beginning of the rebuttal, in a section addressed to all reviewers.
- * The rebuttal should include the reviewer comments in point-by-point format (please note that we provide all reviewers will the reports as they appear at the end of this message).
- * Provide the rebuttal to the reviewer comments and the cover letter as separate files.

We hope that you will be able to resubmit the manuscript within 20 weeks from the receipt of this message. If this is the case,

you will be protected against potential scooping. Otherwise, we will be happy to consider a revised manuscript as long as the significance of the work is not compromised by work published elsewhere or accepted for publication at *Nature Biomedical Engineering*.

We hope that you will find the referee reports helpful when revising the work. Please do not hesitate to contact me should you have any questions.

Best wishes,

Filipe

Dr Filipe Almeida
Associate Editor, *Nature Biomedical Engineering*

Reviewer #1 (Report for the authors (Required)):

In this paper the development of an anti-albumin nanobody for obtaining life-time extension by means of albumin hitchhiking is presented. An increase in life-time was observed from 5 min to 55 hrs (2-3 days). This anti-albumin nanobody was conjugated using click chemistry method to a stimulator of interferon genes or STING. The improved pharmacokinetics due to life-time extension was demonstrated by the extended activation of the innate immunity response in several mouse tumor models. In addition, the authors developed an anti-PD-L1 nanobody which was genetically fused to the anti-albumin nanobody resulting in a bi-specific nanobody construct conjugated to STING: AP-diABZI. Again, the efficacy of this compound was demonstrated in several mouse tumor models. The authors conclude that the anti-albumin nanobody offers an approach to augment the response to different immunotherapeutic modalities.

The paper describes the application of STING, optimized by introducing a life-time extender, in the stimulation of the innate immune response for the treatment of cancer. Different tumor models for the mouse were investigated, the EMT6 breast cancer model and the B16.F10 melanoma tumor model. Several aspects of tumor characteristics were investigated, not only survival of the mice, but also detailed effects on cell composition of the tumor. This resulted in an impressive amount of work with a clear positive effect of their final compound, AP-diABZI, on tumor treatment efficacies, although never a complete remission of the tumor was obtained.

The main problem with this paper is that the approach of using an anti-albumin nanobody as a life-time extender is presented as a novel approach while the first application of a such a nanobody for life-time extension of possible immunotherapeutic nanobodies was described already in 2011 (ref 29) and since then, many others have used the same approach (see for instance Shen et al 2021: <https://doi.org/10.1016/j.isci.2021.103014>). Albumin-hitchhiking has also been achieved with the application of an albumin-binding domain which is even smaller as the nanobody and similarly effective in inducing life-time extension (see for instance: Xenaki et al. for nanobodies doi: 10.7150/thno.57510). Similar data can be found for life-time extension of humabodies and affibodies. The affinity of this albumin-binding domain has already been modified by introducing point mutations, so the future development of albumin hitchhiking approach as mentioned in the discussion has already been achieved. The authors should put their research in the appropriate context and refer to already existing data, before this paper can be accepted for publication.

As their main conclusion of life-time extension is not novel, this manuscript confirms old data by showing again that for in vivo treatment of tumors, life-time extension is an absolute requirement. The novelty of this paper is the application of STING as a stimulator for the innate immune system and effects on tumor development. This may have broad application for the treatment of tumors. However, a complete remission was never observed, meaning that this STING approach will need to be employed in combination with other anti-cancer therapeutics.

Technical criticisms:

P236-240: the conclusion that the albumin binding nanobody introduces the tumor accumulation is not correct. As a proof for this conclusion is figure 2G, where molecules without life extension module such as nEGFR are compared with the life-time extender nAlb. The difference in the signal is due to the life-time extender not to tumor accumulation. Tumor accumulation means more nAlb in a tumor than in other organs of the same mouse. This is also shown in figure 2G. However, the method to quantify fluorescently labeled proteins is poorly described but this seems to be done on section, per surface. This does not take into account the amount of cells per volume. In other words, the quantity of the therapeutic molecule should be presented as the percentage of injected dose per gr of tissue, a standard way of investigating the presence of a therapeutic protein in tissue.

Minor points:

P155: is the diABZI compound membrane permeable as this would be required for endosomal escape?

P300: "Based on the long half-life of nAlb and its capacity to dramatically enrich cargo distribution to tumor sites". What would be the rationale for the 'dramatic enrichment of cargo in the tumor'?

P390: Also for the anti-PD-L1 nanobody the authors seem to ignore that such a nanobody already exists for several years: for instance Zhang et al 2017 doi: 10.1038/celldisc.2017.4
Referring to the literature about these anti-PD-L1 is required.

P392: what is the affinity of this anti-PD-L1 nanobody?

P434: also for the AP nanobodies, tissue presence has not been determined in the correct way.

Reviewer #2 (Report for the authors (Required)):

This work designed a modular, safe, and efficient nanobody delivery platform based on synthetic biology technology. By hitchhiking on albumin, it achieves efficient tumor penetration of STING agonists, which can effectively increase the infiltration of NK cells and T cells and produce excellent therapeutic effects in multiple tumor models. At the same time, by integrating a PD-L1 binding domain to create a bi-valent fusion protein, the tumor-targeting ability and anti-tumor therapeutic effect of the treatment system are significantly enhanced, showing satisfactory therapeutic effects in preclinical tumor treatment models. This work provides a programmable, multifunctional, and safe delivery platform for immunotherapy. However, we believe that this work can be further improved.

Here are some other issues that need to be addressed:

1. It's well accepted that MDSCs are immunosuppressive cells. However, as shown in Fig 2j and Fig 2k, nAlb-diABZI resulted in an increased frequency of MDSCs. What factors may be responsible for this phenomenon and what impact it will have on subsequent immunotherapy ?
2. As shown in Fig 4f, nPD-L1-diABZI, nAlb-diABZI and AP-diABZI could efficiently activate the STING pathway. However, the relative expression of *Ilnb* and *Cxcl10* with the treatment of AP-diABZI, which are representative downstream immune responses of STING activation, are not better than those of nAlb-diABZI. Additionally, the difference in immune response between AP-diABZI and nPD-L1-diABZI on BMDC (Fig S12) should be discussed in the manuscript.
3. Mice treated with AP-diABZI and nAlb-diABZI+ICB exhibited significant antitumor responses in the EMT6 tumor-bearing model (Fig. 5b-f). However, as shown in Fig 5g-l, these assays did not include mice treated with nAlb-diABZI+ICB, making it difficult to understand the different mechanisms of the increased efficacy of AP-diABZI and nAlb-diABZI+ICB. Further discussion may be beneficial.
4. Mice treated with AP-diABZI showed significant antitumor responses in the B16.F10 therapeutic model (Fig.7). However, as shown in Fig. 7e, the production of cytokines associated with antitumor immunity was higher in mice treated with nAlb-diABZI group compared to mice treated with nAlb-diABZI+ICB group. Further discussion may be beneficial.
5. Mice showed weight loss after the first injection of nAlb-diABZI (Fig. S8 and Extended Data Fig. 1). How about the weight changes in mice during treatment in other models (Fig. 3, Fig 5., Fig. 8, Fig. S7).
6. As shown in Extended Data Fig. 5, a single dose of AP-diABZI treatment resulted in the presence of more g-MDSCs than m-MDSCs when compared to the PBS and nAlb-diABZI groups. What are the possible mechanisms that lead to this response?

Reviewer #3 (Report for the authors (Required)):

In this manuscript, Wilson and co-workers developed a type of albumin-hitchhiking drug delivery platform for the systemic administration of STING agonists for cancer immunotherapy. STING agonist diABZI was covalently conjugated with anti-albumin nanobody to realize the albumin-hitchhiking pharmacokinetics properties and increase the tumor tropism, and the programmability of the platform allow the integration of a second functional nanobody domain (like anti-PD-L1). Though the concept of albumin-hitchhiking is not novel in anti-tumor drug delivery field, taking the commercial albumin-bound paclitaxel as example, the special features of this work are that the drugs conjugated were immune agonists rather than chemotherapeutic agents, and that the anti-albumin nanobody hitchhiking strategy was used instead of direct albumin conjugation. Moreover, the designed drug has a single component, well-defined function, and impressive therapeutic outcomes in multiple mouse tumor models, which illustrate its potential for clinical application and translation. In summary, this work is significant in providing a promising multimodality programmable delivery platform technology for systemic delivery of immune agonists. However, some minor writing and experimental details remain to be refined before accepted for publication.

1. One of the most important issues for systemic administration of immune agonists is the off-target toxicity. Although nAlb-diABZI could target albumin and tend to the tumor site, it would inevitably affect other organs as well. The current manuscript lacks necessary evaluation on this point in various organs after administration.

2. The schematic of figure 2a speculated the pathway of nAlb-diABZI after entry into cells, but no specific experimental results are available for verification. Experiments like cell confocal assays should be supplemented to illustrate the fate of drug in early and late endosome after entry, as well as to demonstrate that STING agonist molecules can escape from endosomes to cytoplasm for STING activation.
3. In figure 1j, why agonist conjugated with nanobody (nAlb-diABZI) triggered higher activation level in Irfn and Cxcl1 than original diABZI?
4. In figure 4d-e, how to explain the interferon activities of diABZI conjugated with nPD-L1 (nPD-L1-diABZI and AP-diABZI) were higher than other groups at high agonist concentration.
5. In figure 5 and figure 7, mouse weights results should be supplemented.
6. In introduction section, it is better to add some comparisons with current albumin-hitchhiking drugs, such as the difference of the delivery of immune agonists and chemotherapeutic agents, and the advantages of the nanobody in situ conjugating albumin strategy.
7. The paragraph from line 187 to 196 lacks the reference marking of figure1 h-j.
8. There is a word misspelling in line 632, which “micrometases” should be corrected to “micrometastases”.

Reviewer #4 (Report for the authors (Required)):

This study by Wilson et al describes the development of a nanobody-conjugated STING agonist with albumin-hitchhiking property for activation of tumor immunity. There are multiple strengths of the paper: much longer blood circulation of the nAlb-diABZI complex, increased tumor accumulation, comprehensive immune profiling after therapy, antitumor efficacy in multiple tumor models, and formation of bifunctional AP-diABZI that also target PD-L1. However, there are also many concerns over the current design and data, which can improve the enthusiasm toward this study after careful addressing.

1. Although longer blood circulation is beneficial for tumor accumulation, it may also elevate the toxicity. In fact, data show dramatically higher systemic cytokines (Fig. 7E) as well as some liver toxicity (Extended Data Fig. 1). Given the first antibody-diABZI conjugate XMT-2056 incurred patient death and halted clinical trials, the authors may need to carefully address the toxicity and safety concerns for nAlb-diABZI complex. A careful dose-response study is warranted to show the therapeutic window for the safe usage of the otherwise novel design.
2. Is free diABZI released from antibody-diABZI for STING activation in tumors? The click linker does not seem to be degradable. Active agent if not diABZI that binds and activates STING should be characterized.
3. Would nAlb-diABZI therapy work in tumors with less albumin catabolism capacity?
4. The authors show significant increase in NK, CD4 T and CD8 T cell populations after STING therapy. Cell blocking experiments should be performed to evaluate which immune cell population is responsible for antitumor efficacy.
5. m- and g-MDSC population also increased after nAlb-diABZI therapy (Fig. 3). It has been shown that both MDSC populations render different cell proliferation and metastatic properties in tumors (Ouzounova et al, Nature Comm. 2017, 8, 14979). Can authors comment on whether this is directly due to nAlb-diABZI, or potential systemic cytokine induced effect? Would MDSC drive resistance to nAlb-diABZI therapy?
6. Can authors investigate whether nAlb-diABZI therapy result in formation of albumin-specific mAbs and impact clearance of nAlb-diABZI after 2nd or 3rd injections?
7. On the style side, the authors packed too many subfigures in each main figure. It will be helpful to prioritize the data selection. It diminishes general readability if the font size is too small and organization too complex.

Reviewer #5 (Report for the authors (Required)):

In this manuscript, the authors constructed a bifunctional nanobody for delivering a STING agonist to stimulate innate immune programs for tumor suppression. The nanobody was composed of an anti-albumin nanobody (nAlb) and an anti-PD-L1 nanobody (nPD-L1), hence the conjugate demonstrated extended in vivo circulation by “albumin-hitchhiking”, resulting in enhanced cellular uptake through albumin-mediated endocytosis, confirmed by the antibody blockade of SPARC and GP60. The presence of anti-PD-L1 nanobody further enhanced the immune-response through immune checkpoint inhibition. The treatment increased the infiltration of activated natural killer cells and T cells, eliminated primary tumors, and resulted in immunological memory even 80 days after initial tumor inoculation.

While the immunological assays were comprehensively performed in this work, the design of the delivery platform is rather

straightforward and the novelty is marginal. The programmable linking bi- even multi-functional nanobodies has been well-investigated, using anti-albumin nanobody to prolong in vivo circulation is a general strategy for nanobody applications (Caplacizumab in clinical use, *N Engl J Med* 2019; 380, 335-346, and many others, for example *FASEB J.* 2018, 32(6) 3411-3422); site-specific loading drug in multi-functional nanobodies has been reported (*Chem Commun*, 2019, 55(35), 5175-5178). In addition, the following issues should be addressed.

1. One of my major concerns is the cellular uptake of the drug through nanobody binding to albumin. Different from cell-based assay, in which the total amount of albumin is limited and fixed, the ratio of nAlb-bound albumin in vivo was very low. I did a rough calculation. A 25 g mouse has 1.8 mL blood and contains 42 mg albumin (23.6 mg/mL, <https://bionumbers.hms.harvard.edu/bionumber.aspx?s=n&v=2&id=102312>), which is about 0.6 micromoles. When mice were treated with 2 mg/Kg nanobody, which was about 3.5 nanomole in a mouse. The ratio of nAlb-bound albumin was < 1% among total albumin in vivo.

Assuming the nanobody binding does not increase the endocytosis efficiency of albumin (otherwise the authors should provide the evidence), very little amount of nAlb-bound albumin can enter tumor cells. So how can nAlb and the drug conjugates be accumulated in tumors through albumin-mediated endocytosis? The majority of albumin did not contain nanobody.

2. The ligation of diABZI to nanobody was through click chemistry with DBCO-PEG11-diABZI, so the drug was linked to the nanobody via a covalent bond. Then, how can the drug release from the protein and exert its function? Lysosomal degradation was postulated in the manuscript; the authors should provide evidence of the drug release and the format (structure) of the released product – whether this product is as effective as diABZI?

3. The albumin-mediated endocytosis of the nanobody was measured on nAlb-Cy5 and nEGFR-Cy5 using anti-SPARC and anti-GP60 antibodies; it is unknown whether the endocytosis efficiency can be affected by nPD-L1 linking and PEG11-diABZI modification, as nPD-L1 can also induce endocytosis in a different way. Indeed, the effect of nPD-L1 on endocytosis of the bivalent nanobody should be analyzed.

4. In Figure 2b, the two types of cells have different EGFR levels, but the levels of the SPARC and GP60 are unknown. The membrane binding of nEGFR to EGFR+ cells was observed, and nAlb did not bind to any cells. Is that because of low levels of albumin receptors?

The method to measure membrane binding and endocytosis of nanobody was indirect. The authors may use confocal fluorescence imaging to directly view the result.

5. Figure 2c, the addition of anti-SPARC and anti-GP60 antibodies decreased the internalization of nAlb-Cy5. It can be seen that the internalization was influenced by antibodies more significantly in B16.F10 cells than in EMT6 cells. Can this result suggest higher efficiency of endocytosis in B16.F10 cells? Could this difference affect the in vivo treatment?

6. Figure 4c: the mass spectrum of the final product AP-diABZI was missing; SDS PAGE cannot tell the homogeneity (ligation ratio) of the product.

7. In the study of “nAlb-diABZI potently stimulates STING activation in the TME to inhibit tumor growth”, the authors analyzed the effect of nAlb-diABZI on B16.F10 (PD-L1 low) and EMT6 (PD-L1 high) tumors. It seems nAlb-diABZI was effective on both types of tumors. Should PD-L1 level affect the treatment? Also in the study of immunological memory in the next sections, were there differences between the two types of cells? The authors should make a comparison and give conclusions.

8. Figure 4m, among the ratio range of 0 – 50 indicated by color gradients, it would be helpful to indicate the color code for 1.

Version 1:

Decision Letter:

Dear Prof Wilson,

Thank you again for submitting to *Nature Biomedical Engineering* your manuscript, "Programmable Albumin-Hitchhiking Nanobodies Enhance the Delivery of STING Agonists to Potentiate Cancer Immunotherapy". The manuscript has been seen by 5 experts, whose reports you will find at the end of this message. You will see that the reviewers appreciate the work, but reviewer #1 raise few technical criticisms that we hope you will be able to address.

When you are ready to resubmit your manuscript, please upload the revised files, a point-by-point rebuttal to the comments from all reviewers, the [reporting summary](https://www.nature.com/authors/policies/ReportingSummary.pdf), and a cover letter that explains the main improvements included in the revision and responds to any points highlighted in this decision.

Please follow the following recommendations:

- * Clearly highlight any amendments to the text and figures to help the reviewers and editors find and understand the changes (yet keep in mind that excessive marking can hinder readability).
- * If you and your co-authors disagree with a criticism, provide the arguments to the reviewer (optionally, indicate the relevant points in the cover letter).
- * If a criticism or suggestion is not addressed, please indicate so in the rebuttal to the reviewer comments and explain the reason(s).
- * Consider including responses to any criticisms raised by more than one reviewer at the beginning of the rebuttal, in a section addressed to all reviewers.
- * The rebuttal should include the reviewer comments in point-by-point format (please note that we provide all reviewers will the reports as they appear at the end of this message).
- * Provide the rebuttal to the reviewer comments and the cover letter as separate files.

We hope that you will be able to resubmit the manuscript within 20 weeks from the receipt of this message. If this is the case, you will be protected against potential scooping. Otherwise, we will be happy to consider a revised manuscript as long as the significance of the work is not compromised by work published elsewhere or accepted for publication at *Nature Biomedical Engineering*.

We hope that you will find the referee reports helpful when revising the work, which we look forward to receive. Please do not hesitate to contact me should you have any questions.

Best wishes,

Filipe

Dr Filipe Almeida
Senior Editor, Nature Biomedical Engineering

Reviewer #1 (Report for the authors (Required)):

This paper has been considerably improved. However, not all remarks have been answered satisfactorily and additional experiments also raised new questions. The paper is acceptable for publication after further improvement based on my comments:

1. Page 29: Half-life extension by albumin binding peptides/nanobodies have now been described. However, the use of the streptococcal G-derived albumin binding domain as HLE has still been ignored, while this is a very promising new development. See for instance doi:10.1093/protein/gzq058. The affinity of this ABD can be adjusted to adjust half-life: see for instance: DOI: 10.7150/thno.57510. Affibodies and humabodies have been used in combination with this ABD, doi: 10.3390/ijms21082999 and doi: 10.1016/j.neo.2023.100962.
2. Line 208: Puzzling for me are data from figure 1h,i,j : Why is there no difference in interferon activity (h,i), but a clear difference for interferon activated gene expression (j) ?
3. Line 213-223: The description of figure 2 should start with the results of figure 2a instead of starting with a discussion about figure 2e.
4. Line 213-223: Tumor accumulation as a result of HLE is depicted in fig 2e,f. It is indeed interesting to find an explanation for this phenomenon but this should be done in the discussion section and not in the results section. Besides the mentioned EPR effect the most simple explanation has not been mentioned: due to HLE the nanobody is not immediately removed by the kidneys and now it has the time to be taken up by the cells via a non-specific pinocytosis route
5. Figure 2: the fluorescent images need higher magnification.

Reviewer #2 (Report for the authors (Required)):

I am satisfied with the response and the revision.

Reviewer #3 (Report for the authors (Required)):

In the revision, the authors addressed the comments effectively. Notably, the key improvements include:

1. Addition of a comparative discussion on other albumin-hitchhiking drugs and the rationale for employing nanobodies.
2. Inclusion of a comprehensive preclinical toxicity analysis.
3. Supplement of experimental data to further elucidate the mechanism of cell internalization and STING activation by nAlb-diABZI.

The supplementary experimental data and textual descriptions in the revision are satisfactory, and there are no further comments on this article.

Reviewer #4 (Report for the authors (Required)):

The authors have comprehensively addressed my prior concerns. I am satisfied with the revision.

Reviewer #5 (Report for the authors (Required)):

The authors have addressed my concerns; I don't have further questions.

Version 2:

Decision Letter:

Dear Prof Wilson,

Thank you for your revised manuscript, "Programmable Albumin-Hitchhiking Nanobodies Enhance the Delivery of STING Agonists to Potentiate Cancer Immunotherapy". Having consulted with Reviewer #1 (whose comments you will find at the end of this message), I am pleased to write that we shall be happy to publish the manuscript in *Nature Biomedical Engineering*.

We will be performing detailed checks on your manuscript, and in due course will send you a checklist detailing our editorial and formatting requirements. You will need to follow these instructions before you upload the final manuscript files.

Best wishes,

Filipe

Dr Filipe Almeida
Senior Editor, <http://www.nature.com/nbme> > *Nature Biomedical Engineering*

Reviewer #1 (Report for the authors (Required)):

The authors have further improved the manuscript and in my opinion it is now acceptable for publication.

Response to Reviewers:

The authors thank the reviewers for their constructive feedback. We have made changes accordingly in the revision of the manuscript. Specific alterations relating to individual reviewer comments along with the original comments or critiques are provided below. A revised version of the manuscript is provided where changes have been highlighted. Citations in parentheses () contained within this response refer to the reference numbers at the end of this document whereas those that are superscripted correspond to citations found in the manuscript text. We believe that revisions made in response to the reviewer comments have strengthened this work. We hope that the comments and questions raised by the reviewers have all been addressed in a satisfactory manner and that this revised manuscript is acceptable for publication in *Nature Biomedical Engineering*.

Reviewer #1 (Report for the authors (Required)):

In this paper the development of an anti-albumin nanobody for obtaining life-time extension by means of albumin hitchhiking is presented. An increase in life-time was observed from 5 min to 55 hrs (2-3 days). This anti-albumin nanobody was conjugated using click chemistry method to a stimulator of interferon genes or STING. The improved pharmacokinetics due to life-time extension was demonstrated by the extended activation of the innate immunity response in several mouse tumor models. In addition, the authors developed an anti-PD-L1 nanobody which was genetically fused to the anti-albumin nanobody resulting in a bi-specific nanobody construct conjugated to STING: AP-diABZI. Again, the efficacy of this compound was demonstrated in several mouse tumor models. The authors conclude that the anti-albumin nanobody offers an approach to augment the response to different immunotherapeutic modalities.

The paper describes the application of STING, optimized by introducing a life-time extender, in the stimulation of the innate immune response for the treatment of cancer. Different tumor models for the mouse were investigated, the EMT6 breast cancer model and the B16.F10 melanoma tumor model. Several aspects of tumor characteristics were investigated, not only survival of the mice, but also detailed effects on cell composition of the tumor. This resulted in an impressive amount of work with a clear positive effect of their final compound, AP-diABZI, on tumor treatment efficacies, although never a complete remission of the tumor was obtained.

The main problem with this paper is that the approach of using an anti-albumin nanobody as a life-time extender is presented as a novel approach while the first application of a such a nanobody for life-time extension of possible immunotherapeutic nanobodies was described already in 2011 (ref 29) and since then, many others have used the same approach (see for instance Shen at al 2021: <https://doi.org/10.1016/j.isci.2021.103014>). Albumin-hitchhiking has also been achieved with the application of an albumin-binding domain which is even smaller as the nanobody and similarly effective in inducing live-time extension (see for instance: Xenaki at al. for nanobodies doi: 10.7150/thno.57510). Similar data can be found for life-time extension of humabodies and affibodies. The affinity of this albumin-binding domain has already been modified by introducing point mutations, so the future development of albumin hitchhiking approach as mentioned in the discussion has already been achieved. The authors should put their research in the appropriate context and refer to already existing data, before this paper can be accepted for publication.

As their main conclusion of life-time extension is not novel, this manuscript confirms old data by showing again that for in vivo treatment of tumors, life-time extension is an absolute requirement. The novelty of this paper is the application of STING as a stimulator for the innate immune system and effects on tumor development. This may have broad application for the treatment of tumors. However, a complete remission was never observed, meaning that this STING approach will need to be employed in combination with other anti-cancer therapeutics.

We thank the Reviewer for their thorough assessment of our manuscript. We agree with the Reviewer that we did not adequately place our work in the context of previous literature describing the use of albumin-binding domains to improve the delivery of cancer therapeutics. While restraints in manuscript length and reference number precluded an in-depth analysis of this important prior work, we have made changes to the text to discuss

previous research using albumin as a drug carrier, how this motivated our strategy for delivery of STING agonists, and why we chose a protein-based albumin binder; these changes appear on **Pages 4, 29, and 30** and below:

Page 4: “Albumin is a promising drug carrier based on its long circulation half-life and proclivity to accumulate at tumor sites via both passive and active transport mechanisms.²⁷⁻²⁹ Albumin and albumin-binding chaperones have been widely employed to improve the delivery of chemotherapeutics, exemplified by albumin-bound paclitaxel (Abraxane®)³⁰, as well as protein³¹, peptide³², and nucleic acid therapeutics³⁰. Inspired by this previous work that motivates the unexplored potential of albumin as a carrier for STING agonists, we engineered a high-affinity anti-albumin nanobody (i.e., single-domain antibody) for site-selective enzymatic bioconjugation of STING agonists via biorthogonal chemistry.”

Page 29: “An appealing feature of anti-albumin nanobodies, and other protein-based albumin hitchhiking agents (e.g., affibodies, humabodies) over other albumin binders (e.g., lipids, Evans blue) is the high degree of molecular programmability that can be achieved through protein engineering.”

Page 30: “To date, most research on albumin-based drug carriers for cancer (e.g., Abraxane, aldoxorubicin) has focused on delivery of chemotherapy drugs that target cancer cells. By contrast, immunostimulatory agents such as STING agonists can stimulate complex antitumor immunological programs that may be more dependent on immunological variables (e.g., neoantigen load, immune status of the TME) than on the efficiency of drug accumulation in tumor tissue or delivery to cancer cells.”

Page 30: “Indeed, anti-albumin nanobodies have been engineered with variable affinity⁸² and this may afford a future opportunity for precisely modulating plasma half-life to establish immunopharmacological principles to optimize systemic innate immune agonist delivery.”

As now more clearly articulated in our revised manuscript on **Pages 4 and 30**, while albumin and albumin-binding carriers have been used to increase drug accumulation at tumor sites, this strategy has not been explored for delivery of STING agonists, a promising class of therapeutics with mechanisms of action that are distinct from chemotherapeutics. Therefore, harnessing albumin-hitchhiking to improve the delivery of STING agonists represents a conceptually different approach from previous work in the field that has mostly leveraged synthetic nanoparticles. We also believe that our approach represents a significant technological innovation as we devised a robust workflow for synthesis of nanobody-STING agonist conjugates that combines site-selective enzymatic bioconjugation with biorthogonal ligation to a novel “clickable” STING agonist at a precisely defined ratio. This is an enabling synthetic scheme for covalent conjugation of a leading STING agonist to drug carriers that we believe has broad potential future applications. Additionally, as exemplified by our integration of a PD-L1 binding domain, our strategy provides “plug-and-play” programmability for optimizing agonist delivery. This yielded in single, well-defined, multi-functional molecule that resulted in complete remission and durable immune memory in an orthotopic, syngeneic EMT6 model of breast cancer (see **Figure 5c,d**) and we further discuss this on **Page 29** and below:

Page 29: “This yielded a single, well-defined, multifunctional STING agonist that increased tumor accumulation in a PD-L1-dependent manner, while also blocking an important immune checkpoint, resulting in spontaneous induction of tumor antigen-specific T cells that inhibited tumor growth and provided immunological memory that protected against tumor rechallenge.”

Figure 5: (c) spider plots of individual tumor growth curves, and **(d)** Kaplan-Meier survival plots for mice with EMT6 tumors treated as indicated. CR = complete responder; SEM with *P* value determined by two-way ANOVA

with post-hoc Tukey's correction for multiple comparisons; **** $P < 0.0001$ on day 22 for all groups compared to PBS. Endpoint criteria of 1500 mm³ tumor volume with P value was determined by log-rank test; **** $P < 0.0001$ compared to PBS control.

However, the Reviewer is correct that single-agent complete remission was not observed in the B16.F10 tumor model, a notoriously poorly immunogenic tumor model. For example, in their seminal study on immunotherapy combinations, Irvine and co-workers demonstrated that four therapeutics acting through complementary mechanisms were required to achieve near complete remission in mice with established B16.F10 tumors (1). Therefore, it would be anticipated that combination immunotherapy would be necessary to achieve complete and durable responses in some tumor models, including B16.F10 melanoma. There is also growing consensus within the clinical immuno-oncology community that optimal responses to immunotherapy will require combination regimens that target complementary mechanisms of resistance (2-4), and clinical trials to evaluate new immunotherapy agents are often performed in combination with currently approved immunotherapies (e.g., immune checkpoint inhibitors). Therefore, while the development of a single agent immunotherapy that promotes complete remission in poorly immunogenic tumors remains the holy grail for the field, this may ultimately not be possible for all cancers or patient populations who would benefit from properly designed and implemented combination immunoregimens, potentially including those that integrate STING agonists.

Technical criticisms:

P236-240: the conclusion that the albumin binding nanobody introduces the tumor accumulation is not correct. As a proof for this conclusion is figure 2G, where molecules without life extension module such as nEGFR are compared with the life-time extender nAlb. The difference in the signal is due to the life-time extender not to tumor accumulation. Tumor accumulation means more nAlb in a tumor than in other organs of the same mouse. This is also shown in figure 2G. However, the method to quantify fluorescently labeled proteins is poorly described but this seems to be done on section, per surface. This does not take into account the amount of cells per volume. In other words, the quantity of the therapeutic molecule should be presented as the percentage of injected dose per gr of tissue, a standard way of investigating the presence of a therapeutic protein in tissue.

We appreciate the Reviewer's comment, and we agree that tumor accumulation should be more rigorously determined by quantifying the percent of the injected dose per gram of tissue (%ID/g) for the tumor and other major organs. In response, we have quantified the mass of Cy5 that accumulates in the tumor, kidney, liver, spleen, lungs, and heart 24h following intravenous injection of nAlb-Cy5. Consistent with data quantifying tissue fluorescence using IVIS imaging, we found that nAlb-Cy5 accumulates in tumor tissue to a greater extent than other organs, with ~11% ID/g in orthotopic EMT6 breast tumors. This is significantly higher than the amount in the kidney (~5% ID/g), which is the organ with the second highest amount of Cy5 present, which is consistent with renal clearance of nanobodies and/or liberated Cy5. We also evaluated tumor accumulation of Cy5 following administration of nAlb-Cy5 to mice with subcutaneous B16.F10 tumors and again observed increased accumulation in tumors (~8% ID/g) over other tissues. We have included this data in a new figure, **Figure 2h,i**, and have also edited the text on **Page 9** as followed:

Page 9: "At 24 hours post-administration, mice were euthanized, and major organs and tumors were imaged with an *in vivo* imaging system (IVIS) instrument to evaluate Cy5 biodistribution (**Fig. 2f,g**) and tissue was homogenized for quantification of Cy5 using fluorescent spectroscopy (**Fig. 2h**). We observed minimal Cy5 fluorescence in major organs for both nEGFR-Cy5 and nAlb-Cy5 conjugates, but substantial tumor accumulation of only the nAlb-Cy5 conjugate, corresponding to ~11% injected dose/gram tissue, significantly higher than other organs; similar findings were observed in a B16.F10 tumor model (**Fig. 2i**)."

Figure 2: (h-i) Quantification of percent injected dose per gram of tissue (% ID/g) 24 h following intravenous administration of vehicle (PBS) and nAlb-Cy5 at 2 mg/kg to **(h)** female Balb/c mice with orthotopic EMT6 breast tumors (n=5) and **(i)** female C57BL/6 mice with subcutaneous B16.F10 tumors (n=5). *P* values determined by two-way ANOVA with post-hoc Tukey's correction for multiple comparisons; **P*≤0.05, ***P*≤0.01, ****P*≤0.001, and *****P*<0.0001.

As we further elaborate upon in the text on **Page 7** and below, albumin and albumin binders have been demonstrated to accumulate in tumor tissue through a variety of mechanisms including passive diffusion through leaky tumor vasculature (i.e., EPR effect), active accumulation via transcytosis, albumin endocytosis and catabolism by cancer and immune cells, and tumor retention via binding to SPARC secreted by cancer and/or immune cells (5, 6) (indeed, we found that SPARC was expressed in tumors; see **Page 9** and **Figure S11** below). As is the case for many nanomedicines and macromolecular therapeutics, half-life extension plays an important role in increasing tumor accumulation through both passive and active transport mechanisms. Therefore, we agree with the reviewer that the increased half-life afforded through use of albumin binding is important to the efficacy observed as this contributes to tumor accumulation and increased STING activation in the TME. We thank the reviewer for raising this point as we believe that more the rigorous quantification of nAlb-Cy5 accumulation has strengthened the work and reinforced our finding that albumin-hitchhiking can improve STING agonist delivery to tumor sites.

Page 7: While still incompletely understood and variable across cancers and patients²⁹, albumin can accumulate in tumor tissue through several interrelated mechanisms, including passive diffusion through leaky tumor vasculature, active transport via endothelial cell transcytosis, binding to SPARC (secreted protein acidic and rich in cysteine) produced by cancer cells, and cellular uptake and catabolism by cancer and tumor-associated immune cells such as macrophages.^{27, 29} Albumin has been reported to enter cancer cells and tumor-associated myeloid cells (e.g., macrophages) through both albumin-dependent, receptor-mediated pathways as well as by micropinocytosis.^{27, 29}

Page 9: “Albumin binding to SPARC expressed in tumor tissue has also been implicated in increased accumulation of albumin-binding therapeutics,²⁷ and we found that SPARC is expressed in both EMT6 and B16.F10 tumors (**Fig. S11**) and may therefore contribute to nAlb accumulation.”

Figure S11: (a) Western blot analysis of SPARC (42 kDa) and HSP-90 (90 kDa) proteins in B16.F10 and EMT6 tissue. **(b)** Representative fluorescent microscopy images of tumor sections stained for DAPI (blue), CD31 (magenta), and SPARC (white) 24 h following administration of PBS, nAlb-Cy5, or nAlb-Cy5 + nAlb-diABZI (scale bar: 100 μm).

Minor points:

P155: is the diABZI compound membrane permeable as this would be required for endosomal escape?

The diABZI compound is membrane permeable and this has been described previously (7). By contrast, nAlb-Cy5 uptake is inhibited 4°C as shown in **Figures 2a and Extended Data Fig. 1e-g** and is therefore internalized via active endocytosis process. In response to other reviewer comments, we have also demonstrated that nAlb-diABZI is trafficked to endolysosomes (**Figure 2c,d and Extended Data Fig. 1i**) and we have also identified the product that is released from the nanobody following internalization and endolysosomal degradation (see **Figure S6,7**) and found that this compound had similar activity to diABZI *in vitro* (**Figures S10**). Finally, to assess the unlikely possibility that nAlb-diABZI might be capable of mediating active endosomal escape (i.e., by disrupting endosomal membranes), we used galectin 8 (Gal8) reporter cell assay that we and others have used to evaluate endosomal escape induced by nanoparticles (8-10). In brief, Gal8-GFP is a fusion protein of Gal8, a protein that binds glycans, and the fluorescent GFP protein. When the endosome of a cell is disrupted, Gal8-GFP redistributes from the cytosol to the ruptured endosomes, where it binds the newly exposed glycans. Following treatment with an endosomolytic agent (e.g., a lipid nanoparticle), the number of GFP puncta can be counted as a measurement of disrupted endosomes and a metric of endosomal escape. Using this assay, we did not observe any evidence of endosomal membrane destabilization induced by nAlb-diABZI and we have included this data in **Extended Data Fig. 1h**. Taken together, this data indicates that nAlb-diABZI, but not free diABZI, enters cells via endocytosis and is degraded in lysosomes to release a membrane-permeable diABZI variant that can passively cross the endo/lysosomal membrane to access STING. This is described in revised text on **Pages 7 and 8** as followed:

Pages 7 and 8: “First, we first validated that intracellular uptake of nAlb-Cy5 was abrogated at 4°C indicating an active endocytotic mechanism (**Fig. 2a, Extended Data Fig. 1**); by contrast diABZI can enter cells by passive transport across the plasma membrane.²³ We next assessed if albumin binding enhanced nAlb internalization by EMT6 and myeloid cells. To test this, we first used flow cytometry compare the cellular uptake of nAlb-Cy5 to a negative control nanobody targeting GFP, nGFP-Cy5 (**Fig. S5**), in serum containing media, finding insignificant or minor differences in cellular uptake between nAlb-Cy5 and nGFP-Cy5 (**Extended Data Fig. 1**). While eliminating serum from culture media decreased nAlb-Cy5 uptake, this occurred to the same extent for nGFP-Cy5, again indicating that cellular uptake occurs predominantly in an albumin receptor-independent manner in these cell types. Albumin can also be internalized by cancer and tumor-associated immune cell populations through non-receptor-mediated micropinocytosis. To evaluate this, we inhibited micropinocytosis in EMT6 cells, RAW264.7 macrophages, and BMDMs using 5-[N-ethyl-N-isopropyl] amiloride (EIPA), which significantly reduced nAlb-Cy5 uptake (**Fig. 2b**). Given that macropinosomes often traffic to lysosomes, we next assessed colocalization of nAlb-Cy5 with lysotracker and found that a substantial and similar fraction (>50%) of nAlb-Cy5 and nGFP-Cy5 was colocalized with lysosomes or late endosomes in EMT6 and RAW264.7 cells (**Fig. 2c,d**). As expected, nAlb-diABZI did not mediate endosomal disruption as assessed using a previously described galectin 8 (Gal8) endosomal recruitment assay (**Extended Data Fig. 1**)⁴¹.

To gain insight into how amide-linked diABZI is released from the nanobody upon cellular internalization, we incubated nAlb-diABZI with lysosomes isolated from rat liver (Tritosomes), which are used to investigate stability and catabolism of molecules trafficked to an endosome-lysosome pathway, and used MALDI mass spectroscopy to assess the emergence of a PEGylated diABZI adduct that would be predicted due to amide bond cleavage by lysosomal proteases (**Fig. S6**). We observed the presence of this peak as early as 1 h following incubation with Tritosomes, suggesting that a fraction of nAlb-diABZI is lysosomally degraded to release a PEGylated diABZI variant. We synthesized this compound (**Fig. S7-9**) and evaluated *in vitro* activity in THP1 IFN-I reporter cells, finding that it had a similar EC50 value to the previously described diABZI molecule, which can enter cells through passive transport²³ (**Fig. S10**).”

Figure 2. (a) Dose-response curve for nanobody-Cy5 conjugate surface binding and intracellular uptake at 37 °C and 4 °C measured by flow cytometry in EGFR⁻ (THP-1) *in vitro*. (b) Uptake of nAlb-Cy5 (2 μM) in RAW 264.7, EMT6, and BMDM cells with the addition of control PBS (-EIPA) or macropinocytosis inhibitor (+EIPA). (c) Colocalization of Cy5 (red) with lysotracker green (green) and Hoechst (blue) in RAW 264.7 cells with (d) percent colocalization determination for nAlb-Cy5 and nGFP-Cy5 in RAW 264.7 and EMT6 cells. (scale bar: 100 μm)

Extended Data Figure 1: (e-g) Uptake of nAlb-Cy5 (2 μM) in (e) BMDC cells, (f) BMDM cells, and (g) EMT6 cells at 37 °C and 4 °C. *P* values determined by Student's *t*-test. (h) Integrated pixel intensity of Gal8-GFP puncta, per cell, for DBCO-PEG11-diABZI, nAlb-diABZI, and AP-diABZI at 0.25 μM (mean ± SEM, *n*=3-9). *P* values determined via one-way ANOVA, with Tukey's post-hoc. (i) Colocalization analysis of nAlb-Cy5 (2 μM) and nGFP-Cy5 (2 μM) for Raw 264.7 cells to visualize the cell nuclei (Hoechst, blue), lysosomes (Lysogreen, green), and Cy5 (Cy5, red). Merged images of the stains provided, in addition to a 2.5x zoom of the merged image (white box). (scale bar: 100 μm)

Figure S6: Tritosome degradation assay for **(a)** diABZI-amine, **(b)** DBCO-PEG₁₁-diABZI, and **(c-d)** nAlb-amine with highlighted molecular weight ranges at 780 Da (diABZI-amine construct, blue dashed line), 1688 Da (DBCO-PEG₁₁-diABZI construct, orange dashed line), and 1906 Da (Amine-PEG₃-Triazole-PEG₁₁-diABZI).

Figure S7: Synthesis of Amine-PEG₃-Triazole-PEG₁₁-diABZI.

Figure S10: *In vitro* dose-response curves for THP1-Dual reporter cells comparing diABZI to Amine-PEG₃-Triazole-PEG₁₁-diABZI.

P300: “Based on the long half-life of nAlb and its capacity to dramatically enrich cargo distribution to tumor sites”. What would be the rationale for the ‘dramatic enrichment of cargo in the tumor’?

While our data demonstrate that nAlb accumulates in tumor tissue relative to other organs, we do not feel that this represents a ‘dramatic enrichment’ and so we have revised this sentence on **Page 9** as followed:

Page 9: “Based on the significant tumor accumulation of nAlb-Cy5 we next used flow cytometry to determine which tumor-associated cell populations internalized the conjugate (Fig. 2k,l, Fig. S12).”

We have also expanded our discussion on **Page 7** on the potential mechanism by which albumin has been described to accumulate in tumor tissue, including passive diffusion through leaky tumor vasculature (i.e., EPR effect), active accumulation via transcytosis, tumor retention via binding to SPARC secreted by cancer cells, and albumin endocytosis and catabolism by cancer and immune cells:

Page 7: While still incompletely understood and variable across cancers and patients²⁹, albumin can accumulate in tumor tissue through several interrelated mechanisms, including passive diffusion through leaky tumor vasculature, active transport via endothelial cell transcytosis, binding to SPARC (secreted protein acidic and rich in cysteine) produced by cancer cells, and cellular uptake and catabolism by cancer and tumor-associated immune cells such as macrophages.^{27, 29} Albumin has been reported to enter cancer cells and tumor-associated myeloid cells (e.g., macrophages) through both albumin-dependent, receptor-mediated pathways as well as by micropinocytosis.^{27, 29}

P390: Also for the anti-PD-L1 nanobody the authors seem to ignore that such a nanobody already exists for several years: for instance Zhang et al 2017 doi: 10.1038/celldisc.2017.4 Referring to the literature about these anti-PD-L1 is required.

We appreciate the author’s concern, and we did not intend to ignore that anti-PD-L1 nanobodies have already been described. Indeed, in this work we used a previously described nanobody and we provided the appropriate citation in our original manuscript (reference number 56; Broos *et al.* Oncotarget, 8: 41932, 2017). However, to avoid any confusion regarding the source of this nanobody sequence, we have revised the text on **Page 15** to make it clear that this nanobody has been previously described:

Page 15: “As a translationally-relevant example, we introduced a second previously described nanobody domain that binds to PD-L1 (anti-programmed cell death ligand 1).⁵⁶”

P392: what is the affinity of this anti-PD-L1 nanobody?

The reviewer raises an important question. In response, we have conducted a new experiment to determine the K_D of the anti-PD-L1 nanobody using isothermal calorimetry and have determined an affinity of 12.4 nM for murine PD-1, which is consistent with the previous work describing this nanobody which measured a K_D of 4 nM. This data is included in **Figure S1c**.

Figure S1: Isothermal calorimetry (ITC) for (c) nPD-L1 at pH 7.5 with PD-L1 receptor.

P434: also for the AP nanobodies, tissue presence has not been determined in the correct way.

Using fluorescent spectroscopy, we have quantified the amount of Cy5 in EMT6 tumor tissue 48h following intravenous administration of AP-Cy5, measuring $\sim 2.3\%$ ID/g tissue; this data is included below in **Figure R1**. Unfortunately, we did not save the other organs and therefore are not able to quantify Cy5 content in other tissues

as we did above for nAlb-Cy5. However, we observed good correlation between tissue fluorescence measured by IVIS and quantification of Cy5 content within tissue, and therefore we feel that IVIS data presented in **Figure 4k-m** accurately reflects the relative distribution of AP-Cy5 between tumor and other tissues and therefore supports our conclusion that AP-Cy5 accumulates to a greater degree than nAlb-Cy5 or nPD-L1-Cy5.

Figure R1: Quantification of injected percent dose per gram of tumor 48 h following intravenous administration of AP-Cy5 at 2 mg/kg to female Balb/c mice with orthotopic EMT6 breast tumors (n=5).

We have also included this quantification of AP-Cy5 accumulation in the text on **Page 18**:

Page 18: “Further, while the nPD-L1-Cy5 conjugate was observed at similarly low levels in major organs (liver and kidneys) and the tumor at 48 h (**Fig. 4j**), the AP-Cy5 conjugate demonstrated significant tumor accumulation (corresponding to 2.19 ± 0.43 %ID/gram tumor) relative to major organs (**Fig. 4k**) and significant increase over nAlb alone (**Fig. 4l-m**).”

Reviewer #2 (Report for the authors (Required)):

This work designed a modular, safe, and efficient nanobody delivery platform based on synthetic biology technology. By hitchhiking on albumin, it achieves efficient tumor penetration of STING agonists, which can effectively increase the infiltration of NK cells and T cells and produce excellent therapeutic effects in multiple tumor models. At the same time, by integrating a PD-L1 binding domain to create a bi-valent fusion protein, the tumor-targeting ability and anti-tumor therapeutic effect of the treatment system are significantly enhanced, showing satisfactory therapeutic effects in preclinical tumor treatment models. This work provides a programmable, multifunctional, and safe delivery platform for immunotherapy. However, we believe that this work can be further improved.

Here are some other issues that need to be addressed:

1. It's well accepted that MDSCs are immunosuppressive cells. However, as shown in Fig 2j and Fig 2k, nAlb-diABZI resulted in an increased frequency of MDSCs. What factors may be responsible for this phenomenon and what impact it will have on subsequent immunotherapy ?

The reviewer raises an important point as MDSCs have been implicated as a major resistance mechanism to immunotherapy. Indeed, previous reports have shown that STING activation can increase MDSC infiltration into tumors and may also contribute to increased metastasis in some models (11-14). To determine if MDSCs contributed to resistance to nAlb-diABZI or AP-diABZI we performed two additional experiments in the EMT6 breast cancer model. First, we evaluated nAlb-diABZI and AP-diABZI in combination with orally administered SX-682, a CXCR1/2 inhibitor that blocks MDSC recruitment to tumors that is being evaluated in clinical trials (**Figure S18**). Second, we administered anti-Gr1 antibodies to deplete MDSCs (**Figure S19**). Surprisingly, in both studies, we observed a modest *reduction* in nAlb-diABZI and AP-diABZI efficacy, indicating that MDSCs do not strongly contribute to resistance and may even play a small role in treatment efficacy. While this was unexpected, similar negative effects of MDSC inhibition on antitumor responses have been observed by others (15) and may reflect the complex roles that MDSCs can have in response to immunotherapy. This data is summarized on **Pages 15 and 22** as followed:

Page 15: “In particular, MDSCs have been reported to reduce the efficacy of some STING-activating therapies⁴⁷⁻⁴⁹ and we therefore evaluated nAlb-diABZI in combination with orally administered SX-682, which inhibits CXCR1/2 chemokine receptors involved in MDSC recruitment⁵⁰, but, surprisingly, found that SX-682 tended to reduce nAlb-diABZI efficacy (Fig. S18). We also used anti-Gr1 antibodies to deplete MDSCs (primarily gMDSCs)⁵¹ and again found a modest reduction in nAlb-diABZI efficacy (Fig. S19).”

Page 22: “However, as observed with nAlb-diABZI treatment, inhibition of MDSCs using SX-682 or anti-GR1 antibody depletion reduced AP-diABZI treatment efficacy (Fig. S18, S19, S27).”

Figure S18: (a) Schematic of EMT6 tumor inoculation and treatment schedule for PBS, nAlb-diABZI (1.25 µg) and AP-diABZI (1.25 µg) during MDSC inhibitor treatment using SX-682 compared to regular chow. (b) Tumor growth curves, (c) Kaplan-Meier survival plots (n=8-9). All doses were injected I.V. at the labeled amount of diABZI per injection. (c) Kaplan-Meier survival curves of mice treated with indicated formulation using 1500 mm³ tumor volume as endpoint criteria with *P* value was determined by log-rank test.

Figure S19: (a) Schematic of EMT6 tumor inoculation and treatment schedule for PBS, nAlb-diABZI (1.25 µg), and for AP-diABZI (1.25 µg) with anti-GR-1 MDSC depletion. (b) Tumor growth curves, (c) Kaplan-Meier survival plots (n=8). All doses were injected I.V. at the labeled amount of diABZI per injection. (c) Kaplan-Meier survival curves of mice treated with indicated formulation using 1500 mm³ tumor volume as endpoint criteria with *P* value was determined by log-rank test.

2. As shown in Fig 4f, nPD-L1-diABZI, nAlb-diABZI and AP-diABZI could efficiently activate the STING pathway. However, the relative expression of *Ifnb* and *Cxcl10* with the treatment of AP-diABZI, which are representative downstream immune responses of STING activation, are not better than those of nAlb-diABZI. Additionally, the difference in immune response between AP-diABZI and nPD-L1-diABZI on BMDC (Fig S12) should be discussed in the manuscript.

The reviewer raises an excellent point. The kinetics of gene expression in response to STING activation are tightly and differentially regulated and therefore apparent differences in the magnitude of response as measured by gene expression *in vitro* may also reflect the time point selected for analysis. To account for this possibility, we sought to establish a better understanding of how nanobody diABZI conjugates impacted STING-mediated gene expression as a function of time. We therefore conducted a new experiment in which we treated bone-marrow derived macrophages (BMDM) with nanobody-diABZI conjugates, and evaluated gene expression levels of *Ifnb1*, *Tnfa*, and *Cxcl10* at 0, 0.5, 1, 2, 3, 4, 6, 8, and 24 h. As shown in revised Figure 4f, all nanobody-diABZI conjugates (nPD-L1-diABZI, nAlb-diABZI, AP-diABZI) were more active than the DBCO-PEG-diABZI parent compound, which, like commercially available diABZI, we have consistently found to be poorly active in BMDMs. We found all conjugates to increase expression of *Ifnb1*, *Tnfa*, and *Cxcl10* with similar kinetics. Peak gene expression occurred around 6 h, where we observed a modest increase in gene expression induced by nAlb-diABZI relative to variants containing a PD-L1 binding domain, potentially reflecting different internalization kinetics or pathways that may impact diABZI release. We also re-evaluated nAlb-diABZI and AP-diABZI in

BMDCs as a function of time and again found that both similarly active with respect to the magnitude and kinetics of gene expression (**Figure S22**). We have also revised the text on **Page 16** as followed:

Page 16: “The *in vitro* activity of nPD-L1-diABZI and AP-diABZI was tested in A549-Dual and THP1-Dual IFN- β reporter cells (**Fig. 4d,e**) and by qPCR for analysis of STING-associated cytokines/chemokine gene expression in primary BMDMs and BMDCs (**Fig. 4f, Fig. S22**). We found that all nanobody-diABZI conjugates were potentially active in both reporter cell lines without evidence of cytotoxicity (**Fig. S23**) and that nanobody-diABZI conjugates were more active than the parent DBCO-diABZI in BMDMs and triggered STING-associated gene expression with similar kinetics (**Fig. 4f**); both nAlb-diABZI and AP-diABZI were also active in murine bone marrow-derived dendritic cells (BMDC; **Fig. S22**).”

Figure 4: (f) qPCR analysis of genes associated with STING activation in bone marrow derived macrophages (BMDMs) in response to treatment at discrete time points (0, 0.5, 1, 2, 3, 4, 6, 8, 24 h) with indicated agonist at 0.25 μ M (n=3). *P* values determined by one-way ANOVA with post-hoc Tukey’s correction for multiple comparisons; **P*≤0.05, ***P*≤0.01, ****P*≤0.001, and *****P*<0.0001 compared to PBS control.

Figure S22: *In vitro* qPCR analysis of genes associated with STING activation in bone marrow derived dendritic cells (BMDCs) as analyzed (**a-c**) over 24 h or (**d-f**) at 6 h post-treatment with nanobody-diABZI conjugates at 0.25 μ M (n=5-6). *P* values determined by one-way ANOVA with post-hoc Tukey’s correction for multiple comparisons; ***P*≤0.01 and *****P*<0.0001 compared to PBS control.

3. Mice treated with AP-diABZI and nAlb-diABZI+ICB exhibited significant antitumor responses in the EMT6 tumor-bearing model (Fig. 5b-f). However, as shown in Fig 5g-l, these assays did not include mice treated with nAlb-diABZI+ICB, making it difficult to understand the different mechanisms of the increased efficacy of AP-diABZI and nAlb-diABZI+ICB. Further discussion may be beneficial.

The reviewer raises an interesting point. First, regarding serum cytokine data that was presented in original Figure 5g, we did collect data for the nAlb-diABZI + ICB group and we now include this in **Figure S25**. We observed only modest differences in serum cytokine levels induced by AP-diABZI and nAlb-diABZI+ICB, with slight decreases in the levels of CXCL1 (*P*<0.05), CXCL10 (*P*<0.05), IFN- β (*P*<0.05) and no changes observed

for any other cytokine analyzed. We postulate that these small changes are likely a consequence of the different pharmacokinetics of AP-diABZI and nAlb-diABZI that effect serum cytokine concentrations as a function of time. Based on the similar cytokine profile and antitumor efficacy observed between AP-diABZI and nAlb-diABZI + ICB, we elected not to perform costly NanoString gene expression profiling for the nAlb-diABZI + ICB group in our original studies. However, to respond to the reviewer, we repeated these experiments to compare AP-diABZI with nAlb-diABZI + ICB. Unfortunately, a technical challenge during mRNA extraction rendered several of the AP-diABZI samples unsuitable for analysis and therefore we are not able to provide a direct comparison between AP-diABZI and nAlb-diABZI + ICB. However, the mRNA quality from the nAlb-diABZI + ICB and PBS tumors was suitable for analysis, and we have included this data in **Figure S26**. Note that similar fold changes in the Z-score of gene expression signatures relative to PBS was observed for both AP-diABZI and nAlb-diABZI + ICB, suggesting that both treatments induce similar changes to the tumor immune microenvironment. Given the similar patterns of cytokine production, gene expression, and antitumor efficacy observed between AP-diABZI and nAlb-diABZI + ICB we do not feel that adding additional discussion regarding small differences between these two groups will strengthen the primary conclusions of the manuscript. We hope that the reviewer understands our position not to further elaborate upon this point in the text, but we appreciate the reviewer bringing this to our attention as it compelled us to include cytokine and gene expression data related to nAlb-diABZI + ICB in the Supplemental Information.

Figure S25: EMT6 Serum cytokine/chemokine concentrations 4 h post-injection with indicated nanobody-diABZI conjugate (n=6-10). *P* values determined by one-way ANOVA with post-hoc Tukey's correction for multiple comparisons; **P*≤0.05, ***P*≤0.01, ****P*≤0.001, and *****P*<0.0001 compared to PBS control.

Figure S26: (a) Volcano plots representing significance ($-\log_{10}$) and fold change (\log_2) for gene expression analysis in nAlb-diABZI + ICB vs. PBS ($n=3$) (c-d) Heat maps of NanoString gene cluster matrices showing Z score fold changes for (b) functional gene annotations, (c) biological signatures, and (d) cell types. All results are the mean \pm SEM.

4. Mice treated with AP-diABZI showed significant antitumor responses in the B16.F10 therapeutic model (Fig.7). However, as shown in Fig. 7e, the production of cytokines associated with antitumor immunity was higher in mice treated with nAlb-diABZI group compared to mice treated with nAlb-diABZI+ICB group. Further discussion may be beneficial.

We thank the reviewer for raising this point. First, in response to a reviewer comment to simplify figures, we have moved data in original **Figure 7e** to **Extended Data Fig. 8** and have also represented the data as bar graphs with statistical comparisons between the different groups. In comparing serum cytokine levels induced by nAlb-diABZI to nAlb-diABZI + ICB using a one-way ANOVA with Tukey's test for multiple comparisons, there are only three cytokines that are significantly different between the groups: IFN α ($P<0.05$), IL6 ($P<0.01$), and IL12p70 ($P<0.05$). We believe that our original presentation of the data as a heat map with only statistical comparisons to vehicle control presented caused the small differences between these groups to appear more substantial than they are. While it is conceivable that PD-L1 inhibition may influence the acute cytokine response induced by nAlb-diABZI through both direct (e.g., by modulating immunocellular composition/phenotype of tissues and therefore sensitivity to STING agonists) or indirect (e.g., by stimulating cytokines that modulate the response to STING agonists), we do not believe that adding discussion about these modest differences will strengthen the primary conclusions of the manuscript, which is already exceeds journal guidelines in length. We hope that the reviewer understands our position not to further elaborate upon this point in the text, but we appreciate the reviewer bringing this to our attention as it compelled us to analyze the data more closely and to present all the data and statistical comparisons in the Extended Data section.

Extended Data Figure 8: Serum cytokine concentration in B16.F10 tumor bearing C57BL/6 female mice 4 hours after the first treatment (n=6-10) as **(a)** heat maps and **(b)** bar plots. *P* values determined by one-way ANOVA with post-hoc Tukey's correction for multiple comparisons; **P*≤0.05, ***P*≤0.01, ****P*≤0.001, and *****P*<0.0001 compared to the PBS control.

5. Mice showed weight loss after the first injection of nAlb-diABZI (Fig. S8 and Extended Data Fig. 1). How about the weight changes in mice during treatment in other models (Fig. 3, Fig. 5., Fig. 8, Fig. S7).

We have included weight loss curves for all experiments and models performed in this work as a new **Figure S15**:

Figure S15: Mouse weight loss curves during treatment course with diABZI and diABZI conjugated proteins from indicated studies as represented by subfigure titles.

6. As shown in Extended Data Fig. 5, a single dose of AP-diABZI treatment resulted in the presence of more g-MDSCs than m-MDSCs when compared to the PBS and nAlb-diABZI groups. What are the possible mechanisms that lead to this response?

We thank the reviewer for this interesting question. Granulocytic MDSCs (g-MDSCs) arise primarily from the same granulocyte myeloid cell lineage as neutrophils and are distinguished by their immunosuppressive function that can occur in response to prolonged exposure to cytokines such as GM-CSF, IL-6, and IL-1 β , which may be activated in response to STING activation (16). Indeed, previous reports have shown that STING activation can increase MDSC infiltration into tumors (17). As shown in what is now **Extended Data Figure 6**, treatment of nAlb-diABZI and AP-diABZI results in a significant infiltration/expansion of CD11b⁺Ly6G⁺Ly6C^{mid}SSC^{hi} neutrophils that arise in response to the inflammatory cues induced by STING activation, and it is likely that the g-MDSCs arise from this population in the tumor tissue. Hence, we believe that the increased abundance of g-MDSCs over m-MDSCs primarily reflects the dramatic increase in tumor-associated neutrophils over tumor-associated monocytes/macrophages that results from STING activation in the TME in response to nAlb-diABZI and AP-diABZI. However, it should also be noted that it is challenging to phenotypically distinguish neutrophils and g-MDSCs using surface markers and so side scatter (SSC) is often used to differentiate based on granularity. Therefore, the ultimate test of whether cells are MDSCs is whether they exert immunosuppressive functions. To further address this and to further understand the role of MDSCs in response to nAlb-diABZI or AP-diABZI, we performed two additional experiments in the EMT6 breast cancer model. First, we evaluated nAlb-diABZI and AP-diABZI in combination with orally administered SX-682, a CXCR1/2 inhibitor that blocks MDSC recruitment to tumors that is being evaluated in clinical trials. Second, we administered anti-Gr1 antibodies to deplete MDSCs. Surprisingly, in both studies, we observed a modest *reduction* in nAlb-diABZI and AP-diABZI efficacy (**Figure S18-19**), indicating that MDSCs do not strongly contribute to resistance and may even play a small role in treatment efficacy. While this was unexpected, similar negative effects of MDSC inhibition on antitumor responses have been observed by others (15) and may reflect the complex roles that MDSCs can have in response to immunotherapy. This data is summarized on **Pages 15 and 22** as followed:

Page 15: “In particular, MDSCs have been reported to reduce the efficacy of some STING-activating therapies⁴⁷⁻⁴⁹ and we therefore evaluated nAlb-diABZI in combination with orally administered SX-682, which inhibits CXCR1/2 chemokine receptors involved in MDSC recruitment⁵⁰, but, surprisingly, found that SX-682 tended to reduce nAlb-diABZI efficacy (**Fig. S18**). We also used anti-Gr1 antibodies to deplete MDSCs (primarily gMDSCs)⁵¹ and again found a modest reduction in nAlb-diABZI efficacy (**Fig. S19**).”

Page 22: “However, as observed with nAlb-diABZI treatment, inhibition of MDSCs using SX-682 or anti-GR1 antibody depletion reduced AP-diABZI treatment efficacy (**Fig. S18, S19, S27**).”

Extended Data Figure 6: Flow cytometric analysis of EMT6 tumor bearing female Balb/c mice 48 h following a single dose of nAlb-diABZI (1.25 μ g, n=8), AP-diABZI (1.25 μ g, n=8), or PBS (n=7). **(a)** tSNE plots of live cells in EMT6 tumors, colored by cell population with relative expression level of Ki67, CD69, PD-1, and PD-L1 as indicated on heat map. DC: dendritic cell; M ϕ : macrophage; NK: natural killer cell; MDSC: myeloid-derived suppressor cell. **(b)** Analysis of the frequency of live CD45⁺ cells and frequency of PD-L1 and Ki67 expressing CD45⁺ cells in the tumor. **(c-d)** Heat maps summarizing the fold change in the percentage of **(c)** indicated cell population and **(d)** frequency of NK cells, CD8⁺ T cells, and CD4⁺ T cells expressing the indicated marker in EMT6 tumors. **(B-D)** *P* values determined by one-way ANOVA with post-hoc Tukey's correction for multiple comparisons to PBS; **P*≤0.05, ***P*≤0.01, ****P*≤0.001, and *****P*<0.0001 compared to the PBS control. **(e-f)** Representative flow cytometry dot plots summarizing the distribution of Ki67⁺ to CD69⁺ and PD-1⁺ CD8⁺ T cells and CD4⁺ T cells in EMT6 tumors. All results are the mean ± SEM.

Figure S18: **(a)** Schematic of EMT6 tumor inoculation and treatment schedule for PBS, nAlb-diABZI (1.25 μ g) and AP-diABZI (1.25 μ g) during MDSC inhibitor treatment using SX-682 compared to regular chow. **(b)** Tumor growth curves, **(c)** Kaplan-Meier survival plots (n=8-9). All doses were injected I.V. at the labeled amount of diABZI per injection. **(c)** Kaplan-Meier survival curves of mice treated with indicated formulation using 1500 mm³ tumor volume as endpoint criteria with *P* value was determined by log-rank test.

Figure S19: (a) Schematic of EMT6 tumor inoculation and treatment schedule for PBS, nAlb-diABZI (1.25 μg), and for AP-diABZI (1.25 μg) with anti-GR-1 MDSC depletion. **(b)** Tumor growth curves, **(c)** Kaplan-Meier survival plots (n=8). All doses were injected I.V. at the labeled amount of diABZI per injection. **(c)** Kaplan-Meier survival curves of mice treated with indicated formulation using 1500 mm³ tumor volume as endpoint criteria with *P* value was determined by log-rank test.

Reviewer #3 (Report for the authors (Required)):

In this manuscript, Wilson and co-workers developed a type of albumin-hitchhiking drug delivery platform for the systemic administration of STING agonists for cancer immunotherapy. STING agonist diABZI was covalently conjugated with anti-albumin nanobody to realize the albumin-hitchhiking pharmacokinetics properties and increase the tumor tropism, and the programmability of the platform allow the integration of a second functional nanobody domain (like anti-PD-L1). Though the concept of albumin-hitchhiking is not novel in anti-tumor drug delivery field, taking the commercial albumin-bound paclitaxel as example, the special features of this work are that the drugs conjugated were immune agonists rather than chemotherapeutic agents, and that the anti-albumin nanobody hitchhiking strategy was used instead of direct albumin conjugation. Moreover, the designed drug has a single component, well-defined function, and impressive therapeutic outcomes in multiple mouse tumor models, which illustrate its potential for clinical application and translation. In summary, this work is significant in providing a promising multimodality programmable delivery platform technology for systemic delivery of immune agonists. However, some minor writing and experimental details remain to be refined before accepted for publication.

1. One of the most important issues for systemic administration of immune agonists is the off-target toxicity. Although nAlb-diABZI could target albumin and tend to the tumor site, it would inevitably affect other organs as well. The current manuscript lacks necessary evaluation on this point in various organs after administration.

The reviewer raises an important point; therefore, we also performed a comprehensive preclinical toxicity analysis through evaluation of blood chemistry and organ pathology (**Extended Data Figure 2**). While we did not observe any evidence of organ toxicity, including liver toxicity, we nonetheless repeated this study and directly involved experienced veterinary pathologists on collection and processing of blood for analysis. Consistent with our previous results, we observed insignificant or minor differences in all blood biochemistry markers (e.g., AST, AST) (**Extended Data Figure 2**). Additionally, as described in our original manuscript, all organs were carefully examined by a blinded, board-certified veterinary pathologist who did not find any clinically significant differences between untreated control mice and mice treated three times with the 1.25 μg dose. Because of this, we observe that nAlb-diABZI and AP-diABZI are well-tolerated at our maximal dose of 1.25 μg, which we have also demonstrated can be reduced while still achieving antitumor efficacy.

Extended Data Figure 2: (a) Scheme for treating healthy C57BL/6 female mice with PBS, 1.25 μ g nAlb-diABZI, or 1.25 μ g AP-diABZI. (b) Body weight change of mice during treatments. (c) Quantification of plasma cytokines 4 hours after the first treatment and 24 hours after the final treatment ($n=3-5$, $*P\leq 0.05$, $**P\leq 0.001$, $****P<0.0001$ by one-way ANOVA compared to PBS). (d) Blood biochemistry and (e) immunohistochemistry analysis of healthy C57BL/6 mice. After 3 treatments, mice were euthanized and blood samples were collected to determine changes in RBCs, white blood cells (WBCs), neutrophils, platelets, and lymphocytes. Serum samples were also used to analyze liver and kidney function, by measuring changes in ALT/AST, blood urea nitrogen (BUN), and creatinine. Tissue was sectioned, H&E stained, and imaged. Images use a scale bar of 200 μ m. No significant changes were observed. ($*P\leq 0.05$, $**P\leq 0.01$, and $***P\leq 0.001$ compared to PBS by one-way ANOVA).

2. The schematic of figure 2a speculated the pathway of nAlb-diABZI after entry into cells, but no specific experimental results are available for verification. Experiments like cell confocal assays should be supplemented to illustrate the fate of drug in early and late endosome after entry, as well as to demonstrate that STING agonist molecules can escape from endosomes to cytoplasm for STING activation.

We agree with the reviewer that the schematic in our original **Figure 2a** was overly speculative, and we have revised the manuscript in several ways to address this. First, we have eliminated this schematic from **Figure 2** since dissecting cellular internalization pathways of nAlb-diABZI, which likely also varies between cell types, particularly *in vivo*, is not a primary focus of this work. Second, in response to other reviewer comments, we have conducted new experiments with improved controls that indicate that nAlb internalization is not dependent on albumin-receptor mediated endocytosis and is instead driven primarily by micropinocytosis (**Figure 2b**, **Extended Data Fig. 1**). Third, we conducted additional experiments to further understand the mechanism by which nAlb-diABZI is processed intracellularly to promote STING activation. Using fluorescent microscopy, we demonstrate that nAlb-Cy5 colocalizes with lysosomes (Mander's overlap coefficient >0.5) as measured using a LysoTracker dye (**Figure 2c,d**, **Extended Data Fig. 1**). Based on this, we next incubated nAlb-diABZI with rat liver tritosomes, which are isolated and stabilized lysosomes and employed to evaluate the stability of protein-drug conjugates, and we used MALDI mass spectroscopy to evaluate the emergence of potential

cleavage products (**Figure S6**). As early as 1 h following incubation with tritosomes, we observed the formation of an NH₂-PEG₃-triazole-DBCO-PEG₁₁-diABZI product (**Figure S7**) corresponding to cleavage of the amide bond ligating the diABZI molecule to the nanobody. We then synthesized and purified this product and tested *in vitro* activity compared to the previously reported diABZI compound in THP1 IFN-I reporter cells and found that the released compound was a highly potent STING agonist with comparable activity to diABZI (**Figure S10**). Finally, to assess the possibility that nAlb-diABZI might be capable of mediating endosomal escape, we used galectin 8 (Gal8) reporter cell assay that we and others have used to evaluate endosomal escape induced by nanoparticles (8-10). In brief, Gal9-GFP is a fusion protein of Gal9, a protein that binds glycans, and the fluorescent GFP protein. When the endosome of a cell is disrupted, Gal9-GFP redistributes from the cytosol to the ruptured endosomes, where it binds the newly exposed glycans. Following treatment with an endosomolytic agent (e.g., a lipid nanoparticle), the number of GFP puncta can be counted as a measurement of disrupted endosomes and a metric of endosomal escape. Using this assay, we did not observe any evidence of endosomal escape induced by nAlb-diABZI and we have included this data in **Extended Data Figure 1h**. Taken together, this data indicates that nAlb-diABZI, but not free diABZI, enters cells via endocytosis and is degraded in lysosomes to release a membrane-permeable diABZI variant that can passively cross the endo/lysosomal membrane to access STING. This is described in revised text on **Pages 7 and 8** as followed:

Pages 7 and 8: “First, we first validated that intracellular uptake of nAlb-Cy5 was abrogated at 4°C indicating an active endocytotic mechanism (**Fig. 2a, Extended Data Fig. 1**); by contrast diABZI can enter cells by passive transport across the plasma membrane.²³ We next assessed if albumin binding enhanced nAlb internalization by EMT6 and myeloid cells. To test this, we first used flow cytometry compare the cellular uptake of nAlb-Cy5 to a negative control nanobody targeting GFP, nGFP-Cy5 (**Fig. S5**), in serum containing media, finding insignificant or minor differences in cellular uptake between nAlb-Cy5 and nGFP-Cy5 (**Extended Data Fig. 1**). While eliminating serum from culture media decreased nAlb-Cy5 uptake, this occurred to the same extent for nGFP-Cy5, again indicating that cellular uptake occurs predominantly in an albumin receptor-independent manner in these cell types. Albumin can also be internalized by cancer and tumor-associated immune cell populations through non-receptor-mediated micropinocytosis. To evaluate this, we inhibited micropinocytosis in EMT6 cells, RAW264.7 macrophages, and BMDMs using 5-[N-ethyl-N-isopropyl] amiloride (EIPA), which significantly reduced nAlb-Cy5 uptake (**Fig. 2b**). Given that macropinosomes often traffic to lysosomes, we next assessed colocalization of nAlb-Cy5 with lysotracker and found that a substantial and similar fraction (>50%) of nAlb-Cy5 and nGFP-Cy5 was colocalized with lysosomes or late endosomes in EMT6 and RAW264.7 cells (**Fig. 2c,d**). As expected, nAlb-diABZI did not mediate endosomal disruption as assessed using a previously described galectin 8 (Gal8) endosomal recruitment assay (**Extended Data Fig. 1**)⁴¹.

To gain insight into how amide-linked diABZI is released from the nanobody upon cellular internalization, we incubated nAlb-diABZI with lysosomes isolated from rat liver (Tritosomes), which are used to investigate stability and catabolism of molecules trafficked to an endosome-lysosome pathway, and used MALDI mass spectroscopy to assess the emergence of a PEGylated diABZI adduct that would be predicted due to amide bond cleavage by lysosomal proteases (**Fig. S6**). We observed the presence of this peak as early as 1 h following incubation with Tritosomes, suggesting that a fraction of nAlb-diABZI is lysosomally degraded to release a PEGylated diABZI variant. We synthesized this compound (**Fig. S7-9**) and evaluated *in vitro* activity in THP1 IFN-I reporter cells, finding that it had a similar EC₅₀ value to the previously described diABZI molecule, which can enter cells through passive transport²³ (**Fig. S10**).”

Figure 2. (a) Dose-response curve for nanobody-Cy5 conjugate surface binding and intracellular uptake at 37 °C and 4 °C measured by flow cytometry in EGFR⁺ (THP-1) *in vitro*. (b) Uptake of nAlb-Cy5 (2 μM) in RAW 264.7, EMT6, and BMDM cells with the addition of control PBS (-EIPA) or macropinocytosis inhibitor (+EIPA). (c) Colocalization of Cy5 (red) with lysotracker green (green) and Hoechst (blue) in RAW 264.7 cells with (d) percent colocalization determination for nAlb-Cy5 and nGFP-Cy5 in RAW 264.7 and EMT6 cells. (scale bar: 100 μm)

Extended Data Figure 1: *In vitro* analysis of nanobody delivery. (a) Uptake of MSA-AF647 (1 μM) in BMDM cells as well as (b) nAlb-Cy5 (2 μM) and nGFP-Cy5 (2 μM) in EMT6 cells with the addition of control PBS (+serum) or in a serum starved condition (-serum). *P* values determined by two-way ANOVA with post-hoc Tukey's correction for multiple comparisons; *****P*<0.0001. (c) Uptake of MSA-AF647 (1 μM) in BMDC cells and (d) EMT6 cells at 37 °C and 4 °C. *P* values determined by Student's t-test. (e-g) Uptake of nAlb-Cy5 (2 μM) in (e) BMDC cells, (f) BMDM cells, and (g) EMT6 cells at 37 °C and 4 °C. *P* values determined by Student's t-test. (h) Integrated pixel intensity of Gal8-GFP puncta, per cell, for DBCO-PEG11-diABZI, nAlb-diABZI, and AP-diABZI at 0.25 μM (mean ± SEM, n=3-9). *P* values determined via one-way ANOVA, with Tukey's post-hoc. (i) Colocalization analysis of nAlb-Cy5 (2 μM) and nGFP-Cy5 (2 μM) for Raw 264.7 cells to visualize the cell nuclei (Hoechst, blue), lysosomes (Lysogreen, green), and Cy5 (Cy5, red). Merged images of the stains provided, in addition to a 2.5x zoom of the merged image (white box). (scale bar: 100 μm)

Figure S6: Tritosome degradation assay for **(a)** diABZI-amine, **(b)** DBCO-PEG₁₁-diABZI, and **(c-d)** nAlb-amine with highlighted molecular weight ranges at 780 Da (diABZI-amine construct, blue dashed line), 1688 Da (DBCO-PEG₁₁-diABZI construct, orange dashed line), and 1906 Da (Amine-PEG₃-Triazole-PEG₁₁-diABZI).

Figure S7: Synthesis of Amine-PEG₃-Triazole-PEG₁₁-diABZI.

Figure S10: *In vitro* dose-response curves for THP1-Dual reporter cells comparing diABZI to Amine-PEG₃-Triazole-PEG₁₁-diABZI.

3. In figure 1j, why agonist conjugated with nanobody (nAlb-diABZI) triggered higher activation level in *Ifnb* and *Cxcl1* than original diABZI?

The reviewer raises an interesting point. The kinetics of gene expression in response to STING activation are tightly and differentially regulated and therefore apparent differences in the magnitude of response as measured by gene expression *in vitro* may also reflect the time point selected for analysis. To account for this possibility, we sought to establish a better understanding of how nAlb-diABZI and DBCO-diABZI, which has similar *in vitro* activity to commercially available diABZI (Figure 1h-i), impact STING-mediated gene expression as a function of time. We therefore conducted a new experiment in which we treated EMT6 cells and measured gene expression levels of *Ifnb1*, *Tnfa*, and *Cxcl10* at 0, 0.5, 1, 2, 3, 4, 6, 8, and 24 h. This data is provided to the reviewer as Figure R2 below. Here we found that nAlb-diABZI and DBCO-diABZI had comparable activity, though differences between treatments and specific genes could be observed at different timepoints; for example, DBCO-diABZI results in slightly higher *Cxcl10* expression than nAlb-diABZI at 4h, whereas nAlb-diABZI results in increased *Tnfa* expression at 6 h. This likely reflects the complex and dynamic response to STING agonists that will be influenced by many factors, including cellular uptake pathways, internalization and drug release kinetics, and cell-specific regulation of gene expression in response to IRF3 and NF- κ B signaling. However, we found that nAlb is primarily endocytosed by tumor-associated myeloid cells (see Figure 2k,l) and that macrophages (bone marrow-derived macrophages; BMDMs) are much more sensitive to nAlb-diABZI than EMT6 cells (Figure R3, shown with *Ifnb* gene expression) and so we have replaced this data with analogous data comparing the activity of nAlb-diABZI and diABZI in BMDMs (Figure 1j). We also note that this is consistent with a new experiment performed in response to another reviewer in which we treated BMDMs with nanobody-diABZI conjugates, and evaluated gene expression levels of *Ifnb1*, *Tnfa*, and *Cxcl10* at 0, 0.5, 1, 2, 3, 4, 6, 8, and 24 h. As shown in revised Figure 4f, all nanobody-diABZI conjugates (nPD-L1-diABZI, nAlb-diABZI, AP-diABZI) were more active than the DBCO-PEG-diABZI parent compound, which, like commercially available diABZI, we have consistently found to be poorly active in BMDMs. Considering this complexity, we also note that the only conclusion that we draw from these studies is that nAlb-diABZI is active cell types of relevance, which is reflected in the text on Page 16:

Page 16: “We found that all nanobody-diABZI conjugates were potently active in both reporter cell lines without evidence of cytotoxicity (Fig. S23) and that nanobody-diABZI conjugates were more active than the parent DBCO-diABZI in BMDMs and triggered STING-associated gene expression with similar kinetics (Fig. 4f); both nAlb-diABZI and AP-diABZI were also active in murine bone marrow-derived dendritic cells (BMDC; Fig. S22).”

Figure R2: qPCR analysis of gene expression in EMT6 cells various time points, indicated as 0, 0.5, 1, 2, 3, 4, 6, 8, and 24 h, after *in vitro* treatment with 0.25 μ M of free DBCO-diABZI, nAlb-diABZI, or AP-diABZI (n=3). Replicates are noted as biological, and data shown as mean \pm SEM.

Figure R3: qPCR analysis of *Ifnb* gene expression in EMT6 and B16.F10 cells at 6 h. *P* value determined by Student's t-test, $n=3$ per group with $**P\leq 0.01$ and $***P\leq 0.001$. Replicates are noted as biological, and data shown as mean \pm SEM.

Figure 1: (j) qPCR analysis of gene expression in murine bone marrow derived macrophages (BMDM) treated *in vitro* with $0.25 \mu\text{M}$ of free diABZI or nAlb-diABZI conjugate ($n=3$). *P* values determined by one-way ANOVA with post-hoc Tukey's correction for multiple comparisons; $*P\leq 0.05$, $**P\leq 0.01$, $***P\leq 0.001$, and $****P<0.0001$ compared to PBS control. Replicates are noted as biological, and data shown as mean \pm SEM.

Figure 4: (f) qPCR analysis of genes associated with STING activation in bone marrow derived macrophages (BMDMs) in response to treatment at discrete time points (0, 0.5, 1, 2, 3, 4, 6, 8, 24 h) with indicated agonist at $0.25 \mu\text{M}$ ($n=3$). *P* values determined by one-way ANOVA with post-hoc Tukey's correction for multiple comparisons; $*P\leq 0.05$, $**P\leq 0.01$, $***P\leq 0.001$, and $****P<0.0001$ compared to PBS control.

4. In figure 4d-e, how to explain the interferon activities of diABZI conjugated with nPD-L1 (nPD-L1-diABZI and AP-diABZI) were higher than other groups at high agonist concentration.

While the potency (EC_{50}) of all nanobody-diABZI conjugates is largely similar (i.e., 10s to 100s nM range), the reviewer makes an interesting observation that the maximal IFN-I response (as assessed by bioluminescence in IRF3 reporter cell lines), tends to be a bit higher for constructs containing nPD-L1. In our experience using these reporter cells, we have often observed an inverse correlation between potency and maximal IFN-I response that may result from counter regulation of JAK/STAT signaling in response to more potent agonists. This may also reflect differences in the kinetics of the response, with more potent agonists triggering the production of luciferase early (e.g., within 6 h) that subsequently degrades or destabilizes prior to bioluminescence measurements at 24 h. It may also be the case that this reflects changes in cellular internalization kinetics or pathways for conjugates with PD-L1 binding domains as both A549 and THP1 cells are

reported to express PD-L1. Therefore, we feel that data generated using these reporter cell assays is best used to estimate the relative potency of innate immune agonists and that care should be taken interpreting the significance of the maximum activity as well as relatively small differences in EC₅₀ values. However, in response to another reviewer comment, we have compared the activity of nAlb-diABZI and AP-diABZI by measuring the magnitude and kinetics of STING-drive gene expression in primary macrophages (BMDMs) and dendritic cells (BMDCs) and have included this data in **Figure 4f** and **Figure S22**. These studies suggest potentially dynamic and complex signaling in response to nAlb1-diABZI and AP-diABZI that may also be cell-type dependent and different in an *in vivo* setting; hence, we believe that dissecting these differences is beyond the scope of the current work and we have limited our conclusions arising from this data.

Figure 4: (f) qPCR analysis of genes associated with STING activation in bone marrow derived macrophages (BMDMs) in response to treatment at discrete time points (0, 0.5, 1, 2, 3, 4, 6, 8, 24 h) with indicated agonist at 0.25 μ M (n=3). *P* values determined by one-way ANOVA with post-hoc Tukey’s correction for multiple comparisons; **P*≤0.05, ***P*≤0.01, ****P*≤0.001, and *****P*<0.0001 compared to PBS control.

Figure S22: *In vitro* qPCR analysis of genes associated with STING activation in bone marrow derived dendritic cells (BMDCs) as analyzed (**a-c**) over 24 h or (**d-f**) at 6 h post-treatment with nanobody-diABZI conjugates at 0.25 μ M (n=5-6). *P* values determined by one-way ANOVA with post-hoc Tukey’s correction for multiple comparisons; ***P*≤0.01 and *****P*<0.0001 compared to PBS control.

5. In figure 5 and figure 7, mouse weights results should be supplemented.

We have included weight loss curves for all experiments and models performed in this work as a new **Figure S15** and have referenced this figure when reporting data related to these studies.

Figure S15: Mouse weight loss curves during treatment course with diABZI and diABZI conjugated proteins from indicated studies as represented by subfigure titles.

6. In introduction section, it is better to add some comparisons with current albumin-hitchhiking drugs, such as the difference of the delivery of immune agonists and chemotherapeutic agents, and the advantages of the nanobody in situ conjugating albumin strategy.

We thank the reviewer for this suggestion and in response we have expanded upon our discussion of other albumin-hitchhiking drugs and our rationale for using nanobodies on **Pages 4, 5 and 30** as followed; please note that journal length and reference number requirements limit further analysis and discussion of this important previous work:

Page 4: “Albumin is a promising drug carrier based on its long circulation half-life and proclivity to accumulate at tumor sites via both passive and active transport mechanisms.²⁷⁻²⁹ Albumin and albumin-binding chaperones have been widely employed to improve the delivery of chemotherapeutics, exemplified by albumin-bound paclitaxel (Abraxane®)³⁰, as well as protein³¹, peptide³², and nucleic acid therapeutics³⁰. Inspired by this previous work that motivates the unexplored potential of albumin as a carrier for STING agonists, we engineered a high-affinity anti-albumin nanobody (i.e., single-domain antibody) for site-selective enzymatic bioconjugation of STING agonists via biorthogonal chemistry.”

Pages 4-5: “While several promising albumin-binding molecules have been described, including small molecules, fatty acids, and peptides,^{27, 29} we elected to build our platform from a nanobody with high affinity for albumin because nanobodies are modular and programmable via genetic engineering, are molecularly well-defined, are amenable to scalable industrial manufacturing, and are components of approved and clinically-advanced therapeutics, including ozoralizumab, which contains an anti-albumin nanobody domain.³¹ Additionally, we sought to avoid the potential risk of accelerated albumin clearance that can occur due to direct covalent drug conjugation strategies^{27, 29} and to minimize the liver accumulation associated with the use of lipid-based albumin binders³⁰, a challenge also faced by many promising nanoparticle-based STING agonists.^{19, 21, 33, 34}”

Page 30: “To date, most research on albumin-based drug carriers for cancer (e.g., Abraxane, aldoxorubicin) has focused on delivery of chemotherapy drugs that target cancer cells. By contrast, immunostimulatory agents such as STING agonists can stimulate complex antitumor immunological programs that may be more dependent on immunological variables (e.g., neoantigen load, immune status of the TME) than on the efficiency of drug accumulation in tumor tissue or delivery to cancer cells.”

7. The paragraph from line 187 to 196 lacks the reference marking of figure1 h-j.

Thank you for bringing this to our attention; this was an oversight and we have corrected the text to refer to **Figure 1h-j**.

8. There is a word misspelling in line 632, which “micrometases” should be corrected to “micrometastases”.

Thank you for catching this error and we have corrected this.

Reviewer #4 (Report for the authors (Required)):

This study by Wilson et al describes the development of a nanobody-conjugated STING agonist with albumin-hitchhiking property for activation of tumor immunity. There are multiple strengths of the paper: much longer blood circulation of the nAlb-diABZI complex, increased tumor accumulation, comprehensive immune profiling after therapy, antitumor efficacy in multiple tumor models, and formation of bifunctional AP-diABZI that also target PD-L1. However, there are also many concerns over the current design and data, which can improve the enthusiasm toward this study after careful addressing.

1. Although longer blood circulation is beneficial for tumor accumulation, it may also elevate the toxicity. In fact, data show dramatically higher systemic cytokines (Fig. 7E) as well as some liver toxicity (Extended Data Fig. 1). Given the first antibody-diABZI conjugate XMT-2056 incurred patient death and halted clinical trials, the authors may need to carefully address the toxicity and safety concerns for nAlb-diABZI complex. A careful dose-response study is warranted to show the therapeutic window for the safe usage of the otherwise novel design.

The reviewer raises an important point of high translational significance. First, we agree that the systemic cytokine response is likely to be the primary source of dose-limiting toxicities for nAlb-diABZI; this is a challenge shared with virtually all immunostimulatory agents (e.g., PRR agonists, cytokines, etc.), including several other promising STING agonist platforms (18-21) and nanoparticle-based agonists that have entered clinical trials (22, 23) with similar cytokine profiles in preclinical models. This motivated our exploration into the effect of dose on antitumor efficacy and weight loss (**Figure S14**) where our goal was to establish a dose that resulted in less than <5-7% weight loss while still demonstrating efficacy in a poorly immunogenic B16.F10 tumor model. We found that a dose of 5 µg resulted in >10% weight loss, though only after the initial injection, while doses of 0.5 and 0.05 resulted in minimal weight loss (<5%) while still inhibiting tumor growth and extending survival time. These studies guided our selection of a 1.25 µg dose for all studies, where we consistently observe ~5% weight loss or less that is typically most pronounced after the first dose. Nonetheless, in response to reviewer comments, we also evaluated the efficacy of AP-diABZI at both our standard 1.25 µg dose and a 5x lower dose of 0.25 µg in the B16.F10 model. While, as expected, decreasing the dose reduced efficacy, AP-diABZI still significantly inhibited tumor growth without any significant changes in mouse body weight, consistent with our original dose-finding studies (**Figure S32**). Taken together, these studies demonstrate that the therapeutic benefit of AP-diABZI can be achieved over a range of ~0.05-1.25 µg diABZI with mice experiencing a <5% loss in body weight that recovers after injection. As another reviewer noted, we also performed a comprehensive preclinical toxicity analysis through evaluation of blood chemistry and organ pathology (**Extended Data Fig. 2**). While we did not observe any evidence of organ toxicity, including liver toxicity, we nonetheless repeated this study and directly involved experienced veterinary pathologists on collection and processing of blood for analysis. Consistent with our previous results, we observed insignificant or minor differences in all blood biochemistry markers (e.g., AST, ALT). Additionally, as described in our original manuscript, all organs were carefully examined by a blinded, board-certified veterinary pathologist who did not find any clinically significant differences between untreated control mice and mice treated three times with the 1.25 µg dose. Therefore, nAlb-diABZI and AP-diABZI are well-tolerated at our maximal dose of 1.25 µg, which we have also demonstrated can be reduced while still achieving antitumor efficacy. It should also be noted that the FDA has lifted the hold on XMT-2056 and the clinical trial has resumed with a lowered dose-escalation starting dose. This reflects the challenges in translating immunotherapy agents from preclinical models to patients with advanced disease and variable immune status. Further, at least in the peer-reviewed and publicly available data, preclinical assessment of XMT-2056 was performed in xenograft models using human tumors (SK-OV3) in immunodeficient mice (24), which may be an inadequate

model for evaluating STING agonists that may rely on T cells for optimal efficacy, as we have shown in our studies. While clearly additional studies would be required prior to translation of nAlb-diABZI or AP-diABZI, a potential advantage of our platform is that both nanobodies employed here are cross-reactive between mouse and human targets (25, 26). We discuss this data and analysis on **Pages 11-12, 19, and 24**:

Pages 11-12: “Using a treatment regimen that we and others have employed for evaluation of STING agonists,^{33, 43} we intravenously administered nAlb-diABZI to mice bearing ~75 mm³ B16.F10 tumors at doses ranging from 5-0.05 µg diABZI content, finding that all doses significantly ($P < 0.0001$) inhibited tumor growth and extended survival time. Notably, the 5 µg dose significantly ($P < 0.0001$) enhanced efficacy relative to a 3x higher dose of diABZI, demonstrating the enhancement in potency enabled through albumin-hitchhiking. While the 5 µg dose resulted in ~10-12% weight loss, this was transient and occurred only after the first injection (**Fig. S14a**). Nonetheless, towards maximizing the safety profile of the treatment, we selected a dose of 1.25 µg, confirmed antitumor efficacy of both a single and three-dose regimen in the B16.F10 model (**Fig. S14d-g; Fig. S15**), and performed a preclinical analysis of nAlb-diABZI toxicity (**Extended Data Fig. 2**). Healthy mice were administered vehicle (PBS) or nAlb-diABZI (1.25 µg diABZI) intravenously three times spaced three days apart, weight loss was monitored daily, and blood was collected 4 and 24 hr after the first injection for analysis of serum cytokines (**Extended Data Fig. 2**). In response to nAlb-diABZI, mice experienced only a mild (~5%) and transient weight loss similar to that described for nanoparticle-based delivery of STING agonists^{18, 19, 21} with elevated plasma levels of STING-driven cytokines with antitumor functions (e.g., type I IFN, IL-12 4 h following injection, which returned to near baseline by 24 h. Mice were euthanized a week following the last injection, blood was collected for biochemistry analysis (**Extended Figure 2d**), and major organs were isolated for histological evaluation (**Extended Figure 2e**) by a board certified veterinary pathologist, who observed no clinically significant changes between the untreated control mice and nAlb-diABZI treated mice, consistent with insignificant or minor changes in blood biochemistry and cellular composition. Based on this favorable safety profile at a therapeutically effective dose in a challenging B16.F10 tumor model, we selected a dose of 1.25 µg for all subsequent studies.”

Page 19: “Importantly, no additional toxicity was observed for AP-diABZI relative to nAlb-diABZI (**Extended Data Fig. 2**).”

Page 24: “We also found that AP-diABZI was more effective than free diABZI administered at 24 times (30 µg) the dose in the B16.F10 model (**Fig. S32**).”

Figure S14: Free diABZI (15 µg) and nAlb-diABZI (5, 0.5, 0.05 µg) were injected I.V. (a) Mouse weights, (b) Tumor growth curves, and (c) Kaplan-Meier survival plots for mice with B16.F10 tumors treated as indicated. (d) Schematic of B16.F10 tumor inoculation and treatment schedule. Free diABZI (1.25 µg) and nAlb-diABZI (1.25 µg) were injected I.V. (e) Tumor growth curves, (f) spider plots of individual tumor growth curves, and (g) Kaplan-Meier survival plots for mice with B16.F10 tumors treated as indicated. Data represented with error in SEM with P value determined by two-way ANOVA with post-hoc Tukey’s correction for multiple comparisons; (b,e) **** $P < 0.0001$ on day 20 for all groups compared to PBS. (c,g) Kaplan-Meier survival curves of mice treated with indicated formulation using 1500 mm³ tumor volume as endpoint criteria with P value was determined by log-rank test; **** $P < 0.0001$ compared to PBS control.

Figure S32: (a) Schematic of B16.F10 tumor inoculation and treatment schedule for PBS, diABZI (30 µg) and two doses of AP-diABZI (1.25 µg and 0.25 µg). **(b)** Tumor growth curves, **(c)** Kaplan-Meier survival plots (n=5-6). All doses were injected I.V. at the labeled amount of diABZI per injection. **(c)** Kaplan-Meier survival curves of mice treated with indicated formulation using 1500 mm³ tumor volume as endpoint criteria with P value was determined by log-rank test.

Extended Data Figure 2: (a) Scheme for treating healthy C57BL/6 female mice with PBS, 1.25 µg nAlb-diABZI, or 1.25 µg AP-diABZI. **(b)** Body weight change of mice during treatments. **(c)** Quantification of plasma cytokines 4 hours after the first treatment and 24 hours after the final treatment (n=3-5, $P \leq 0.05$, $^{*}P \leq 0.001$, $^{***}P < 0.0001$ by one-way ANOVA compared to PBS). **(d)** Blood biochemistry and **(e)** immunohistochemistry analysis of healthy C57BL/6 mice. After 3 treatments, mice were euthanized and blood samples were collected to determine changes in RBCs, white blood cells (WBCs), neutrophils, platelets, and lymphocytes. Serum samples were also used to analyze liver and kidney function, by measuring changes in ALT/AST, blood urea nitrogen (BUN), and creatinine. Tissue was sectioned, H&E stained, and imaged. Images use a scale bar of 200 µm. No significant changes were observed. ($^{*}P \leq 0.05$, $^{**}P \leq 0.01$, and $^{***}P \leq 0.001$ compared to PBS by one-way ANOVA).

Figure S15: Mouse weight loss curves during treatment course with diABZI and diABZI conjugated proteins from indicated studies as represented by subfigure titles.

2. Is free diABZI released from antibody-diABZI for STING activation in tumors? The click linker does not seem to be degradable. Active agent if not diABZI that binds and activates STING should be characterized.

We thank the reviewer for this important question. In our design, we used a stable covalent bond between the nanobody and the diABZI compound and integrated a PEG spacer to improve solubility. While our approach is highly amenable to integration of cleavable linkers (e.g., cathepsin B cleavable), we elected to start with a stable linker for simplicity and to create a platform from to further build and optimize; additionally, the use of stable linkers minimizes premature drug release, which is an established limitation of some linkers used for development of antibody-drug conjugates (27). We postulated that diABZI was released following lysosomal degradation and we conducted several new experiments to evaluate this. First, we demonstrated using fluorescent microscopy that nAlb-Cy5 colocalizes with lysosomes (Mander's overlap coefficient of >0.5) as measured using a LysoTracker dye (**Figure 2c,d, Extended Data Fig. 1i**). Based on this, we next incubated nAlb-diABZI with rat liver tritosomes, which are isolated and stabilized lysosomes and employed to evaluate the stability of protein-drug conjugates, and we used MALDI to evaluate the emergence of potential cleavage products (**Figure S6**). As early as 1 h following incubation with tritosomes, we observed the formation of an NH₂-PEG₃-triazole-DBCO-PEG₁₁-diABZI product (**Figure S7**) corresponding to cleavage of the amide bond ligating the diABZI molecule to the nanobody. We then synthesized and purified this product (see **Figure S7**) and tested *in vitro* activity compared to the previously reported diABZI compound in THP1 IFN-I reporter cells and found that the released compound was a highly potent STING agonist with comparable activity to diABZI (**Figure S10**). This is described in revised text on **Page 8** as followed:

Page 8: "Given that macropinosomes often traffic to lysosomes, we next assessed colocalization of nAlb-Cy5 with lysotracker and found that a substantial and similar fraction (>50%) of nAlb-Cy5 and nGFP-Cy5 was colocalized with lysosomes or late endosomes in EMT6 and RAW264.7 cells (**Fig. 2c,d**)." As expected, nAlb-diABZI did not mediate endosomal disruption as assessed using a previously described galectin 8 (Gal8) endosomal recruitment assay (**Extended Data Fig. 1**)⁴¹.

To gain insight into how amide-linked diABZI is released from the nanobody upon cellular internalization, we incubated nAlb-diABZI with lysosomes isolated from rat liver (Tritosomes), which are used to investigate stability and catabolism of molecules trafficked to an endosome-lysosome pathway, and used MALDI mass spectroscopy to assess the emergence of a PEGylated diABZI adduct that would be predicted due to amide bond cleavage by lysosomal proteases (**Fig. S6**). We observed the presence of this peak as early as 1 h following incubation with Tritosomes, suggesting that a fraction of nAlb-diABZI is lysosomally degraded to release a PEGylated diABZI variant. We synthesized this compound (**Fig. S7-9**) and evaluated *in vitro* activity in THP1

IFN- β reporter cells, finding that it had a similar EC50 value to the previously described diABZI molecule, which can enter cells through passive transport²³ (Fig. S10).”

Figure 2. (c) Colocalization of Cy5 (red) with lysotracker green (green) and Hoechst (blue) in RAW 264.7 cells with (d) percent colocalization determination for nAlb-Cy5 and nGFP-Cy5 in RAW 264.7 and EMT6 cells. (scale bar: 100 μ m)

Extended Data Figure 1: (i) Colocalization analysis of nAlb-Cy5 (2 μ M) and nGFP-Cy5 (2 μ M) for Raw 264.7 cells to visualize the cell nuclei (Hoechst, blue), lysosomes (Lysogreen, green), and Cy5 (Cy5, red). Merged images of the stains provided, in addition to a 2.5x zoom of the merged image (white box). (scale bar: 100 μ m)

Figure S6: Tritosome degradation assay for (a) diABZI-amine, (b) DBCO-PEG₁₁-diABZI, and (c-d) nAlb-amine with highlighted molecular weight ranges at 780 Da (diABZI-amine construct, blue dashed line), 1688 Da (DBCO-PEG₁₁-diABZI construct, orange dashed line), and 1906 Da (Amine-PEG₃-Triazole-PEG₁₁-diABZI).

Figure S7: Synthesis of Amine-PEG₃-Triazole-PEG₁₁-diABZI.

Figure S10: *In vitro* dose-response curves for THP1-Dual reporter cells comparing diABZI to Amine-PEG₃-Triazole-PEG₁₁-diABZI.

3. Would nAlb-diABZI therapy work in tumors with less albumin catabolism capacity?

The reviewer raises an interesting and important question that has also been a topic of significant debate regarding albumin-based delivery of chemotherapeutics (5). Unlike chemotherapy, where the primary target is cancer cells, nAlb-diABZI can activate STING in multiple cell populations. For example, flow cytometric analysis suggests that nAlb-Cy5 is predominantly internalized by tumor-associated myeloid cells (**Figure 2k,i**) and we found macrophages to be more responsive to nAlb-diABZI than EMT6 tumor cells (**Figure R3**), suggesting that the myeloid composition of tumors may be an important determinant of the response. Similarly, responses to immunotherapy are dictated by many factors, including the frequency and immunogenicity of tumor antigens and the immunocellular composition of the TME, and, therefore, may not be strictly correlated with drug accumulation within tumors. Notably, in our analysis of nAlb-Cy5 biodistribution, we found ~11% ID/gram tumor in the EMT6 model and a comparable ~8.4% ID/gram tumor in B16.10 model (**Figure 2h,i**), yet a substantial difference in response to both nAlb-diABZI alone and in combination with anti-PD-L1 (see **Figures 5, 7, and 8**) that can most likely be attributed to the lower immunogenicity of B16.F10 tumors. Therefore, nAlb-diABZI accumulation and catabolism likely represent one, but not the only, factor that determines efficacy. We have expanded upon our discussion of this point on **Page 30**:

Page 30: To date, most research on albumin-based drug carriers for cancer (e.g., Abraxane, aldoxorubicin) has focused on delivery of chemotherapy drugs that target cancer cells. By contrast, immunostimulatory agents such as STING agonists can stimulate complex antitumor immunological programs that may be more dependent on immunological variables (e.g., neoantigen load, immune status of the TME) than on the efficiency of drug accumulation in tumor tissue or delivery to cancer cells. For example, in our analysis of nAlb-Cy5 biodistribution, we found ~11% ID/gram tumor in the EMT6 model and a comparable ~8.4% ID/gram tumor in B16.10 model (**Fig. 2**), yet a substantial difference in response to both nAlb-diABZI alone and in combination with anti-PD-L1 that may be attributed to the relatively low immunogenicity to B16.F10 tumors.

Figure R3: qPCR analysis of *Ifnb* gene expression in EMT6 and B16.F10 cells at 6 h. *P* value determined by Student's t-test, *n*=3 per group with ***P*≤0.01 and ****P*≤0.001. Replicates are noted as biological, and data shown as mean ± SEM.

Figure 2: (h-i) Quantification of percent injected dose per gram of tissue (% ID/g) 24 h following intravenous administration of vehicle (PBS) and nAlb-Cy5 at 2 mg/kg to (h) female Balb/c mice with orthotopic EMT6 breast tumors (*n*=5) and (i) female C57BL/6 mice with subcutaneous B16.F10 tumors (*n*=5). *P* values determined by two-way ANOVA with post-hoc Tukey's correction for multiple comparisons; **P*≤0.05, ***P*≤0.01, ****P*≤0.001, and *****P*<0.0001.

4. The authors show significant increase in NK, CD4 T and CD8 T cell populations after STING therapy. Cell blocking experiments should be performed to evaluate which immune cell population is responsible for antitumor efficacy.

We thank the reviewer for this suggestion and in response we have conducted new experiments in which we have antibody depleted NK, CD4, and CD8 T cells and have evaluated response to AP-diABZI in both the EMT6 (Figure 6f-h) and B16.F10 (Extended Data Fig. 9) tumor model. In both models, we found that the response was almost entirely dependent on CD8⁺ T cells and strongly dependent on NK cells but was independent of CD4⁺ T cells. These data are presented in Figure 6f-h and Extended Data Fig. 9 and the text has been revised on Pages 22-25 as followed:

Pages 22-23: “To test this, we antibody depleted NK cells, CD8⁺ T cells, and CD4⁺ T cells and evaluated antitumor responses elicited by AP-diABZI treatment. Again, we observed a 100% CR rate for AP-diABZI, but treatment efficacy was almost completely inhibited with CD8⁺ T cell NK cell depletion, with CD8⁺ T cell depletion having a slightly stronger effect (Fig. 6f-h); no effect of CD4⁺ T cell depletion was observed. Therefore, both NK cells and CD8⁺ T cells are essential to the potent efficacy of AP-diABZI in an EMT-6 breast tumor model.”

Page 24-25: “To determine the primary cellular effectors to AP-diABZI in the B16.F10 model, we antibody depleted CD4⁺ and CD8⁺ T cells and NK cells again finding that the antitumor response was mediated predominantly by CD8⁺ T and NK cells (Extended Data Fig. 9).”

Figure 6: (f) Schematic of EMT6 tumor inoculation and treatment schedule with depletion antibodies (n=7-13). Anti-Asialo GM1 (NK) IgG, anti-CD8 IgG, and anti-CD4 IgG were injected I.P. at 100-200 μ g and AP-diABZI was injected I.V. at 1.25 μ g of diABZI per injection. **(g)** Tumor growth curves, and **(h)** Kaplan-Meier survival plots for mice with EMT6 tumors treated as indicated. CR = complete responder; SEM with *P* value determined by two-way ANOVA with post-hoc Tukey's correction for multiple comparisons; *****P*<0.0001 on day 22 for all groups compared to PBS. Endpoint criteria of 1500 mm³ tumor volume with *P* value was determined by log-rank test; *****P*<0.0001 compared to PBS control. Replicates are noted as biological, and data shown as mean \pm SEM.

Extended Data Figure 9: (a) Schematic of B16.F10 tumor inoculation and treatment schedule with depletion antibodies (n=8-9). Anti-NK1.1 (NK) IgG, anti-CD8 IgG, and anti-CD4 IgG were injected I.P. at 200 μ g and AP-diABZI was injected I.V. at 1.25 μ g of diABZI per injection. **(b)** Tumor growth curves and **(c)** Kaplan-Meier survival plots for mice with B16.F10 tumors treated as indicated. SEM with *P* value determined by two-way ANOVA with post-hoc Tukey's correction for multiple comparisons; *****P*<0.0001 on day 22 for all groups compared to PBS. Endpoint criteria of 1500 mm³ tumor volume with *P* value was determined by log-rank test; *****P*<0.0001 compared to PBS control. Replicates are noted as biological, and data shown as mean \pm SEM.

5. m- and g-MDSC population also increased after nAlb-diABZI therapy (Fig. 3). It has been shown that both MDSC populations render different cell proliferation and metastatic properties in tumors (Ouzounova et al, Nature Comm. 2017, 8, 14979). Can authors comment on whether this is directly due to nAlb-diABZI, or potential systemic cytokine induced effect? Would MDSC drive resistance to nAlb-diABZI therapy?

The reviewer raises an important point as MDSCs have been implicated as a major resistance mechanism to immunotherapy. Indeed, previous reports have shown that STING activation can increase MDSC infiltration into tumors and may also contribute to increased metastasis in some models (11-14). To determine if MDSCs contributed to resistance to nAlb-diABZI or AP-diABZI we performed two additional experiments in the EMT6 breast cancer model. First, we evaluated nAlb-diABZI and AP-diABZI in combination with orally administered SX-682, a CXCR1/2 inhibitor that blocks MDSC recruitment to tumors that is being evaluated in clinical trials. Second, we administered anti-Gr1 antibodies to deplete MDSCs. Surprisingly, in both studies, we observed a modest *reduction* in nAlb-diABZI and AP-diABZI efficacy (Fig. S18-19. Fig. S27), indicating that MDSCs do not strongly contribute to resistance and may even play a small role in treatment efficacy. While this was unexpected, similar negative effects of MDSC inhibition on antitumor responses have been observed by others (15) and may reflect the complex roles that MDSCs can have in response to immunotherapy. We describe these findings on **Pages 15 and 22:**

Page 15: "In particular, MDSCs have been reported to reduce the efficacy of some STING-activating therapies⁴⁷⁻⁴⁹ and we therefore evaluated nAlb-diABZI in combination with orally administered SX-682, which inhibits CXCR1/2 chemokine receptors involved in MDSC recruitment⁵⁰, but, surprisingly, found that SX-682 tended to

reduce nAlb-diABZI efficacy (Fig. S18). We also used anti-Gr1 antibodies to deplete MDSCs (primarily gMDSCs)⁵¹ and again found a modest reduction in nAlb-diABZI efficacy (Fig. S19).”

Page 22: “However, as observed with nAlb-diABZI treatment, inhibition of MDSCs using SX-682 or anti-GR1 antibody depletion reduced AP-diABZI treatment efficacy (Fig. S18, S19, S27).”

Figure S18: (a) Schematic of EMT6 tumor inoculation and treatment schedule for PBS, nAlb-diABZI (1.25 µg) and AP-diABZI (1.25 µg) during MDSC inhibitor treatment using SX-682 compared to regular chow. (b) Tumor growth curves, (c) Kaplan-Meier survival plots (n=8-9). All doses were injected I.V. at the labeled amount of diABZI per injection. (c) Kaplan-Meier survival curves of mice treated with indicated formulation using 1500 mm³ tumor volume as endpoint criteria with P value was determined by log-rank test.

Figure S19: (a) Schematic of EMT6 tumor inoculation and treatment schedule for PBS, nAlb-diABZI (1.25 µg), and for AP-diABZI (1.25 µg) with anti-GR-1 MDSC depletion. (b) Tumor growth curves, (c) Kaplan-Meier survival plots (n=8). All doses were injected I.V. at the labeled amount of diABZI per injection. (c) Kaplan-Meier survival curves of mice treated with indicated formulation using 1500 mm³ tumor volume as endpoint criteria with P value was determined by log-rank test.

We also evaluated AP-diABZI in a MMTV-PyMT model of spontaneous breast cancer and found that systemic administration of AP-diABZI significantly reduced tumor burden without increasing lung metastasis. This data is shown in **Extended Data Figure 4** and described on **Page 20** as stated below. Additionally, in our original manuscript we demonstrated that AP-diABZI potently inhibited lung colonization of B16.F10 melanoma cells (**Figure 8**). Therefore, while the possibility that nAlb-diABZI or AP-diABZI could promote metastasis in other tumor models should not be discounted, our studies indicate that this is unlikely to be a major barrier to treatment efficacy at least in these preclinical models of breast cancer and melanoma.

Page 20: “We also evaluated AP-diABZI in a MMTV-PyMT model of spontaneous breast cancer, finding that systemic administration of AP-diABZI significantly reduced tumor burden without evidence of increased lung metastasis (**Extended Data Fig. 4**), which has been implicated as a potentially deleterious consequence of STING signaling in some preclinical models.”

Extended Data Figure 4: Female *FVB/N-Tg (MMTV-PyVT)^{634Mul}* mice with tumors were treated with PBS or AP-diABZI (1.25 μ g of diABZI per dose) once a week for 3 weeks starting at approximately 8-10 weeks of age. **(a)** Growth rates of tumors during treatment until the study was terminated on day 22. At necropsy, tumors were removed and **(b)** weighted in addition to **(c)** histological analysis of lungs to determine tumor burden.

6. Can authors investigate whether nAlb-diABZI therapy result in formation of albumin-specific mAbs and impact clearance of nAlb-diABZI after 2nd or 3rd injections?

This is an excellent question of high translational significance. We therefore conducted two additional studies to address this concern. First, we evaluated anti-VHH antibody titer in serum on day 14 following a therapeutic dosing regimen of nAlb-diABZI on day 0, 3, and 6 in healthy mice, and did not detect anti-VHH antibodies via ELISA; this data is shown in **Figure S20**. We also compared the plasma half-life of nAlb-Cy5 in untreated mice or 14 days following treatment with the therapeutic nAlb-diABZI regimen (i.e., dosed 3 times, 3 days apart), and observed a similar half-life between untreated mice (~59 h) and treated mice (~64 h). Together, these data suggest that anti-nAlb antibodies are not generated at a level that would affect pharmacokinetics or nAlb-diABZI efficacy. This data is presented in new **Figure S21** and the text has been revised on **Page 15** as followed:

Page 15: “In addition to immunological resistance mechanisms, the efficacy of nAlb-diABZI may also be inhibited through generation of anti-nAlb or anti-diABZI antibodies that may lead to accelerated blood clearance. While albumin has been described to generate immune tolerance to antigenic cargo and nanobodies typically have low immunogenicity, we addressed this possibility by intravenously administering healthy wild-type C57BL/6 mice with nAlb-diABZI on day 0, 3, and 6 and evaluated anti-VHH antibody titer in serum on day 14 and also compared the plasma half-life of nAlb-Cy5 to untreated mice. We did not detect an anti-VHH antibody response in serum (**Fig. S20**) and observed a similar nAlb-Cy5 half-life between untreated mice (~59 h) and treated mice (~64 h) (**Fig. S21**), suggesting that antibody-mediated nanobody clearance was unlikely to reduce nAlb-diABZI efficacy in our studies; however, this possibility cannot be discounted in humans where dose and treatment regimen, amongst other variables, will be different and therefore will need to be further investigated.”

Figure S20: anti-VHH antibody titer for nAlb response in **(a)** heat map and **(b)** averaged dilution curves after treatment with PBS, nAlb-diABZI (1 dose), or early pre-treatment (3 doses) with nAlb-diABZI.

Figure S21: (a) Pharmacokinetics of nAlb-Cy5 and pre-treated nAlb-Cy5 ($n=5$), with elimination phase half-life indicated in legend. **(b)** Quantification of radiant efficiencies 24 h following intravenous administration of PBS, nAlb-Cy5, or nAlb-Cy5 after pre-treated nAlb-diABZI (3 doses) at 2 mg/kg.

7. On the style side, the authors packed too many subfigures in each main figure. It will be helpful to prioritize the data selection. It diminishes general readability if the font size is too small and organization too complex.

We thank the reviewer for this suggestion, and we agree that some of our figures could be simplified. This is a challenge faced for many manuscripts that contain large amounts of data and the need to balance presentation of data that is most important to the primary conclusions of the study without relegating too much data to the Supporting Information which imposes a barrier to comprehension as readers must jump back and forth between documents. Nonetheless, we have attempted to simplify our figures, noting also that we generated considerable data during the revision process, some of which strengthens the work and our primary conclusions and was therefore integrated into the primary figures. In particular, we have made the following changes to the figures that we feel has improved readability: a reduction in **Figure 5** (moved serum cytokine data to supporting information), a reduction in **Figure 6** (moved flow cytometry analysis to supporting information), a reduction in **Figure 7** (moved serum cytokine data to supporting information), and a reduction in **Figure 8** (moved histology to supporting information).

Reviewer #5 (Report for the authors (Required)):

In this manuscript, the authors constructed a bifunctional nanobody for delivering a STING agonist to stimulate innate immune programs for tumor suppression. The nanobody was composed of an anti-albumin nanobody (nAlb) and an anti-PD-L1 nanobody (nPD-L1), hence the conjugate demonstrated extended in vivo circulation by “albumin-hitchhiking”, resulting in enhanced cellular uptake through albumin-mediated endocytosis, confirmed by the antibody blockade of SPARC and GP60. The presence of anti-PD-L1 nanobody further enhanced the immune-response through immune checkpoint inhibition. The treatment increased the infiltration of activated natural killer cells and T cells, eliminated primary tumors, and resulted in immunological memory even 80 days after initial tumor inoculation.

While the immunological assays were comprehensively performed in this work, the design of the delivery platform is rather straightforward and the novelty is marginal. The programmable linking bi- even multi-functional nanobodies has been well-investigated, using anti-albumin nanobody to prolong in vivo circulation is a general strategy for nanobody applications (Caplacizumab in clinical use, *N Engl J Med* 2019; 380, 335-346, and many others, for example *FASEB J.* 2018, 32(6) 3411-3422); site-specific loading drug in multi-functional nanobodies has been reported (*Chem Commun*, 2019, 55(35), 5175-5178). In addition, the following issues should be addressed.

We thank the Reviewer for their thorough assessment of our manuscript. We understand the reviewer’s position regarding the novelty of our approach given previous work employing albumin and albumin-binding chaperones to enhance drug accumulation at tumor sites. However, this concept has not yet been explored for delivery of STING agonists, a promising class of therapeutics with mechanisms of action that are distinct from chemotherapeutics. Therefore, harnessing albumin-hitchhiking to improve the delivery of STING agonists

represents a conceptually different approach from previous work in the field that has mostly leveraged synthetic nanoparticles. We also believe that our approach represents a significant technological innovation as we devised a robust workflow for synthesis of nanobody-STING agonist conjugates that combines site-selective enzymatic bioconjugation with biorthogonal ligation to a novel “clickable” STING agonist at a precisely defined ratio. This is an enabling synthetic scheme for covalent conjugation of a leading STING agonist to drug carriers that has broad potential future applications. Additionally, as exemplified by integration of a PD-L1 binding domain, our strategy provides “plug-and-play” programmability for optimizing agonist delivery using a single, well-defined molecule. However, we also acknowledge that the manuscript would be improved by better placing our work in the context of previous literature describing the use of albumin-binding domains to improve the delivery of therapeutics. While restraints in manuscript length and reference length precluded an in-depth analysis of this previous important work, we have made substantial changes to the text to discuss previous research using albumin as a drug carrier and how this motivated our strategy for delivery of STING agonists; these changes are summarized on **Pages 4, 29, and 30** and below:

Page 4: *“Albumin is a promising drug carrier based on its long circulation half-life and proclivity to accumulate at tumor sites via both passive and active transport mechanisms.²⁷⁻²⁹ Albumin and albumin-binding chaperones have been widely employed to improve the delivery of chemotherapeutics, exemplified by albumin-bound paclitaxel (Abraxane®)³⁰, as well as protein³¹, peptide³², and nucleic acid therapeutics³⁰. Inspired by this previous work that motivates the unexplored potential of albumin as a carrier for STING agonists, we engineered a high-affinity anti-albumin nanobody (i.e., single-domain antibody) for site-selective enzymatic bioconjugation of STING agonists via biorthogonal chemistry.”*

Page 29: *An appealing feature of anti-albumin nanobodies, and other protein-based albumin hitchhiking agents (e.g., affibodies, humabodies) over other albumin binders (e.g., lipids, Evans blue) is the high degree of molecular programmability that can be achieved through protein engineering.*

Page 30: *“To date, most research on albumin-based drug carriers for cancer (e.g., Abraxane, adoxorubicin) has focused on delivery of chemotherapy drugs that target cancer cells. By contrast, immunostimulatory agents such as STING agonists can stimulate complex antitumor immunological programs that may be more dependent on immunological variables (e.g., neoantigen load, immune status of the TME) than on the efficiency of drug accumulation in tumor tissue or delivery to cancer cells.”*

1. One of my major concerns is the cellular uptake of the drug through nanobody binding to albumin. Different from cell-based assay, in which the total amount of albumin is limited and fixed, the ratio of nAlb-bound albumin in vivo was very low. I did a rough calculation. A 25 g mouse has 1.8 mL blood and contains 42 mg albumin (23.6 mg/mL, <https://bionumbers.hms.harvard.edu/bionumber.aspx?s=n&v=2&id=102312>), which is about 0.6 micromoles. When mice were treated with 2 mg/Kg nanobody, which was about 3.5 nanomole in a mouse. The ratio of nAlb-bound albumin was < 1% among total albumin in vivo.

Assuming the nanobody binding does not increase the endocytosis efficiency of albumin (otherwise the authors should provide the evidence), very little amount of nAlb-bound albumin can enter tumor cells. So how can nAlb and the drug conjugates be accumulated in tumors through albumin-mediated endocytosis? The majority of albumin did not contain nanobody.

We thank the reviewer for this insightful question that also reflects the field’s still emerging understanding of how albumin accumulates in tumors and its utility as a drug carrier. As discussed in several outstanding reviews (5, 6) and described on **Pages 7 and 9** and below, there are several interrelated mechanisms by which albumin accumulates in tumors, including passive diffusion through leaky tumor vasculature (i.e., EPR effect), active accumulation via transcytosis, tumor retention via binding to SPARC secreted by cancer cells, and albumin endocytosis and catabolism by cancer and immune cells. The relative contribution of these factors is thought to be dependent on several variables, including tumor type, mutational status, immune composition, and vascular permeability, amongst other variables. As the reviewer accurately notes, only a small fraction of circulating albumin will be bound to nAlb upon administration and a subfraction of this will access and accumulate in tumor tissue via the mechanisms described above. In new experiments, we have quantified the amount of Cy5 that accumulates in EMT6 breast tumors 24 h following intravenous administration of nAlb-Cy5 and found ~11% injected dose/g tumor tissue, more than in other tissues (**Figure 2h**); similar results were observed in the B16.F10

model (**Figure 2i**). This is consistent with IVIS imaging data (**Figure 2** and **Figure 4**) and with previous literature demonstrating that albumin (and albumin binders) can accumulate in tumors. Note also that this represents a single snapshot in time and does not account for nAlb-Cy5 that may have accessed the tumor but was catabolized and cleared in the preceding 24h.

Page 7: While still incompletely understood and variable across cancers and patients²⁹, albumin can accumulate in tumor tissue through several interrelated mechanisms, including passive diffusion through leaky tumor vasculature, active transport via endothelial cell transcytosis, binding to SPARC (secreted protein acidic and rich in cysteine) produced by cancer cells, and cellular uptake and catabolism by cancer and tumor-associated immune cells such as macrophages.^{27, 29} Albumin has been reported to enter cancer cells and tumor-associated myeloid cells (e.g., macrophages) through both albumin-dependent, receptor-mediated pathways as well as by micropinocytosis.^{27, 29}

Page 9: “Albumin binding to SPARC expressed in tumor tissue has also been implicated in increased accumulation of albumin-binding therapeutics,²⁷ and we found that SPARC is expressed in both EMT6 and B16.F10 tumors (**Fig. S11**) and may therefore contribute to nAlb accumulation.”

Figure 2: (h-i) Quantification of percent injected dose per gram of tissue (% ID/g) 24 h following intravenous administration of vehicle (PBS) and nAlb-Cy5 at 2 mg/kg to **(h)** female Balb/c mice with orthotopic EMT6 breast tumors (n=5) and **(i)** female C57BL/6 mice with subcutaneous B16.F10 tumors (n=5). *P* values determined by two-way ANOVA with post-hoc Tukey’s correction for multiple comparisons; **P*≤0.05, ***P*≤0.01, ****P*≤0.001, and *****P*<0.0001.

Upon entering the tumor, nAlb-diABZI must be internalized to activate STING and our flow cytometric analysis indicates that nAlb-Cy5 is endocytosed by a relatively small percentage (~10-15%) of several cell types including CD45⁺ cells, which are predominantly cancer cells, as well as myeloid cells, including macrophages, DCs, and neutrophils (**Figure 2i**). While the mechanisms by which these different cell types endocytose albumin or nAlb is not well understood, new data collected during revisions indicates that endocytosis of nAlb-Cy5 in EMT6 cells and myeloid cells occurs mostly *independently* of albumin (**Extended Data Fig. 1b**). As described in more detail in responses to questions below, we believe that our interpretation of our original data showing that nAlb-Cy5 uptake by EMT6 cells was enhanced in the presence of albumin was likely an experimental artifact caused by overnight serum starvation that may have impacted endocytosis pathways. In our new studies, we examined this more carefully in both EMT6 and myeloid cells by adding Cy5-labeled nanobody targeting GFP (nGFP-Cy5) as a non-albumin-binding control, which we found to be endocytosed to a comparable extent as nAlb-Cy5 (**Extended Data Fig. 1b**). Taken together, these data suggest that nAlb-diABZI is internalized by multiple cell populations within the TME, mostly by myeloid cells, in a primarily albumin-independent manner. Additionally, in comparing the relative responsiveness of BMDMs and EMT6 cells to nAlb-diABZI using PCR to quantify STING-mediated gene expression, we found BMDMs to be substantially more responsive (**Figure R3**). This suggests that endocytosis by myeloid cells may be more important to STING activation in tumor tissue than uptake by cancer cells, at least in this tumor model. Macropinocytosis has also been implicated as a major mechanism of albumin uptake by cancer and tumor-associated macrophages (5), and we therefore treated EMT6 cells, RAW264.7 macrophages, and BMDMs with the micropinocytosis inhibitor 5-[N-ethyl-N-isopropyl] amiloride (EIPA) which significantly reduced uptake nAlb-Cy5 (**Figure 2b**). Taken together, these data suggest that nAlb-diABZI is internalized by multiple cell populations within the TME, mostly by myeloid cells, predominantly via non-receptor-mediated macropinocytosis. Therefore, if micropinocytosis is the predominant mechanism of

cellular uptake, this would eliminate or reduce competition between free albumin and nAlb-bound albumin for receptor-mediated uptake since micropinocytosis can result in rapid cellular internalization of multiple nAlb-Cy5 molecules per endocytic event from surrounding extracellular space. Phagocytes have been reported to internalize the equivalent of their entire surface area approximately every 30 minutes (28), allowing for near continuous uptake of nAlb that accumulates in tumor tissue over time. Additionally, in new experiments we have demonstrated that SPARC is expressed in EMT6 and B16.F10 tumors (**Figure S11**), which has been implicated in binding and retaining albumin in tumor tissue to increase local concentrations and therefore increased rates of macropinocytotic uptake. We thank the reviewer for this astute analysis that compelled us to reconsider our in vitro data, to conduct new studies with improved controls, and to analyze our findings more carefully through this lens. In addition to new and revised figures, we have revised the text on **Pages 7 and 8** as followed:

Pages 7 and 8: “While still incompletely understood and variable across cancers and patients²⁹, albumin can accumulate in tumor tissue through several interrelated mechanisms, including passive diffusion through leaky tumor vasculature, active transport via endothelial cell transcytosis, binding to SPARC (secreted protein acidic and rich in cysteine) produced by cancer cells, and cellular uptake and catabolism by cancer and tumor-associated immune cells such as macrophages.^{27, 29} Albumin has been reported to enter cancer cells and tumor-associated myeloid cells (e.g., macrophages) through both albumin-dependent, receptor-mediated pathways as well as by micropinocytosis.^{27, 29} Though mechanisms of cellular albumin internalization may vary between tumor and cell types, we sought to gain insight into how nAlb-diABZI enters cells and activates STING. First, we first validated that intracellular uptake of nAlb-Cy5 was abrogated at 4 °C indicating an active endocytotic mechanism (**Fig. 2a, Extended Data Fig. 1**); by contrast diABZI can enter cells by passive transport across the plasma membrane.²³ We next assessed if albumin binding enhanced nAlb internalization by EMT6 and myeloid cells. To test this, we first used flow cytometry compare the cellular uptake of nAlb-Cy5 to a negative control nanobody targeting GFP, nGFP-Cy5 (**Fig. S5**), in serum containing media, finding insignificant or minor differences in cellular uptake between nAlb-Cy5 and nGFP-Cy5 (**Extended Data Fig. 1**). While eliminating serum from culture media decreased nAlb-Cy5 uptake, this occurred to the same extent for nGFP-Cy5, again indicating that cellular uptake occurs predominantly in an albumin receptor-independent manner in these cell types. Albumin can also be internalized by cancer and tumor-associated immune cell populations through non-receptor-mediated micropinocytosis. To evaluate this, we inhibited micropinocytosis in EMT6 cells, RAW264.7 macrophages, and BMDMs using 5-[N-ethyl-N-isopropyl] amiloride (EIPA), which significantly reduced nAlb-Cy5 uptake (**Fig. 2b**). Given that macropinosomes often traffic to lysosomes, we next assessed colocalization of nAlb-Cy5 with lysotracker and found that a substantial and similar fraction (>50%) of nAlb-Cy5 and nGFP-Cy5 was colocalized with lysosomes or late endosomes in EMT6 and RAW264.7 cells (**Fig. 2c,d**). As expected, nAlb-diABZI did not mediate endosomal disruption as assessed using a previously described galectin 8 (Gal8) endosomal recruitment assay (**Extended Data Fig. 1**)⁴¹.”

Figure R3: qPCR analysis of *Ifnb* gene expression in EMT6 and B16.F10 cells at 6 h. *P* value determined by Student’s t-test, *n*=3 per group with ***P*≤0.01 and ****P*≤0.001. Replicates are noted as biological, and data shown as mean ± SEM.

Extended Data Figure 1: *In vitro* analysis of nanobody delivery. **(a)** Uptake of MSA-AF647 (1 μ M) in BMDM cells as well as **(b)** nAlb-Cy5 (2 μ M) and nGFP-Cy5 (2 μ M) in EMT6 cells with the addition of control PBS (+serum) or in a serum starved condition (-serum). *P* values determined by two-way ANOVA with post-hoc Tukey's correction for multiple comparisons; *****P*<0.0001. **(c)** Uptake of MSA-AF647 (1 μ M) in BMDC cells and **(d)** EMT6 cells at 37 °C and 4 °C. *P* values determined by Student's *t*-test. **(e-g)** Uptake of nAlb-Cy5 (2 μ M) in **(e)** BMDC cells, **(f)** BMDM cells, and **(g)** EMT6 cells at 37 °C and 4 °C. *P* values determined by Student's *t*-test. **(h)** Integrated pixel intensity of Gal8-GFP puncta, per cell, for DBCO-PEG11-diABZI, nAlb-diABZI, and AP-diABZI at 0.25 μ M (mean \pm SEM, *n*=3-9). *P* values determined via one-way ANOVA, with Tukey's post-hoc.

Figure 2. **(a)** Dose-response curve for nanobody-Cy5 conjugate surface binding and intracellular uptake at 37 °C and 4 °C measured by flow cytometry in EGFR⁻ (THP-1) *in vitro*. **(b)** Uptake of nAlb-Cy5 (2 μ M) in RAW 264.7, EMT6, and BMDM cells with the addition of control PBS (-EIPA) or macropinocytosis inhibitor (+EIPA). **(c)** Colocalization of Cy5 (red) with lysotracker green (green) and Hoechst (blue) in RAW 264.7 cells with **(d)** percent colocalization determination for nAlb-Cy5 and nGFP-Cy5 in RAW 264.7 and EMT6 cells. (scale bar: 100 μ m)

Figure S11: **(a)** Western blot analysis of SPARC (42 kDa) and HSP-90 (90 kDa) proteins in B16.F10 and EMT6 tissue. **(b)** Representative fluorescent microscopy images of tumor sections stained for DAPI (blue), CD31 (magenta), and SPARC (white) 24 h following administration of PBS, nAlb-Cy5, or nAlb-diABZI in combination with nAlb-diABZI (scale bar: 100 μ m).

2. The ligation of diABZI to nanobody was through click chemistry with DBCO-PEG11-diABZI, so the drug was linked to the nanobody via a covalent bond. Then, how can the drug release from the protein and exert its function? Lysosomal degradation was postulated in the manuscript; the authors should provide evidence of the drug release and the format (structure) of the released product – whether this product is as effective as diABZI?

We thank the reviewer for this important question. In our design, we used a stable covalent bond between the nanobody and the diABZI compound and integrated a PEG spacer to improve solubility. While our approach is highly amenable to integration of cleavable linkers (e.g., cathepsin B cleavable), we elected to start with a stable linker for simplicity and to create a platform from to further build and optimize; additionally, the use of stable linkers minimizes premature drug release, which is an established limitation of some linkers used for development of antibody-drug conjugates (27). We postulated that diABZI was released following lysosomal degradation and we conducted several new experiments to evaluate this. First, we demonstrated using fluorescent microscopy that nAlb-Cy5 colocalizes with lysosomes (Mander's overlap coefficient of >0.5) as measured using a LysoTracker dye (**Figure 2c,d, Extended Data Fig. 1i**). Based on this, we next incubated nAlb-diABZI with rat liver tritosomes, which are isolated and stabilized lysosomes and employed to evaluate the stability of protein-drug conjugates, and we used MALDI to evaluate the emergence of potential cleavage products (**Figure S6**). As early as 1 h following incubation with tritosomes, we observed the formation of an NH₂-PEG₃-triazole-DBCO-PEG₁₁-diABZI product (**Figure S7**) corresponding to cleavage of the amide bond ligating the diABZI molecule to the nanobody. We then synthesized and purified this product (**Figure S7**) and tested *in vitro* activity compared to the previously reported diABZI compound in THP1 IFN-I reporter cells and found that the released compound was a highly potent STING agonist with comparable activity to diABZI (**Figure S10**). This is described in revised text on **Page 8** as followed:

Page 8: “Given that macropinosomes often traffic to lysosomes, we next assessed colocalization of nAlb-Cy5 with lysotracker and found that a substantial and similar fraction (>50%) of nAlb-Cy5 and nGFP-Cy5 was colocalized with lysosomes or late endosomes in EMT6 and RAW264.7 cells (**Fig. 2c,d**).” As expected, nAlb-diABZI did not mediate endosomal disruption as assessed using a previously described galectin 8 (Gal8) endosomal recruitment assay (**Extended Data Fig. 1**)⁴¹.

To gain insight into how amide-linked diABZI is released from the nanobody upon cellular internalization, we incubated nAlb-diABZI with lysosomes isolated from rat liver (Tritosomes), which are used to investigate stability and catabolism of molecules trafficked to an endosome-lysosome pathway, and used MALDI mass spectroscopy to assess the emergence of a PEGylated diABZI adduct that would be predicted due to amide bond cleavage by lysosomal proteases (**Fig. S6**). We observed the presence of this peak as early as 1 h following incubation with Tritosomes, suggesting that a fraction of nAlb-diABZI is lysosomally degraded to release a PEGylated diABZI variant. We synthesized this compound (**Fig. S7-9**) and evaluated *in vitro* activity in THP1 IFN-I reporter cells, finding that it had a similar EC₅₀ value to the previously described diABZI molecule, which can enter cells through passive transport²³ (**Fig. S10**).”

Figure 2. (c) Colocalization of Cy5 (red) with lysotracker green (green) and Hoechst (blue) in RAW 264.7 cells with **(d)** percent colocalization determination for nAlb-Cy5 and nGFP-Cy5 in RAW 264.7 and EMT6 cells. (scale bar: 100 μ m)

Extended Data Figure 1: (i) Colocalization analysis of nAlb-Cy5 (2 μ M) and nGFP-Cy5 (2 μ M) for Raw 264.7 cells to visualize the cell nuclei (Hoechst, blue), lysosomes (Lysogreen, green), and Cy5 (Cy5, red). Merged images of the stains provided, in addition to a 2.5x zoom of the merged image (white box). (scale bar: 100 μ m)

Figure S6: Tritosome degradation assay for (a) diABZI-amine, (b) DBCO-PEG₁₁-diABZI, and (c-d) nAlb-amine with highlighted molecular weight ranges at 780 Da (diABZI-amine construct, blue dashed line), 1688 Da (DBCO-PEG₁₁-diABZI construct, orange dashed line), and 1906 Da (Amine-PEG₃-Triazole-PEG₁₁-diABZI).

Figure S7: Synthesis of Amine-PEG₃-Triazole-PEG₁₁-diABZI.

Figure S10: *In vitro* dose-response curves for THP1-Dual reporter cells comparing diABZI to Amine-PEG₃-Triazole-PEG₁₁-diABZI.

3. The albumin-mediated endocytosis of the nanobody was measured on nAlb-Cy5 and nEGFR-Cy5 using anti-SPARC and anti-GP60 antibodies; it is unknown whether the endocytosis efficiency can be affected by nPD-L1 linking and PEG₁₁-diABZI modification, as nPD-L1 can also induce endocytosis in a different way. Indeed, the effect of nPD-L1 on endocytosis of the bivalent nanobody should be analyzed.

As described above, the Reviewer's question challenged us to rethink our analysis and to conduct additional and more controlled *in vitro* studies to investigate cellular uptake in EMT6 cancer cells and BMDMs. We did not observe a significant difference in nAlb-Cy5 uptake compared to a non-albumin-binding nGFP-Cy5 control in both serum containing and serum free media (**Extended Data Fig. 1b**) Therefore, while SPARC and gp60 may play roles in nAlb accumulation *in vivo*, our new data indicates that nAlb is internalized in a primarily non-albumin-dependent manner *in vitro*. Nonetheless, we sought to further demonstrate that the nPD-L1 domain could promote receptor-mediated endocytosis. To do this, we evaluated AP-Cy5 uptake in PD-L1 knockout EMT6 cells and found reduced uptake compared to the parental EMT6 line and, interestingly, to a level similar to nAlb-Cy5 (**Figure 4i**). Additionally, we observed reduced AP-Cy5 accumulation in tumors established with PD-L1 KO EMT6 cells (**Figure 4m**).

Figure 4: **(i)** Mean fluorescent intensity (MFI) for nAlb-Cy5 and AP-Cy5 conjugate surface binding at 2 μ M compared to PBS (0 μ M) for EMT6 W.T. and EMT6 PD-L1 K.O. cell lines at 37 $^{\circ}$ C. **(m)** Representative IVIS fluorescent images of excised tumors and major organs (left) and quantification of average radiant efficiencies (right) of tumors and major organs 48 h after administration of AP-Cy5 in mice with EMT6 W.T. and EMT6 PD-L1 K.O. breast tumors ($n=5$). P values determined by one-way ANOVA with post-hoc Tukey's correction for multiple comparisons; * $P \leq 0.05$, ** $P \leq 0.01$, *** $P \leq 0.001$, and **** $P < 0.0001$ compared to the tumor group. Replicates are noted as biological, and data shown as mean \pm SEM.

4. In Figure 2b, the two types of cells have different EGFR levels, but the levels of the SPARC and GP60 are unknown. The membrane binding of nEGFR to EGFR+ cells was observed, and nAlb did not bind to any cells. Is that because of low levels of albumin receptors?
The method to measure membrane binding and endocytosis of nanobody was indirect. The authors may use confocal fluorescence imaging to directly view the result.

The goal of these studies was to demonstrate that integration of a cell targeting domain could enhance uptake relative to nAlb alone and we selected two cell types with different levels of EGFR expression to test this, finding, as expected, increased uptake of nEGFR in the EGFR⁺ cell line. However, because EGFR was only a model target that was not of relevance to our application, and based on the findings described above regarding nAlb internalization, we have removed the previous Figure 2b from the manuscript but include it below as **Figure R4**. Instead, we directly investigated the effect of adding the PD-L1 binding domain in experiments using cells with low (B16.F10) (**Figure 4g**) or high (EMT6) (**Figure 4h**) levels of PD-L1 as well as PD-L1 KO cells (**Figure 4i**) to compare levels of cellular internalization mediated via receptor- and non-receptor-mediated endocytosis. As described in our responses to questions above, new studies with improved controls indicate that nAlb is internalized primarily via non-receptor-mediated, albumin-independent mechanisms and therefore differences in cellular uptake observed in this study can be attributed to PD-L1 (and EGFR binding in **Figure R4**) and subsequent internalization. Lack or a low density of albumin receptors also explains the negligible cell surface binding of nAlb at 4°C for all cell types investigation. Therefore, like the question regarding PD-L1 binding above, our new studies suggest that nAlb is internalized primarily via micropinocytosis whereas uptake of AP and nEGFR is enhanced by receptor binding and endocytosis.

Figure R4: Dose-response curve for nanobody-Cy5 conjugate surface binding and intracellular uptake at 37 °C and 4 °C measured by flow cytometry in EGFR⁻ (THP-1) and EGFR⁺ (A549) cells *in vitro*.

Figure 4: (g-h) Dose response curve for nAlb-Cy5 and AP-Cy5 conjugate surface binding and intracellular uptake at 37 °C and 4 °C measured by flow cytometry in (g) B16.F10 cells (n=2-3) and (h) EMT6 cells (n=3). (i) Mean fluorescent intensity (MFI) for nAlb-Cy5 and AP-Cy5 conjugate surface binding at 2 μM compared to PBS (0 μM) for EMT6 W.T. and EMT6 PD-L1 K.O. cell lines at 37 °C.

5. Figure 2c, the addition of anti-SPARC and anti-GP60 antibodies decreased the internalization of nAlb-Cy5. It can be seen that the internalization was influenced by antibodies more significantly in B16.F10 cells than in EMT6 cells. Can this result suggest higher efficiency of endocytosis in B16.F10 cells? Could this difference affect the *in vivo* treatment?

As described in response to questions above, new studies with improved controls indicate that nAlb-Cy5 is internalized primarily via albumin-independent mechanisms. In comparing the relative uptake of nAlb-Cy5 in B16 and EMT6 cells using flow cytometry, we found greater uptake in EMT6 cells suggesting a higher capacity for micropinocytosis (**Figure R5**). While this could impact *in vivo* efficacy, flow cytometric analysis suggests that nAlb-Cy5 is predominantly internalized by tumor-associated myeloid cells such as macrophages and neutrophils (see **Figure 2i**). Additionally, in our analysis of nAlb-Cy5 biodistribution, we found ~11% ID/gram tumor in the EMT6 model and a comparable ~8.4% ID/gram tumor in B16.10 model (**Figure 2h,i**), yet a substantial difference in response to both nAlb-diABZI alone and in combination with anti-PD-L1 (**Figure 5, Figure 7, Figure 8**) that

can most likely be attributed to the low immunogenicity to B16.F10 tumors. We feel that this is an important point and so we have expanded upon our Discussion on **Page 30** as followed:

Page 30: “By contrast, immunostimulatory agents such as STING agonists can stimulate complex antitumor immunological programs that may be more dependent on immunological variables (e.g., neoantigen load, immune status of the TME) than on the efficiency of drug accumulation in tumor tissue or delivery to cancer cells. For example, in our analysis of nAlb-Cy5 biodistribution, we found ~11% ID/gram tumor in the EMT6 model and a comparable ~8.4% ID/gram tumor in B16.10 model (**Fig. 2**), yet a substantial difference in response to both nAlb-diABZI alone and in combination with anti-PD-L1 that may be attributed to the relatively low immunogenicity to B16.F10 tumors.”

Figure 1: Mean fluorescent intensity (MFI) for nAlb-Cy5 surface binding and uptake at 2 μ M for EMT6 and B16.F10 cell lines at 37 °C and 4 °C. *P* value determined by two-way ANOVA with post-hoc Tukey’s correction for multiple comparisons.

I

Figure 2: (I) Flow cytometric analysis of nAlb-Cy5 cellular uptake by in EMT6 tumors evaluated as the percentage of Cy5⁺ cells within an indicated live cell population 24 h following administration of vehicle (PBS), nAlb-Cy5 alone, or nAlb-Cy5 co-administered with nAlb-diABZI.

Figure 2: (h-i) Quantification of percent injected dose per gram of tissue (% ID/g) 24 h following intravenous administration of vehicle (PBS) and nAlb-Cy5 at 2 mg/kg to **(h)** female Balb/c mice with orthotopic EMT6 breast tumors (n=5) and **(i)** female C57BL/6 mice with subcutaneous B16.F10 tumors (n=5). *P* values determined by two-way ANOVA with post-hoc Tukey's correction for multiple comparisons; **P*≤0.05, ***P*≤0.01, ****P*≤0.001, and *****P*<0.0001.

6. Figure 4c: the mass spectrum of the final product AP-diABZI was missing; SDS PAGE cannot tell the homogeneity (ligation ratio) of the product.

We thank the reviewer for pointing this out. We added the ESI mass spectra for AP-diABZI in **Figure 4c**.

Figure 4: (c) ESI-MS confirming the purity and molecular weight of AP conjugates.

7. In the study of “nAlb-diABZI potently stimulates STING activation in the TME to inhibit tumor growth”, the authors analyzed the effect of nAlb-diABZI on B16.F10 (PD-L1 low) and EMT6 (PD-L1 high) tumors. It seems nAlb-diABZI was effective on both types of tumors. Should PD-L1 level affect the treatment? Also in the study of immunological memory in the next sections, were there differences between the two types of cells? The authors should make a comparison and give conclusions.

We thank the reviewer for this question. In our studies, nAlb-diABZI was effective in both models but more effective in the EMT6 model. B16.F10 tumors are considered poorly immunogenic and less responsive to immunotherapy than EMT6 tumors, which respond partially to anti-PD-L1 immune checkpoint blockade (e.g., **Figure 5b**). While the expression level of PD-L1 on cancer cells can influence responses to immunotherapies this is often not predictive of response (29) as there are many other factors that can also contribute, including PD-L1 expression by tumor-associated myeloid cells, frequency and immunogenicity of tumor antigens, and the immunocellular composition of the TME, amongst other variables. In response to other reviewer comments, we conducted new experiments comparing the tumor accumulation (**Figure 4m**) and efficacy (**Figure 5g,h**) of AP-diABZI in mice with tumors established using PD-L1 knockout or parental EMT-6 cells and found reduced tumor accumulation and efficacy in the PD-L1 knockout model. While these experiments were not performed using nAlb-diABZI, they suggest that PD-L1 expression level by cancer cells may influence response to albumin-hitchhiking STING agonists. However, given the complex interactions that underlie tumor immunogenicity and response to immunotherapy, significant additional experiments would be necessary to clearly demonstrate this, which we believe is beyond the scope of the current manuscript.

Figure 4: (m) Representative IVIS fluorescent images of excised tumors and major organs (left) and quantification of average radiant efficiencies (right) of tumors and major organs 48 h after administration of AP-Cy5 in mice with EMT6 W.T. and EMT6 PD-L1 K.O. breast tumors (n=5). *P* values determined by one-way ANOVA with post-hoc Tukey's correction for multiple comparisons; **P*≤0.05, ***P*≤0.01, ****P*≤0.001, and *****P*<0.0001 compared to the tumor group. Replicates are noted as biological, and data shown as mean ± SEM.

Figure 5: (g) Scheme of EMT6 W.T. and EMT6 PD-L1 K.O. tumor inoculation and treatment schedule (n=5-13). AP-diABZI was injected I.V. at 1.25 µg of diABZI. **(h)** Kaplan-Meier survival plots for mice with EMT6 W.T. or PD-L1 K.O. tumors treated as indicated.

Regarding differences in immunological memory in the two models, we were only able to assess memory in the EMT6 model since we did not observe complete responses in the more challenging B16.F10 model and, therefore, were not able to rechallenge mice with cancer cells to provide evidence of immunological memory. Flow cytometric immunophenotyping of the splenic T cells following treatment with AP-diABZI (**Figure 7m, Extended Data Fig. 10**) demonstrated induction of an antigen-specific effector memory CD8⁺ T cell response in the B16.F10-OVA model, but we were unable to evaluate the function of this response via tumor rechallenge studies and therefore cannot draw conclusions regarding the relative memory response induced in the two tumor models.

8. Figure 4m, among the ratio range of 0 – 50 indicated by color gradients, it would be helpful to indicate the color code for 1.

In our revised version of Figure 4, we have removed this heat map which provided a summary of the data in current Figure 4k.

References for Response to Reviewers:

1. K. D. Moynihan, C. F. Opel, G. L. Szeto, A. Tzeng, E. F. Zhu, J. M. Engreitz, R. T. Williams, K. Rakhra, M. H. Zhang, A. M. Rothschilds, S. Kumari, R. L. Kelly, B. H. Kwan, W. Abraham, K. Hu, N. K. Mehta, M. J. Kauke, H. Suh, J. R. Cochran, D. A. Lauffenburger, K. D. Wittrup, D. J. Irvine, Eradication of large established tumors in mice by combination immunotherapy that engages innate and adaptive immune responses. *Nat Med* **22**, 1402-1410 (2016).
2. K. D. Moynihan, D. J. Irvine, Roles for Innate Immunity in Combination Immunotherapies. *Cancer Res* **77**, 5215-5221 (2017).
3. J. Galon, D. Bruni, Approaches to treat immune hot, altered and cold tumours with combination immunotherapies. *Nat Rev Drug Discov*, (2019).
4. R. Zappasodi, T. Merghoub, J. D. Wolchok, Emerging Concepts for Immune Checkpoint Blockade-Based Combination Therapies. *Cancer Cell* **33**, 581-598 (2018).
5. H. Hu, J. Quintana, R. Weissleder, S. Parangi, M. Miller, Deciphering albumin-directed drug delivery by imaging. *Adv Drug Deliv Rev* **185**, 114237 (2022).
6. E. N. Hoogenboezem, C. L. Duvall, Harnessing albumin as a carrier for cancer therapies. *Adv Drug Deliv Rev* **130**, 73-89 (2018).
7. J. M. Ramanjulu, G. S. Pesiridis, J. Yang, N. Concha, R. Singhaus, S. Y. Zhang, J. L. Tran, P. Moore, S. Lehmann, H. C. Eberl, M. Muelbaier, J. L. Schneck, J. Clemens, M. Adam, J. Mehlmann, J. Romano, A. Morales, J. Kang, L. Leister, T. L. Graybill, A. K. Charnley, G. Ye, N. Nevins, K. Behnia, A. I. Wolf, V. Kasparcova, K. Nurse, L. Wang, A. C. Puhl, Y. Li, M. Klein, C. B. Hopson, J. Guss, M. Bantscheff, G. Bergamini, M. A. Reilly, Y. Lian, K. J. Duffy, J. Adams, K. P. Foley, P. J. Gough, R. W. Marquis, J. Smothers, A. Hoos, J. Bertin, Design of amidobenzimidazole STING receptor agonists with systemic activity. *Nature* **564**, 439-443 (2018).
8. K. V. Kilchrist, S. C. Dimobi, M. A. Jackson, B. C. Evans, T. A. Werfel, E. A. Dailing, S. K. Bedingfield, I. B. Kelly, C. L. Duvall, Gal8 Visualization of Endosome Disruption Predicts Carrier-Mediated Biologic Drug Intracellular Bioavailability. *ACS Nano* **13**, 1136-1152 (2019).
9. M. J. Munson, G. O'Driscoll, A. M. Silva, E. Lazaro-Ibanez, A. Gallud, J. T. Wilson, A. Collen, E. K. Esbjorner, A. Sabirsh, A high-throughput Galectin-9 imaging assay for quantifying nanoparticle uptake, endosomal escape and functional RNA delivery. *Commun Biol* **4**, 211 (2021).
10. J. J. Baljon, A. J. Kwiatkowski, H. M. Pagendarm, P. T. Stone, A. Kumar, V. Bharti, J. A. Schulman, K. W. Becker, E. W. Roth, P. P. Christov, S. Joyce, J. T. Wilson, A Cancer Nanovaccine for Co-Delivery of Peptide Neoantigens and Optimized Combinations of STING and TLR4 Agonists. *ACS Nano* **18**, 6845-6862 (2024).
11. H. Liang, L. Deng, Y. Hou, X. Meng, X. Huang, E. Rao, W. Zheng, H. Mauceri, M. Mack, M. Xu, Y. X. Fu, R. R. Weichselbaum, Host STING-dependent MDSC mobilization drives extrinsic radiation resistance. *Nat Commun* **8**, 1736 (2017).
12. H. Lemos, E. Mohamed, L. Huang, R. Ou, G. Pacholczyk, A. S. Arbab, D. Munn, A. L. Mellor, STING Promotes the Growth of Tumors Characterized by Low Antigenicity via IDO Activation. *Cancer Res* **76**, 2076-2081 (2016).
13. S. F. Bakhom, B. Ngo, A. M. Laughney, J. A. Cavallo, C. J. Murphy, P. Ly, P. Shah, R. K. Sriram, T. B. K. Watkins, N. K. Taunk, M. Duran, C. Pauli, C. Shaw, K. Chadalavada, V. K. Rajasekhar, G. Genovese, S. Venkatesan, N. J. Birkbak, N. McGranahan, M. Lundquist, Q. LaPlant, J. H. Healey, O. Elemento, C. H. Chung, N. Y. Lee, M. Imielenski, G. Nanjangud, D. Pe'er, D. W. Cleveland, S. N. Powell, J. Lammerding, C. Swanton, L. C. Cantley, Chromosomal instability drives metastasis through a cytosolic DNA response. *Nature* **553**, 467-472 (2018).
14. Q. Chen, A. Boire, X. Jin, M. Valiente, E. E. Er, A. Lopez-Soto, L. Jacob, R. Patwa, H. Shah, K. Xu, J. R. Cross, J. Massague, Carcinoma-astrocyte gap junctions promote brain metastasis by cGAMP transfer. *Nature* **533**, 493-498 (2016).
15. T. Xu, J. Dai, L. Tang, L. Sun, L. Si, J. Guo, Systemic administration of STING agonist promotes myeloid cells maturation and antitumor immunity through regulating hematopoietic stem and progenitor cell fate. *Cancer Immunol Immunother* **72**, 3491-3505 (2023).
16. F. Veglia, E. Sanseviero, D. I. Gabrilovich, Myeloid-derived suppressor cells in the era of increasing myeloid cell diversity. *Nat Rev Immunol* **21**, 485-498 (2021).

17. V. M. Kho, V. E. Meekers, P. N. Span, J. Bussink, G. J. Adema, Radiotherapy and cGAS/STING signaling: Impact on MDSCs in the tumor microenvironment. *Cell Immunol* **362**, 104298 (2021).
18. Y. Wang, S. Li, M. Hu, Y. Yang, E. McCabe, L. Zhang, A. M. Withrow, J. P. Ting, R. Liu, Universal STING mimic boosts antitumour immunity via preferential activation of tumour control signalling pathways. *Nat Nanotechnol*, (2024).
19. K. Yang, W. Han, X. Jiang, A. Piffko, J. Bugno, C. Han, S. Li, H. Liang, Z. Xu, W. Zheng, L. Wang, J. Wang, X. Huang, J. P. Y. Ting, Y. X. Fu, W. Lin, R. R. Weichselbaum, Zinc cyclic di-AMP nanoparticles target and suppress tumours via endothelial STING activation and tumour-associated macrophage reinvigoration. *Nat Nanotechnol* **17**, 1322-1331 (2022).
20. D. Shae, K. W. Becker, P. Christov, D. S. Yun, A. K. R. Lytton-Jean, S. Sevimli, M. Ascano, M. Kelley, D. B. Johnson, J. M. Balko, J. T. Wilson, Endosomolytic polymersomes increase the activity of cyclic dinucleotide STING agonists to enhance cancer immunotherapy. *Nat Nanotechnol* **14**, 269-278 (2019).
21. E. L. Dane, A. Belessiotis-Richards, C. Backlund, J. Wang, K. Hidaka, L. E. Milling, S. Bhagchandani, M. B. Melo, S. Wu, N. Li, N. Donahue, K. Ni, L. Ma, M. Okaniwa, M. M. Stevens, A. Alexander-Katz, D. J. Irvine, STING agonist delivery by tumour-penetrating PEG-lipid nanodiscs primes robust anticancer immunity. *Nat Mater* **21**, 710-720 (2022).
22. U. Sahin, P. Oehm, E. Derhovanessian, R. A. Jabulowsky, M. Vormehr, M. Gold, D. Maurus, D. Schwarck-Kokarakis, A. N. Kuhn, T. Omokoko, L. M. Kranz, M. Diken, S. Kreiter, H. Haas, S. Attig, R. Rae, K. Cuk, A. Kemmer-Bruck, A. Breitkreuz, C. Tolliver, J. Caspar, J. Quinkhardt, L. Hebich, M. Stein, A. Hohberger, I. Vogler, I. Liebig, S. Renken, J. Sikorski, M. Leierer, V. Muller, H. Mitzel-Rink, M. Miederer, C. Huber, S. Grabbe, J. Utikal, A. Pinter, R. Kaufmann, J. C. Hassel, C. Loquai, O. Tureci, An RNA vaccine drives immunity in checkpoint-inhibitor-treated melanoma. *Nature* **585**, 107-112 (2020).
23. U. Sahin, E. Derhovanessian, M. Miller, B. P. Kloke, P. Simon, M. Lower, V. Bukur, A. D. Tadmor, U. Luxemburger, B. Schrors, T. Omokoko, M. Vormehr, C. Albrecht, A. Paruzynski, A. N. Kuhn, J. Buck, S. Heesch, K. H. Schreeb, F. Muller, I. Ortseifer, I. Vogler, E. Godehardt, S. Attig, R. Rae, A. Breitkreuz, C. Tolliver, M. Suchan, G. Martic, A. Hohberger, P. Sorn, J. Diekmann, J. Ciesla, O. Waksman, A. K. Bruck, M. Witt, M. Zillgen, A. Rothermel, B. Kasemann, D. Langer, S. Bolte, M. Diken, S. Kreiter, R. Nemecek, C. Gebhardt, S. Grabbe, C. Holler, J. Utikal, C. Huber, C. Loquai, O. Tureci, Personalized RNA mutanome vaccines mobilize poly-specific therapeutic immunity against cancer. *Nature* **547**, 222-226 (2017).
24. J. R. Duvall, J. D. Thomas, R. A. Bukhalid, K. C. Catcott, K. W. Bentley, S. D. Collins, T. Eitas, B. D. Jones, E. W. Kelleher, K. Lancaster, M. Protopopova, S. S. Ray, E. Ter-Ovanesyan, L. Xu, L. Yang, J. Zurita, M. Damelin, D. Toader, T. B. Lowinger, Discovery and Optimization of a STING Agonist Platform for Application in Antibody Drug Conjugates. *J Med Chem* **66**, 10715-10733 (2023).
25. R. C. Roovers, M. J. Vosjan, T. Laeremans, R. el Khoulati, R. C. de Bruin, K. M. Ferguson, A. J. Verkleij, G. A. van Dongen, P. M. van Bergen en Henegouwen, A biparatopic anti-EGFR nanobody efficiently inhibits solid tumour growth. *Int J Cancer* **129**, 2013-2024 (2011).
26. K. Broos, M. Keyaerts, Q. Lecocq, D. Renmans, T. Nguyen, D. Escors, A. Liston, G. Raes, K. Breckpot, N. Devoogdt, Non-invasive assessment of murine PD-L1 levels in syngeneic tumor models by nuclear imaging with nanobody tracers. *Oncotarget* **8**, 41932-41946 (2017).
27. K. Tsuchikama, Z. An, Antibody-drug conjugates: recent advances in conjugation and linker chemistries. *Protein Cell* **9**, 33-46 (2018).
28. R. M. Steinman, S. E. Brodie, Z. A. Cohn, Membrane flow during pinocytosis. A stereologic analysis. *J Cell Biol* **68**, 665-687 (1976).
29. A. A. Davis, V. G. Patel, The role of PD-L1 expression as a predictive biomarker: an analysis of all US Food and Drug Administration (FDA) approvals of immune checkpoint inhibitors. *J Immunother Cancer* **7**, 278 (2019).

Response to Reviewers:

The authors thank the reviewer for their constructive feedback. We have made changes accordingly in the revision of the manuscript. Specific alterations relating to reviewer comments along with the original comments or critiques are provided below. A revised version of the manuscript is provided where changes have been highlighted. We believe that revisions made in response to the reviewer comments have strengthened this work. We hope that the comments and questions raised by the reviewers have all been addressed in a satisfactory manner and that this revised manuscript is acceptable for publication in *Nature Biomedical Engineering*.

Reviewer #1 (Report for the authors (Required)):

This paper has been considerably improved. However, not all remarks have been answered satisfactorily and additional experiments also raised new questions. The paper is acceptable for publication after further improvement based on my comments:

We thank the reviewer for their favorable assessment of our revised manuscript, which we agree was considerably improved because of reviewer feedback, and we appreciate these additional comments to further improve the manuscript.

1. Page 29: Half-life extension by albumin binding peptides/nanobodies have now been described. However, the use of the streptococcal G-derived albumin binding domain as HLE has still been ignored, while this is a very promising new development. See for instance doi:10.1093/protein/gzq058. The affinity of this ABD can be adjusted to adjust half-life: see for instance: DOI: 10.7150/thno.57510. Affibodies and humabodies have been used in combination with this ABD, doi: 10.3390/ijms21082999 and doi: 10.1016/j.neo.2023.100962.

We thank the reviewer for bringing this exciting albumin binding domain (ABD) to our attention and we apologize for inadvertently omitting this important advancement in our revised manuscript. We further thank the reviewer for providing these references regarding the *Streptococcal* G-derived albumin binding domain. We have reviewed these and believe that we have identified the first report to describe this approach, an excellent article by the group of Prof. Roland Kontermann (Stork et al., doi:10.1093/protein/gzm061). We have therefore revised the manuscript as followed and have also cited this important work as reference 33:

Page 4: “While several promising albumin-binding molecules have been described, including small molecules, fatty acids, peptides, and *Streptococcus* protein G-derived domains,^{27, 29, 33} we elected to build our platform from a nanobody with high affinity for albumin because nanobodies are modular and programmable via genetic engineering, are molecularly well-defined, are amenable to scalable industrial manufacturing, and are components of approved and clinically-advanced therapeutics, including ozoralizumab, which contains an anti-albumin nanobody domain.³¹”

Page 29: “An appealing feature of anti-albumin nanobodies, and other protein-based albumin hitchhiking agents (e.g., affibodies, humabodies, albumin binding domains derived from *Streptococcus* protein G) over other albumin binders (e.g., lipids, Evans blue) is the high degree of molecular programmability that can be achieved through protein engineering.”

2. Line 208: Puzzling for me are data from figure 1h,i,j : Why is there no difference in interferon activity (h,i), but a clear difference for interferon activated gene expression (j)?

We thank the reviewer for this interesting question. First, the cell types used in **Figure 1h,i,j** are all different and therefore may have different sensitivities to diABZI, either as a free agent or conjugated to the nanobody. Notably, the cells used in **Figure 1h,i** are human cell lines whereas the cells used in **Figure 1j** are primary murine macrophages (bone marrow-derived macrophages, BMDMs). For reasons that are unclear, we have found that BMDMs are not as sensitive to free diABZI as these cell lines, but are responsive to both nAlb-diABZI and AP-diABZI conjugates. This is further confirmed by data in **Figure 4f** where the DBCO-functionalized diABZI, which has similar activity to the free diABZI (e.g., **Figure 4d,e**), is also poorly active in BMDMs while all nanobody-diABZI conjugates are active. It is also notable that the assays used in **Figure 1h,i** are different than in **Figure 1j**, with the former utilizing a type-I interferon signaling luciferase reporter over a range of diABZI concentrations

and the latter using qRT-PCR at a single concentration. Therefore, the differences in activity observed may also be related to sensitivity of these assays and/or the diABZI concentrations used. Because of such factors, we were careful to only conclude that nanobody-diABZI conjugates could activate STING in primary immune cells of interest (i.e., BMDM and BMDC) without making claims that the conjugates were more active than the free compound since this may vary amongst cell types, and we believe that elucidating the relative sensitivity of all potentially relevant cell populations is beyond the scope of the current work.

3. Line 213-223: The description of figure 2 should start with the results of figure 2a instead of starting with a discussion about figure 2e.

We appreciate the reviewer's suggestion. While our intention was to broadly introduce potential mechanisms of albumin accumulation prior to discussing data in **Figure 2**, we understand that much of what was described here is more related to data in **Figure 2e**. Therefore, we have removed the text stated below from this section and have moved a revised version of this to the Discussion to address comment 4 below. As such, this section now begins with a discussion of data in **Figure 2a-d** pertaining to intracellular internalization of nAlb-diABZI.

The following text has been deleted from **Page 7**:

"While still incompletely understood and variable across cancers and patients²⁹, albumin can accumulate in tumor tissue through several interrelated mechanisms, including passive diffusion through leaky tumor vasculature, active transport via endothelial cell transcytosis, binding to SPARC (secreted protein acidic and rich in cysteine) produced by cancer cells, and cellular uptake and catabolism by cancer and tumor-associated immune cells such as macrophages.^{27, 29}"

4. Line 213-223: Tumor accumulation as a result of HLE is depicted in fig 2e,f. It is indeed interesting to find an explanation for this phenomenon but this should be done in the discussion section and not in the results section. Besides the mentioned EPR effect the most simple explanation has not been mentioned: due to HLE the nanobody is not immediately removed by the kidneys and now it has the time to be taken up by the cells via a non-specific pinocytosis route

We agree with the reviewer that this would be more appropriate for the Discussion and so we have revised the Discussion on **Page 29** as followed, which now also explicitly describes the EPR effect as a potential mechanism of increased drug accumulation at tumor sites:

Page 29: "While still incompletely understood, albumin can accumulate in tumor tissue through several interrelated mechanisms that are largely enabled by its long circulation time, including the enhanced permeability and retention (EPR) effect, active transport via endothelial cell transcytosis, binding to SPARC produced by cancer cells, and cellular uptake and catabolism by cancer and tumor-associated immune cells such as macrophages.^{27, 29} Indeed, our data demonstrate that albumin-hitchhiking dramatically increases nanobody and drug half-life, allowing nAlb-diABZI to accumulate in tumor tissue where it is internalized by cancer and tumor-associated immune cells to activate STING."

5. Figure 2: the fluorescent images need higher magnification.

Thank you for this suggestion. While there are three different fluorescent images in **Figure 2** (c, f, and j), because data in c and f are quantified in d and g, respectively, we assumed that the reviewer is referring to images in j which serve to confirm the presence of nAlb-Cy5 within tumor tissue. To respond to the reviewer's request, we have included a higher magnification of **Figure 2j** in the Supplemental Information as **Figure S11**. This higher magnification image is of sufficiently high resolution to allow the reader to digitally zoom into specific parts of the image for more detailed visualization of nAlb-Cy5 distribution within the tumor tissue.